# Theranostic near-infrared-IIb emitting nanoprobes for promoting immunogenic radiotherapy and abscopal effects against cancer metastasis

Hao Li [1,4], Meng Wang[2,4], Biao Huang[3], Su-Wen Zhu[1], Jun-Jie Zhou[1], De-Run Chen[1], Ran Cui [3✉], Mingxi Zhang [2✉] & Zhi-Jun Sun [1✉]

Radiotherapy is an important therapeutic strategy for cancer treatment through direct damage to cancer cells and augmentation of antitumor immune responses. However, the efficacy of radiotherapy is limited by hypoxia-mediated radioresistance and immunosuppression in tumor microenvironment. Here, we construct a stabilized theranostic nanoprobe based on quantum dots emitting in the near-infrared IIb (NIR-IIb, 1,500–1,700 nm) window modified by catalase, arginine–glycine–aspartate peptides and poly(ethylene glycol). We demonstrate that the nanoprobes effectively aggregate in the tumor site to locate the tumor region, thereby realizing precision radiotherapy with few side-effects. In addition, nanoprobes relieve intratumoral hypoxia and reduce the tumor infiltration of immunosuppressive cells. Moreover, the nanoprobes promote the immunogenic cell death of cancer cells to trigger the activation of dendritic cells and enhance T cell-mediated antitumor immunity to inhibit tumor metastasis. Collectively, the nanoprobe-mediated immunogenic radiotherapy can boost the abscopal effect to inhibit tumor metastasis and prolong survival.

[1] The State Key Laboratory Breeding Base of Basic Science of Stomatology (Hubei-MOST) & Key Laboratory of Oral Biomedicine Ministry of Education, School & Hospital of Stomatology, Wuhan University, 430079 Wuhan, China. [2] State Key Laboratory of Advanced Technology for Materials Synthesis and Processing, Wuhan University of Technology, 430070 Wuhan, China. [3] College of Chemistry and Molecular Sciences, Wuhan University, 430072 Wuhan, China. [4] These authors contributed equally: Hao Li, Meng Wang. ✉email: cuiran@whu.edu.cn; mxzhang@whut.edu.cn; sunzj@whu.edu.cn

Cancer, as a major public health problem, has been predicted in recent years to become the leading cause of death worldwide[1]. Radiotherapy (RT) is an important therapeutic strategy for cancer treatment and is applied in ~50% of patients who require curative-intent treatment for solid tumors[2]. However, some intrinsic shortcomings of RT still limit its clinical applications. When the radiation beam is applied locally to the tumor site, neighboring healthy tissues are also exposed to radiation, leading to normal tissue toxicity[3]. Therefore, accurate delineation of target tumor is prerequisite for precision RT. In addition, cancer cells usually resist X-ray radiation because of hypoxic conditions in the tumor microenvironment (TME), a common characteristic feature of most solid tumors[4,5]. Moreover, RT is generally prescribed for the treatment of localized tumors but ignores the evidence that the complex microenvironmental changes triggered by RT can also increase metastatic risks and counteract the long-term efficacy of the treatment[6,7].

Notably, RT not only exerts direct cytotoxic effects on cancer cells through DNA damage by ionizing radiation and oxidative damage but also augments immune responses against tumors which have been proven essential for radiation-induced tumor regression[8]. One common hypothesis is that localized RT initiates immunogenic cell death (ICD) of cancer cells and then induces presentation by dendritic cells (DCs), leading to prime antitumor immunity mediated by tumor-specific cytotoxic T cells[9,10]. More importantly, localized RT not only enriches the cytotoxic T cell infiltrate in tumor tissue resulting in regression of the primary tumor but also produces systemic tumor-directed immunity, leading to elimination of metastatic cancer in the non-irradiated field (called the abscopal effect) and long-term immunological memory, especially when combined with immunotherapy[11,12]. Previous studies indicated that the combination strategy of RT and immunotherapy via immune checkpoint blockade (ICB) provides an opportunity to boost abscopal response rates[11]. Unfortunately, even when combining RT with immunotherapy, the RT-induced antitumor immune response and abscopal effect are stifled by immunosuppression or tolerance in the TME[8,11]. Hypoxia is an important obstacle contributing to immunosuppression and facilitating tumor recurrence, metastasis, and resistance to RT[13,14]. Hypoxia-inducible factor 1α (HIF-1α) subsequently stimulated by hypoxia plays a central role in hypoxia-induced resistance to radiation treatment[13,15]. Hypoxic cancer cells can be resistant to RT directly by HIF-1α signaling[15]. Moreover, HIF-1α signaling can lead to damage to antigen presentation and resistance of cancer cells to T cell-mediated cytotoxicity[16]. In addition to activation of HIF-1α signaling, hypoxia in tumors can also attract immunosuppressive cells, such as regulatory T cells (Tregs) and M2-like macrophages, into the TME. M2-like macrophages generally demonstrate immunosuppression and promotion of cancer growth by suppressing T cell function[17–19]. Thus, relieving the hypoxic environment in the TME can effectively improve the RT-induced antitumor immune response and abscopal effect.

With the rapid development of nanotechnology, applications of nanoparticles to overcome resistance to RT and enhance radiation efficacy against tumors have been realized in many preclinical studies[20,21]. Several nanoparticles with high atomic number (high-Z) elements can act as radiosensitizers and promote radiation energy deposition to enhance RT-induced direct damage to cancer cells[22,23]. Nanoparticles were also used to increase the radiosensitivity of cancer cells and weaken the immunosuppressive effect in the TME by relieving intratumoral hypoxia through peroxide hydrogen ($H_2O_2$) decomposition and $O_2$ production[24–27]. However, it is very difficult to trace the biodistribution of these nanoagents in living bodies, which may cause difficulty in determining the scope and time window of RT. Quantum dots (QDs) emitting at the long end of the second near-infrared window (NIR-IIb, 1,500–1,700 nm) take advantage of

deep tissue imaging with high spatial and temporal resolution, and can be used in combination with tumor imaging and treatment[28,29]. Moreover, the NIR-IIb emitting QDs also contain high-Z elements (Pb, Ag), which have the potential ability for radiosensitizing. Therefore, based on QD nanotechnology, we can develop a multifunctional nanoprobe to resolve the challenge of enhancing the precision and efficacy of RT.

In this study, we rationally design a stabilized theranostic nanoprobe based on NIR-IIb-emitting QDs, which can precisely image tumor regions with high resolution, promote the radiosensitivity and immunogenicity of cancer cells, and relieve intratumoral hypoxia to enhance RT-based therapy strategies. NIR-IIb-emitting PbS/CdS QDs are modified with catalase (Cat), arginine–glycine–aspartate (RGD) peptides and poly(ethylene glycol) (PEG) (Fig. 1a)[30,31]. The resulting QD-Cat-RGD nanoprobes can effectively aggregate in the tumor site, led by the RGD peptide, and emit in the NIR-IIb window to ascertain the shape and position of the tumor and determine the optimal time for RT. Moreover, the QD-Cat-RGD nanoprobes show a beneficial performance as a radiosensitizer with a high-Z atom (Pb) in the nanoprobe, which not only promotes the direct killing effect of cancer cells but also enhances the immunogenicity of cancer cells via ICD to trigger a robust antitumor immune response. Cat, an efficient enzyme that promotes the rapid decomposition of $H_2O_2$, is modified on QDs to relieve intratumoral hypoxia enhancing the radiosensitization of cancer cells and reduce the infiltration of immunosuppressive cells to destroy the immunosuppression in the TME. This multifunctional nanoprobe-based immunogenic RT, combined with immunotherapy using anti-programmed cell death 1 antibodies (aPD-1), can efficiently promote RT-induced abscopal effects to inhibit tumor metastasis and prolong survival.

## Results

**Preparation and characterization of QD-Cat-RGD nanoprobes.** NIR-IIb PbS/CdS core-shell QDs emitting at 1600 nm were synthesized as previously described[30,31]. Oleyamine-branched poly(acrylic acid) (OPA) was coated on the surface of QDs to transfer them into aqueous solution. Then, 8-Arm PEG-NH$_2$ molecules, RGD peptides, and Cat were conjugated on the surface of OPA-modified QDs via N-(3-(dimethylamino)propyl)-N′-ethylcarbodiimide hydrochloride (EDC) chemistry[32]. The carboxylic groups on QDs were activated by EDC molecules and then reacted with the amine groups on Cat to form amide bonds. Transmission electron microscopy (TEM) images demonstrated the regular granular structure and good dispersity of the QD-Cat-RGD nanoprobes (Fig. 1b). Dynamic light scattering (DLS) further revealed that the average hydrodynamic size of QD-Cat-RGD in aqueous solution was $46.7 \pm 1.1$ nm (Supplementary Fig. 1a). In addition, the zeta potential of QD-Cat-RGD changed from $-51.33$ to $-1.12$ mV after conjugation (Supplementary Fig. 1b). The absorption spectrum and fluorescence emission spectrum of QD-Cat-RGD nanoprobes are shown in Supplementary Fig. 1c, d. The fluorescence quantum yield (QY) of the nanoprobe was calculated to be 2.2% (referenced to IR-26 dye with a QY of 0.05%).

To reveal the loading capacity of RGD and Cat on the QD-Cat-RGD nanoprobes, we used a standard BCA protein assay to quantify the unbound Cat after the conjugation and purification of nanoprobes and consequently obtained the amounts of Cat grafted on QDs. The molar ratio of grafted Cat onto QDs was approximately 2.5:1. We further used UV spectrometry to detect the absorbance at 220 nm to quantify the unbound RGD and obtain the amounts of RGD grafted on QDs[33]. We calculated that the molar ratio of grafted RGD to QDs was approximately 30:1.

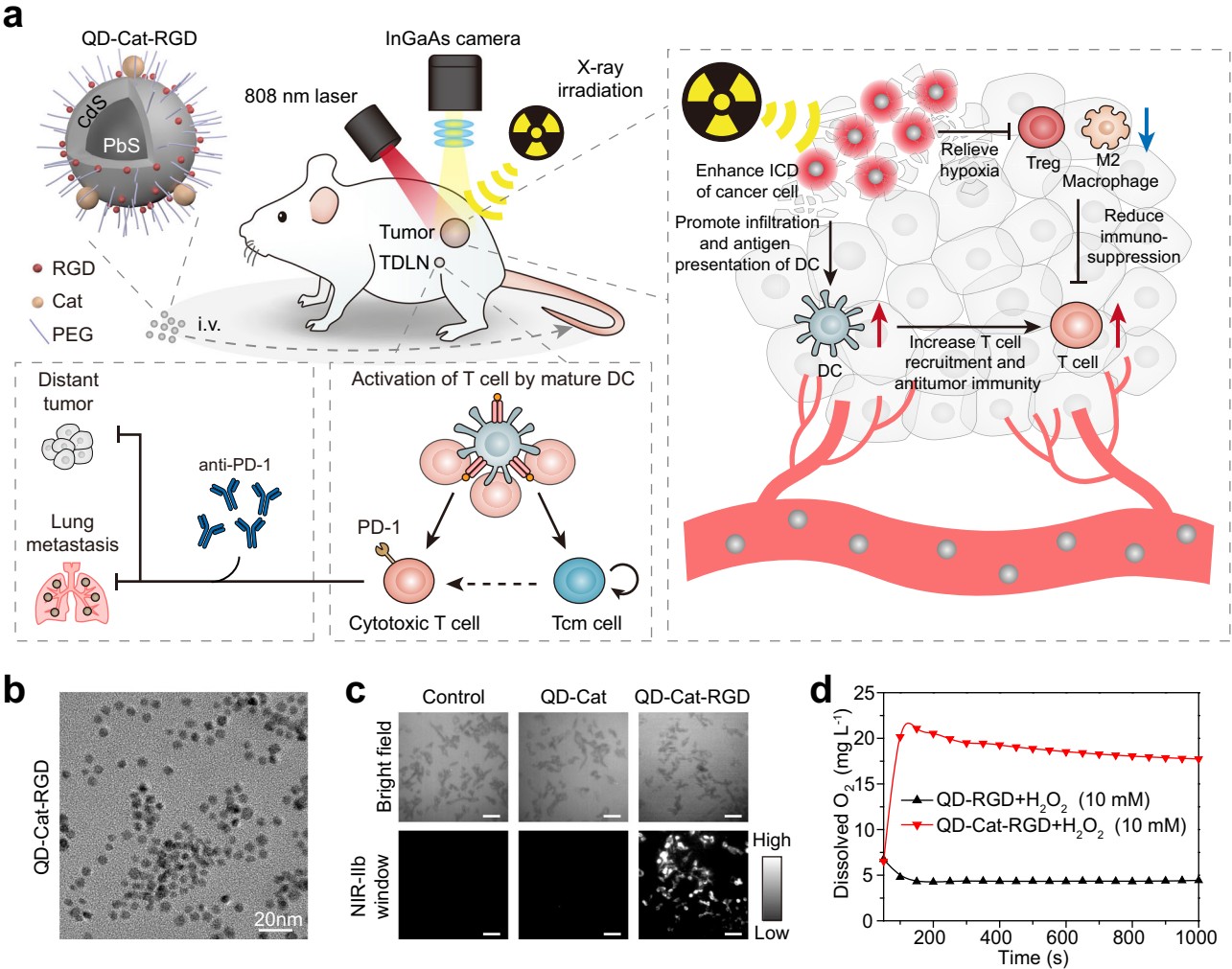

**Fig. 1 Schematic and characterization of QD-Cat-RGD. a** Schematic showing the mechanism by which QD-Cat-RGD nanoprobe-based RT boosts the antitumor immune response and abscopal effect to inhibit cancer metastases. **b** A TEM image of QD-Cat-RGD nanoprobes. Scale bar = 20 nm. **c** Images of 4T1 cells after incubation with PBS (control), QD-Cat (without RGD modification), or QD-Cat-RGD (with RGD modification) in the NIR-IIb window and bright field. Scale bars = 50 μm. **d** Dissolved oxygen generation in $H_2O_2$ solutions (10 mM) with QD-RGD (without Cat modification) or QD-Cat-RGD. The TEM of the QD-Cat-RGD nanoprobe (**b**), the cell imaging experiments (**c**), and $H_2O_2$ decomposition experiment (**d**) were representative of those generated from three independent experiments. RGD arginine–glycine–aspartate peptides, Cat catalase, PEG poly(ethylene glycol), ICD immunogenic cell death, Treg regulatory T cell, DC dendritic cell, Tcm central memory T cell, i.v. intravenous, PD-1 programmed cell death 1, TDLN tumor-draining lymph node, NIR-IIb near-infrared IIb. Source data are provided as a Source Data file.

We next tested the stability of QD-Cat-RGD nanoprobes. The fluorescence intensity of QD-Cat-RGD nanoprobes dispersed in fetal bovine serum (FBS) was reduced by less than 3% after continuous 808 nm laser exposure for 1 h (Supplementary Fig. 1e), which confirmed the good fluorescence stability of the QD-Cat-RGD nanoprobe. In addition, we intravenously injected QD-Cat-RGD nanoprobes into BALB/c mice and collected the feces for several days. As confirmed by the TEM image (Supplementary Fig. 2a), we found that QD-Cat-RGD nanoprobes remained intact even after being excreted from the body. Energy dispersive X-ray spectroscopy (EDS) analysis revealed nanoparticles extracted from the feces containing Pb, S, and Cd species (Supplementary Fig. 2b), which demonstrated the excellent in vivo stability of QD-Cat-RGD nanoprobes.

The RGD peptides can be recognized by integrin $\alpha_v\beta_3$, which is overexpressed in the tumor-associated vasculature and several types of cancer cells, such as 4T1 murine breast cancer cells and B16F10 murine melanoma cells[34,35]. With RGD modification, QD-Cat-RGD nanoprobes acquired the ability to target 4T1 cells (integrin $\alpha_v\beta_3$-positive) compared with the QDs without RGD

modification (Fig. 1c). $H_2O_2$ is produced by malignant cells and excessively accumulates in tumor tissue[20,36]. Cat coated on the surface of the QDs was used to produce $O_2$ and relieve hypoxia by catalytic decomposition of $H_2O_2$ in tumors. As measured by the oxygen probe, rapid generation of $O_2$ could be observed in $H_2O_2$ solution (10 mM) in the presence of QD-Cat-RGD (Fig. 1d). The above results demonstrated that the QD-Cat-RGD nanoprobes integrated multiple functions of imaging guidance, tumor targeting, and $O_2$ production.

**Imaging in the NIR-IIb window with high clarity**. The bright 1,600 nm fluorescence of PdS/CdS QDs is ideal for imaging in vivo because of its high spatial and temporal resolution[30]. To evaluate the image-guided properties of QD-Cat-RGD nanoprobes in vivo, 4T1 tumor-bearing BALB/c mice were intravenously injected with QD-Cat-RGD (150 μL, 2 mg mL$^{-1}$) or QD-Cat (150 μL, 2 mg mL$^{-1}$) nanoprobes. Then, a wide-field 2D InGaAs camera was used to collect emission photons between

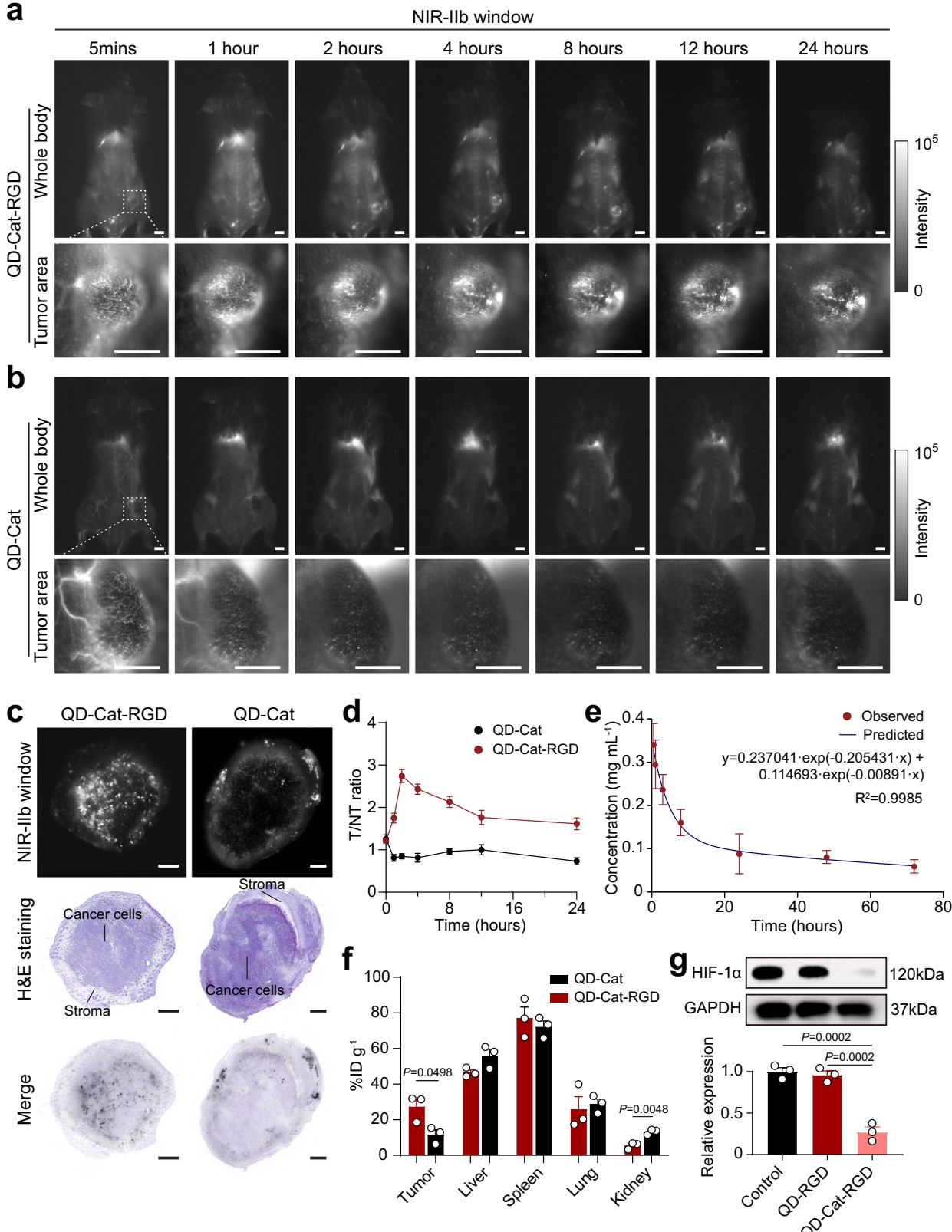

1,500 and 1,700 nm under 808 nm laser excitation at a power density of 25 mW cm$^{-2}$. After QD-Cat-RGD injection, the vasculature system of the mouse body was first imaged in the NIR-IIb window including the abnormal blood vasculature in the tumor (Fig. 2a). Due to the suppressed photon scattering and diminished autofluorescence in the NIR-IIb window, we could

monitor the change in the internal tumor region over time with high-resolution fluorescence imaging precise profiling (Fig. 2a). The strong signals focused at the tumor site confirmed the gradual infiltration of QD-Cat-RGD nanoprobes with an excellent tumor-targeting ability (Fig. 2a). However, the QD-Cat nanoprobes without RGD modification did not show this ability, with

**Fig. 2 In vivo fluorescence imaging in the NIR-IIb window and biodistribution of QD-Cat-RGD. a**, **b** Whole-body and high-magnification fluorescent images of 4T1 tumor-bearing mice intravenously injected with QD-Cat-RGD (**a**) or QD-Cat (**b**) nanoprobes at different time points (5 min, 1 h, 2 h, 4 h, 8 h, 12 h, 24 h) in the NIR-IIb window. Scale bars = 5 mm. The excitation power density was 50 mW cm$^{-2}$ provided by an 808 nm laser. **c** Serial sections of tumor tissue imaged in the NIR-IIb window or stained with H&E. Scale bars = 1 mm. The excitation power density for NIR-IIb fluorescence imaging was 25 mW cm$^{-2}$ provided by an 808 nm laser. The fluorescent images in the NIR-IIb window (**a**–**c**) were representative of those generated from three mice each group. **d** The T (tumor)/NT (nontumor) ratio of QD-Cat-RGD or QD-Cat at different time points (5 min, 1 h, 2 h, 4 h, 8 h, 12 h, 24 h). **e** The blood concentration of Pb in BALB/c mice at different time points (0.5, 1, 3, 8, 24, 48, 72 h) after intravenous injection with QD-Cat-RGD nanoprobes. **f** Delivery efficiency for injected dose (ID) of QD-Cat-RGD or QD-Cat in tumors and main organs (livers, spleens, lungs, and kidneys) of mice injected with QD-Cat-RGD or QD-Cat at 24 h p.i. **g** The protein expression level of HIF-1α in 4T1 tumor tissue in mice at 24 h after intravenous injection with PBS (control), QD-RGD (150 μL, 2 mg mL$^{-1}$), and QD-Cat-RGD (150 μL, 2 mg mL$^{-1}$) measured by western blot analysis. Western blot was done thrice. All data are shown as the mean ± s.e.m. of (n = 3) and n represents the number of independent samples. Statistical significance was calculated via two-tailed Student's t-test (**f**) and one-way ANOVA with Tukey's multiple comparisons test (**g**). NIR-IIb near-infrared IIb. Source data are provided as a Source Data file.

weak signals found in tumor tissue (Fig. 2b). The tumor-to-normal tissue (T/NT) signal ratio of QD-Cat-RGD increased sharply and peaked at approximately 2 h postinjection (p.i.) with a T/NT ratio of approximately 2.742, which was higher than the T/NT ratio of QD-Cat (Fig. 2d). These results can help to determine the optimum time for RT when the QD-Cat-RGD nanoprobes accumulate maximally in tumors compared with normal tissues to improve the efficiency of radiosensitizers. In addition to tumors, QD-Cat-RGD nanoprobes also accumulated in the liver, spleen, and lung 24 h p.i., with little retention in other organs including the heart and kidney (Supplementary Fig. 2c).

To explore the precise position of QD-Cat-RGD infiltrated in tumor tissue, serial slices of tumor tissue were imaged in the NIR-IIb window and stained with hematoxylin and eosin (H&E). The merged images showed that the QD-Cat-RGD nanoprobes with RGD modification could deeply infiltrate in the tumor tissue (Fig. 2c). Without RGD modification, QD-Cat nanoprobes were mainly found in the stroma which was infiltrated with supporting vasculatures (Fig. 2c). Moreover, we sorted the live CD45$^-$ cells which were mainly cancer cells from tumor tissue by fluorescent-activated cell sorting (FACS; the sorting gating strategy is shown in Supplementary Fig. 3a), and investigated the content of nanoprobe associated element Pb in these cells by inductively coupled plasma optical emission spectrometry (ICP-OES). The results showed that the content of Pb in the QD-Cat-RGD group were higher than that in QD-Cat group (Supplementary Fig. 3b), which suggested that the RGD modification significantly improved the targeting efficiency of nanoprobes to cancer cells in vivo. To further reveal the relationship between location of QD-Cat-RGD nanoprobes and hypoxic region in tumor, serial slices of tumor tissue from QD-Cat-RGD group were imaged in the NIR-IIb window, stained with H&E, and immunostained with HIF-1α by immunohistochemistry (IHC). As shown in Supplementary Fig. 4, the fluorescent signals in NIR-IIb can be found in the Region 1 in the tumor (Supplementary Fig. 4d), which was near the stroma infiltrated with several vasculatures (Supplementary Fig. 4e) and with weak nuclear HIF-1α immunostaining (Supplementary Fig. 4g). Moreover, the fluorescent signals in NIR-IIb can also be found in the Region 2 in the tumor (Supplementary Fig. 4d), which was away from vasculatures (Supplementary Fig. 4f) and with strong nuclear HIF-1α immunostaining (Supplementary Fig. 4h). These results suggested that QD-Cat-RGD nanoprobes with the targeting property of RGD might diffuse in the tumor tissue.

Here, we selected Pb element measured by ICP-OES to investigate the pharmacokinetics of nanoprobes and biodistributions in the main organs of the body, including the liver, spleen, lungs, and kidneys, as well as tumors after blood circulation. The blood circulation half-time ($T_{1/2}$) of QD-Cat-RGD was calculated to be approximately 3.37 h (Fig. 2e) according to a two-compartment model[37]. The systemic distribution of Pb revealed

a 17.67% injected dose (ID) g$^{-1}$ tumor accumulation efficiency of QD-Cat-RGD (Fig. 2f), which was higher than the 6.84% ID g$^{-1}$ tumor accumulation efficiency of QD-Cat (Fig. 2f). The above results demonstrated a good property of QD-Cat-RGD by comparing with the QD-Cat (without RGD) group.

We then tested the anti-hypoxia ability of QD-Cat-RGD nanoprobes in vivo. At 24 h after the injection of QD-Cat-RGD or QD-RGD (without Cat modification), the tumor tissues were harvested from the mice. The expression of HIF-1α in tumor tissue as an index of hypoxia was analyzed by western blot, which clearly showed that the expression level of HIF-1α was significantly reduced in the QD-Cat-RGD group compared with the QD-RGD group and control group (Fig. 2g). Together, this evidence confirmed that the well-designed nanoprobe could specifically target the tumor and relieve hypoxia in the TME for RT treatment.

**Reinforcement of ICD of cancer cells induced by RT.** Since QD-Cat-RGD based on PbS/CdS QDs contain high-Z element Pb and Cat which could relieve hypoxia via the catalysis of endogenous H$_2$O$_2$ in the TME, they have a strong potential to promote RT efficacy. To study the effect of nanoprobes and RT on cells under hypoxia, cells were incubated under hypoxic condition (37 °C, 1% O$_2$, 5% CO$_2$). To evaluate the QD-Cat-RGD nanoprobe's potential as a radiosensitizer for RT, a colony formation assay was conducted to assess the sensitivity of 4T1 cancer cells to different treatments in hypoxia in vitro. The results showed that the pro-liferative activation of 4T1 cells treated with QD-RGD- or QD-Cat-RGD-based RT in hypoxia significantly decreased compared with RT in hypoxia (Fig. 3a, b). The radiosensitization of QD-RGD and QD-Cat-RGD was further confirmed by p-H2AX staining, which reflected the DNA damage of cancer cells[38,39]. Consistent with the results of the colony formation assay, the fluorescence intensity of p-H2AX in the nuclei of 4T1 cells was significantly increased in the hypoxia + RT + QD-RGD and hypoxia + RT + QD-Cat-RGD groups compared with the RT group under hypoxic conditions (Fig. 3c). In addition, an annexin V/propidium iodide (PI) staining assay suggested that the number of vital 4T1 cells decreased in the hypoxia + RT + QD-RGD and hypoxia + RT + QD-Cat-RGD groups compared with the other groups (Fig. 3d). We also tested the radiosensitization of QD-Cat-RGD nanoprobes under normoxic conditions and found that the nanoprobes also had radiosensitizing effect under nor-moxic conditions (Supplementary Fig. 5).

Considering that QD-Cat-RGD nanoprobes significantly promoted the RT efficacy of directly killing cancer cells, we then evaluated the effect of QD-Cat-RGD-based RT on promoting the ICD of cancer cells. RT can trigger ICD of cancer cells through emission of tumor-associated antigen and ICD-associated danger-associated molecular patterns (DAMPs), such as exposure of calreticulin on the surface of dying cancer cells, and the release

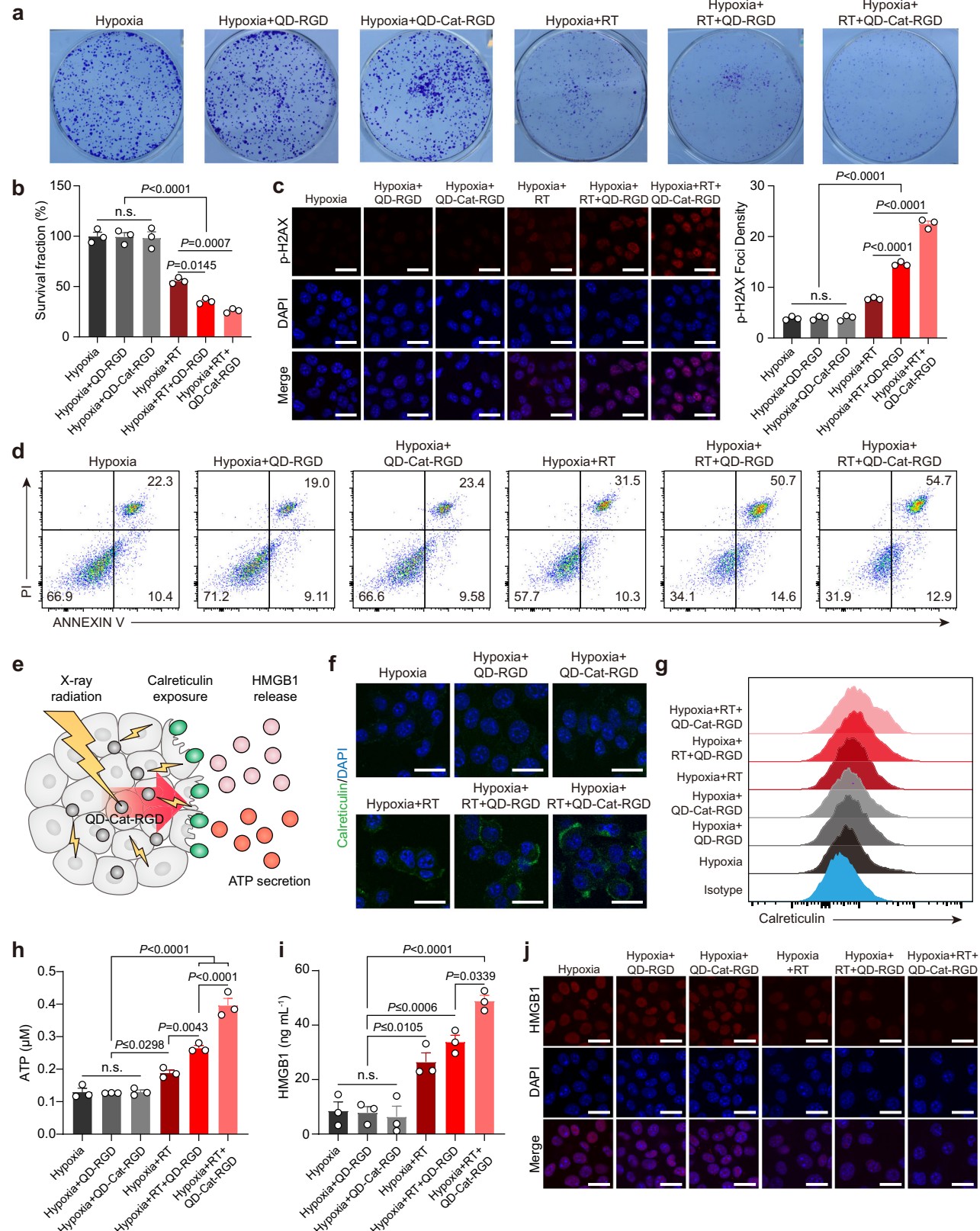

of large amounts of high-mobility group box 1 (HMGB1) and adenosine triphosphate (ATP) into the extracellular milieu, consequently resulting in activation of antitumor immunity (Fig. 3e)[9]. As measured by fluorescence imaging and flow cytometry, the surface expression of calreticulin in 4T1 cells in the hypoxia + RT + QD-RGD and hypoxia + RT + QD-Cat-

RGD groups was significantly increased compared with that in the 4T1 cells treated with hypoxia + RT only (Fig. 3f, g and Supplementary Fig. 6). We also tested ATP and HMGB1 release in cancer cells, two crucial signals during DC recruitment and maturation[9]. At 24 h after RT, 4T1 cells treated with QD-RGD- or QD-Cat-RGD-based RT under hypoxic conditions exhibited

**Fig. 3 QD-Cat-RGD as a radiosensitizer for enhancing ICD of cancer cells under hypoxic conditions.** 4T1 cells were incubated under hypoxic conditions (37 °C, 1% $O_2$, 5% $CO_2$) and irradiated with X-ray doses of 6 Gy in vitro. **a** Representative photographs of stained colonies of 4T1 cells treated with hypoxia, hypoxia + QD-RGD, hypoxia + QD-Cat-RGD, hypoxia + RT (6 Gy), hypoxia + RT + QD-RGD (6 Gy) or hypoxia + RT + QD-Cat-RGD (6 Gy) after 7 days. The photographs of stained colonies (**a**) were representative of those generated from three independent samples each group. **b** Histogram plot of the survival fraction of 4T1 cells with different treatments. **c** Immunofluorescence staining of p-H2AX (Ser139) and quantitative analysis of foci density of foci per cell at 4 h after different treatments. Scale bars = 25 μm. **d** Apoptosis analysis measured by flow cytometry of 4T1 cells at 24 h after different treatments. **e** QD-Cat-RGD reinforces the ICD induced by RT and emits several types of DAMPs from dying cancer cells. **f** Immunofluorescence staining of calreticulin of 4T1 cells with different treatments after 24 h. Scale bars = 25 μm. **g** Flow cytometry analysis of surface calreticulin expression in 4T1 cells with different treatments after 24 h. **h** Histogram plot of secreted ATP production by 4T1 cells with different treatments after 24 h. **i** Histogram plot of HMGB1 released by 4T1 cells with different treatments after 24 h. **j** Immunofluorescence staining of HMGB1 in 4T1 cells with different treatments after 24 h. Scale bars = 25 μm. The images of immunofluorescence staining (**c**, **f**, **j**) were representative of those generated from three independent samples each group. All data are shown as the mean ± s.e.m. of ($n = 3$) and $n$ represents the number of independent samples. Statistical significance was calculated via one-way ANOVA with Tukey's multiple comparisons test. n.s. not significant. Source data are provided as a Source Data file.

significantly increased ATP secretion (Fig. 3h) and HMGB1 release (Fig. 3i), as measured by ATP assay and enzyme-linked immunosorbent assay (ELISA). With HMGB1 release, the expression of HMGB1 in the cell nuclei of 4T1 cells in the hypoxia + RT + QD-RGD and hypoxia + RT + QD-Cat-RGD groups decreased compared with those only treated with hypoxia + RT (Fig. 3j).

Moreover, we further tested the effect of QD-Cat-RGD-based RT on promoting ICD in vivo. The 4T1 tumor-bearing BALB/c mice were intravenously injected with a single dose of QD-Cat-RGD (150 μL, 2 mg mL$^{-1}$) or QD-RGD (150 μL, 2 mg mL$^{-1}$). Then, the mice were treated with localized RT (12 Gy) at 2 h p.i. The tumor tissues were harvested at 24 h after the injection of QD-Cat-RGD or QD-RGD. The results showed that tumor tissue significantly increased the surface expression of calreticulin and ATP and HMGB1 concentrations (Supplementary Fig. 7a–c) at 24 h after the mice treated with QD-RGD- or QD-Cat-RGD-based RT, which was consistent with the in vitro results. This evidence demonstrated that QD-Cat-RGD containing high-Z element Pb and Cat could significantly promote RT efficiency and reinforce the ICD of cancer cells under hypoxic conditions, which could potentially promote more recruitment and maturation of DCs[40].

**Relieving hypoxia to overcome immunosuppression in the TME.** To examine the proposed immune activation of the combination of RT and the QD-Cat-RGD strategy, a mouse model of 4T1 breast cancer was used. When the tumors reached approximately 100 mm³, 4T1 tumor-bearing BALB/c mice were intravenously injected with a single dose of QD-Cat-RGD (150 μL, 2 mg mL$^{-1}$) or phosphate-buffered saline (PBS, 150 μL). Considering that NIR-IIb fluorescence imaging can show the area of tumors in mice after injection of QD-Cat-RGD, we first determined the location and range of RT with the help of NIR-IIb imaging (Supplementary Fig. 8a). Then, we used the lead shield to localize the range of RT determined by previous imaging (Supplementary Fig. 8b, c). The mice were treated with RT (12 Gy) at 2 h p.i. After 14 days, the lymphocytes in tumor tissues and tumor-draining lymph nodes (TDLNs) were analyzed by flow cytometry to demonstrate the immune-related function of QD-Cat-RGD in vivo (Fig. 4a). The tumor volume and weight results showed that tumors in the RT and RT + QD-Cat-RGD groups were significantly reduced compared with those in the control and QD-Cat-RGD groups (Fig. 4b, c and Supplementary Fig. 10). Compared with the RT group, the RT + QD-Cat-RGD group showed a stronger ability to inhibit tumor growth (Fig. 4b, c). We also monitored the weight changes of mice after treatment. After 14 days, the body weights of the mice in the different groups were no significantly difference (Fig. 4d).

We then tested the capacity of QD-Cat-RGD to relieve hypoxia in the TME. Hypoxia in the tumor site can lead to the recruitment of large numbers of macrophages and subsequently impact macrophage polarization by inducing macrophage differentiation towards an immunosuppressive M2-like phenotype (labeled by CD206) instead of an immunostimulatory M1-like phenotype (labeled by CD80) in the TME[17]. In our study, compared with the RT group, the RT + QD-Cat-RGD group showed a stronger ability to inhibit M2-like macrophages (CD206$^{high}$CD11b+F4/80$^{+}$) (Fig. 4e). Moreover, although there was no significant difference in the proportion of M1-like macrophages (CD80$^{high}$CD11b+F4/80$^{+}$) among these four groups (Supplementary Fig. 11a), the ratio of M1-like macrophages to M2-like macrophages was significantly increased in the RT + QD-Cat-RGD group compared with other the three groups (Fig. 4e). The presence of Tregs is also an important component for maintaining the hypoxia-induced immunosuppressive microenvironment[41]. We then tested Treg flow cytometry. Compared with the control group, Tregs (CD4$^{+}$CD25$^{+}$Foxp3$^{+}$) in the RT + QD-Cat-RGD group were more significantly reduced than those in the RT group (Fig. 4f and Supplementary Fig. 11c). The expression of HIF-1α may serve as a marker to reflect hypoxic conditions[15]. In the QD-Cat-RGD and RT + QD-Cat-RGD groups, reduced expression of HIF-1α in cell nuclei was observed in tumor tissues (Fig. 4g). This evidence showed that QD-Cat-RGD-based RT had a good ability to inhibit the growth of tumors and relieve intratumoral hypoxia to overcome immunosuppression in the TME.

**Elicitation of T cell-mediated antitumor immunity.** DCs are specialized antigen-presenting cells with essential roles in the generation of cancer-specific immunity triggered by different treatments[42]. ICD-driven DAMPs can attract DC recruitment and induce presentation of tumor-associated antigens by major histocompatibility complex (MHC)-II expression on DCs[9]. In addition, mature DCs express co-stimulatory molecules (CD80 and CD86) to induce the activation of T cells. Considering the significant increase in DAMPs and tumor antigens of cancer cells after QD-Cat-RGD-based RT, we then investigated whether combining RT with QD-Cat-RGD could enhance the recruitment and maturation of DCs. In the results, DC (CD11c$^{+}$) infiltration in tumors was significantly increased in the RT + QD-Cat-RGD group (Supplementary Fig. 11b). DCs infiltrating the tumor site can migrate to the TDLN to present tumor antigens and co-stimulatory receptors to T cells, eventually leading to the priming of T cell-mediated antitumor immunity[9]. Therefore, we tested the status of DCs in the TDLN, and the results showed that the co-stimulatory molecules on DCs (CD80$^{+}$CD86$^{+}$CD11c$^{+}$) and antigen presentation molecule on DCs (MHC-II$^{high}$CD11c$^{+}$)

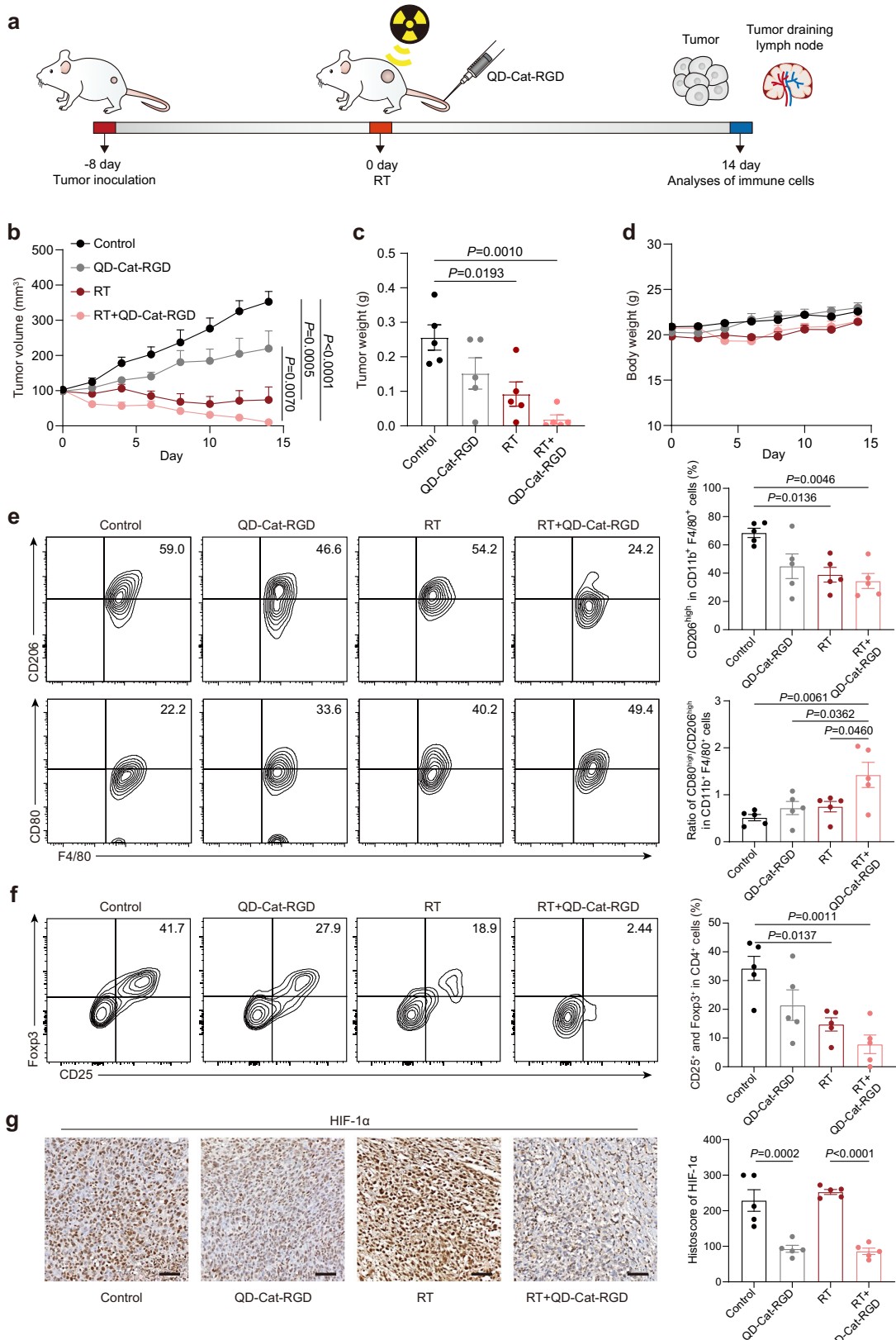

were significantly increased in the TDLN (Fig. 5a, b) after treatment with QD-Cat-RGD-based RT.

T cells, especially cytotoxic CD8$^+$ T cells, are generally considered to be the main effectors of efficient antitumor immunity triggered by DCs[42]. With the strong effect on recruitment and activation of DCs, QD-Cat-RGD-based RT

consequently caused an increase in tumor-infiltrated T cells (CD3$^+$) and cytotoxic CD8$^+$ T cells (CD8$^+$CD3$^+$) compared with RT alone (Fig. 5c, d and Supplementary Fig. 11d), which led to a strong T cell-mediated antitumor immune response[43]. Moreover, the central memory CD8$^+$ T cell populations (CD8$^+$ Tcm, CD44$^+$CD62L$^+$CD8$^+$) in the TDLN of the mice in the

**Fig. 4 QD-Cat-RGD-based RT for inhibiting tumor growth, relieving hypoxic conditions and decreasing immunosuppressive cells in the TME. a** Schematic showing the experiment using QD-Cat-RGD-based RT to treat mice bearing 4T1 tumors. BALB/c mice ($n = 5$ per group) were implanted subcutaneously with $7 \times 10^5$ 4T1 mammary carcinoma cells in the right hind flanks. When the tumor volumes were 100 mm$^3$, mice received different treatments. 4T1 tumors were harvested on day 14. **b** Tumor growth curves in different groups. Photographs of all tumors and tumor growth curves of tumors from individual mice are shown in Supplementary Fig. 10. **c** Histogram plot of tumor weight in different groups. **d** Body weight curve of different groups. **e** Representative flow cytometric analysis images of M2-like macrophages (CD206$^{high}$) and M1-like macrophages (CD80$^{high}$) gating on F4/80$^+$CD11b$^+$CD45$^+$ cells. The relative quantification of M2-like macrophages and the ratio of M1/M2 macrophages. **f** Representative flow cytometric analysis images and relative quantification of Tregs (CD25$^+$ and Foxp3$^+$) gating on CD4$^+$CD3$^+$CD45$^+$ cells. **g** Representative image of immunostaining of HIF-1α and relative quantification of the histoscore of immunostaining. Scale bars = 50 μm. The images of immunostaining (**g**) were representative of those generated from five mice each group. All data are presented as the mean ± s.e.m. ($n = 5$ per group) and $n$ represents the number of independent animals. Statistical significance was calculated via one-way ANOVA with Tukey's multiple comparisons test. Source data are provided as a Source Data file.

RT + QD-Cat-RGD group were significantly increased on day 14 after treatment compared with the control group (Fig. 5e), which suggested the potential of conferring durable remission and systemic abscopal effects[44,45]. This evidence demonstrated that treatment combining RT with QD-Cat-RGD could be considered as an immunogenic RT, which promoted maturation and antigen presentation of DCs and consequently led to more effective T cell-mediated antitumor immunity than RT alone.

To validate the contribution to antitumor effect of cytotoxic T cells in this combination therapy, CD8$^+$ T cells were depleted using antibodies against CD8 (aCD8) in 4T1 tumor-bearing mice treated with QD-Cat-RGD-based RT (Supplementary Fig. 12a). The results demonstrated that the antitumor effects of the combination treatment were abrogated with the depletion of CD8$^+$ T cells (Supplementary Fig. 12b–d). Treatment combining QD-Cat-RGD-based RT with aCD8 caused decreases in tumor-infiltrated T cells (CD3$^+$) and depletion of cytotoxic CD8$^+$ T cells (CD8$^+$CD3$^+$) (Supplementary Fig. 12e,f), confirming that the antitumor effect in primary tumor was mainly induced by CD8$^+$ cytotoxic T cells in the TME.

For exploring the effect from QD-Cat-RGD for alleviating hypoxia and relieving immunosuppression, we added QD-RGD as a control group which was not modified with Cat and had no oxygenation ability. The results showed that QD-Cat-RGD could delay tumor growth (Supplementary Fig. 13a, b) when RT did not apply. However, tumor volumes in the QD-RGD group were not significantly decreased, which implied an antitumor effect of relieving hypoxia by Cat. Furthermore, a significant decrease in the M2-like macrophage infiltration (Supplementary Fig. 13c) and a slight decrease in the Treg infiltration in the TME (Supplementary Fig. 13f) were found in the QD-Cat-RGD group. The intratumoral level of interleukin-10 (IL-10), an immunosuppressive cytokine predominantly secreted by M2 macrophages, was also found to decrease in the QD-Cat-RGD group (Supplementary Fig. 13g). DC and T cell infiltrations were found to increase in the TME after treatment with QD-Cat-RGD (Supplementary Fig. 13j, k). These results revealed that Cat modified on QDs could independently play a role in inhibiting tumor growth by reducing immunosuppressive cell infiltration and increasing T cell infiltration in the TME. These findings were consistent with previous reports demonstrating that alleviating hypoxia could reduce the infiltration of immunosuppressive cells in the TME, which delayed tumor growth[26,46,47]. We next evaluated the synergistic antitumor effect of alleviating immunosuppression by relieving hypoxia and reinforcing RT efficiency by QD-Cat-RGD nanoprobes when combined with RT. We added a group that combined RT with QD-Cat (without Cat modification) and a group that QD-Cat-RGD injection was after RT treatment as comparable groups to explore the synergistic effect in suppressing tumor growth. The combination of alleviating immunosuppression and reinforcing RT efficiency exhibited a synergistic effect in suppressing tumor growth (Supplementary

Fig. 13d). RT + QD-Cat-RGD (after RT injection) and RT + QD-RGD slightly suppressed tumor growth but no significance was found when compared with the RT group (Supplementary Fig. 13d). M2-like macrophage infiltration and intratumoral level of IL-10 in the TME decreased after treatment with QD-Cat-RGD-based RT (Supplementary Fig. 13e, i). Moreover, DCs, T cells, and CD8$^+$ cytotoxic T cells in the TME were significantly increased in the RT + QD-Cat-RGD group (Supplementary Fig. 13l, m, o). These results suggested that Cat and high-Z element Pb in QD-Cat-RGD exhibited a synergistic effect in RT for suppressing tumor growth through inhibiting immunosuppression and facilitating cytotoxic T cell-mediated antitumor immunity.

**The abscopal effect boosted by nanoprobe-mediated RT and ICB.** ICD driven by RT and the activation of a tumor-specific T cell-dependent immune response result in the regression or stabilization of distant untreated tumors (the abscopal effect). Many studies indicate that RT provides an opportunity to increase abscopal response rates when combined with ICB against checkpoints such as programmed cell death 1 (PD-1) and cytotoxic T-lymphocyte-associated protein 4 (CTLA-4)[48,49]. However, the abscopal effect can still be stifled by hypoxic conditions and immunosuppression in the TME even when combining RT with immunotherapy[11]. Several types of immunosuppressive cells infiltrating the TME before or after RT can limit the abscopal response[11]. Considering that QD-Cat-RGD had the capabilities of relieving hypoxia and decreasing tumor infiltrating immunosuppressive cells as well as increasing CD8$^+$ Tcm in the TDLN, we further investigated the efficacy of the QD-Cat-RGD nanoprobe for inducing robust abscopal effects.

To investigate whether QD-Cat-RGD survived and reinforced the abscopal effect induced by localized RT combined with immunotherapy, we used an antibody against PD-1 (aPD-1) as an ICB to reveal this process. Then, we built a metastatic tumor model to reveal the abscopal effect. 4T1 cancer cells were first injected into the right hind flank of mice on day −8 as primary tumors and subsequently injected into the left hind flanks on day −6 as a distant tumor mimic (Fig. 6a). On day 0, the mice were intravenously injected with QD-Cat-RGD (150 μL, 2 mg mL$^{-1}$), and the primary tumors were locally treated with RT (12 Gy) at 2 h p.i. Then, the mice were treated with aPD-1 on day 1, 3, and 5 (Fig. 6a). Tumor growth curves showed that the primary tumor (right side) was inhibited by QD-Cat-RGD-based RT combined with aPD-1 treatment (Fig. 6b and Supplementary Fig. 14). Additionally, the growth of distant tumors (left side) was inhibited (Fig. 6c and Supplementary Fig. 15). Consistent with these results, the numbers of intratumoral T cells (CD3$^+$) and cytotoxic T cells (CD8$^+$CD3$^+$) were significantly increased in distant tumors of the RT + QD-Cat-RGD + aPD-1 group (Fig. 6d–f). Granzyme B, interferon-γ (IFN-γ), and tumor necrosis factor-α (TNF-α), cytotoxic effectors mainly expressed

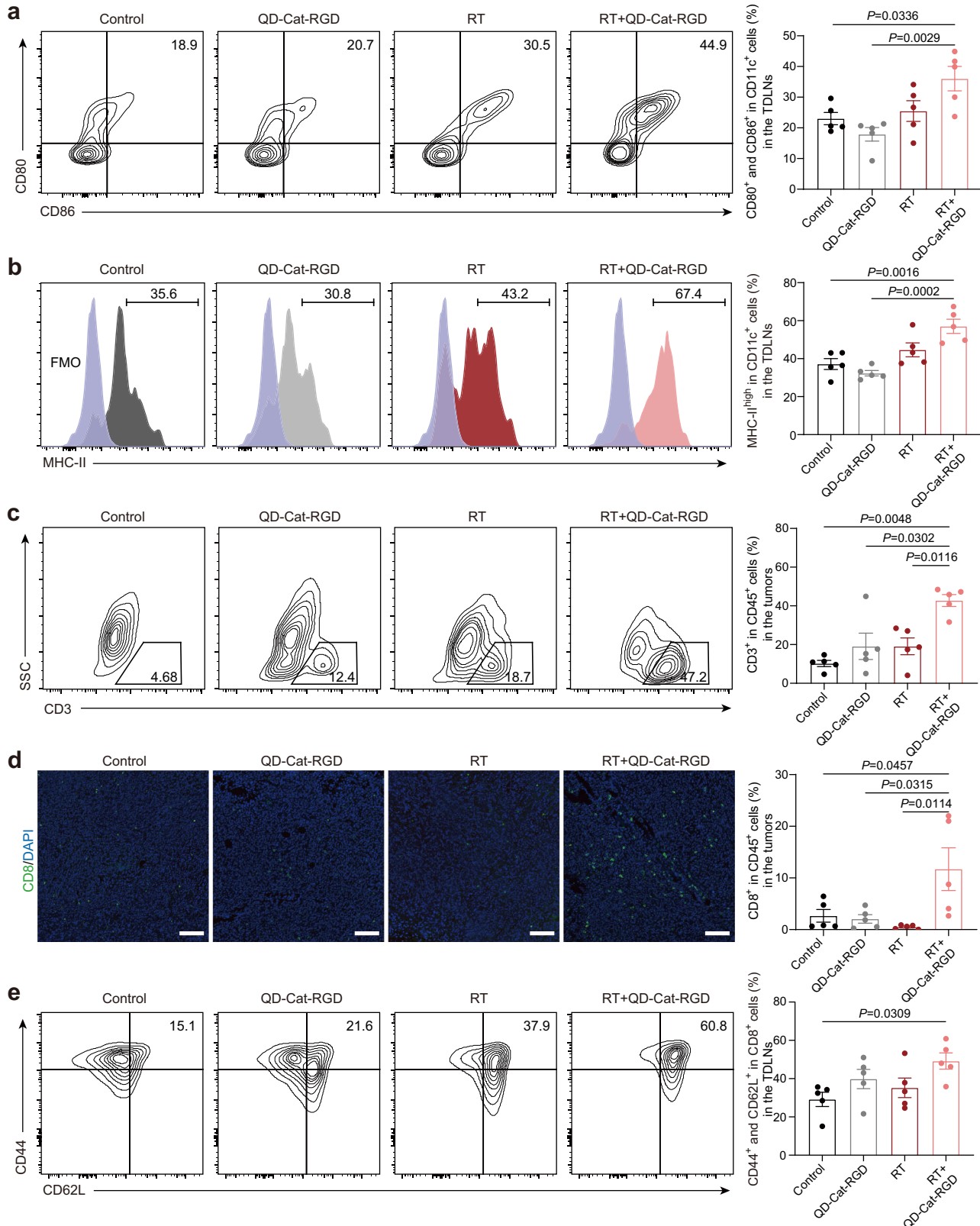

by cytotoxic T cells, were also increased in distant tumors of the RT + QD-Cat-RGD + aPD-1 group, as measured by IHC and ELISA (Fig. 6g–i). We further investigated the mechanism contributing to the suppression of distant tumor. Encouraged by the results that QD-Cat-RGD could relieve hypoxic condition and reduce the infiltration of immunosuppressive cells in TME

which could hinder antitumor immunity mediated by cytotoxic T cells[50,51], we tested the changes of Tregs, M2-like macrophages, and IL-10 in the TME of distant tumor. QD-Cat-RGD could also work in distant tumors and the intratumoral M2-like macrophages, Tregs, and the level of IL-10 were decreased in distant tumors in the RT + QD-Cat-RGD + aPD-1 group

**Fig. 5 QD-Cat-RGD-based RT for promoting DC maturation and antigen presentation in TDLNs and consequently inducing T cell-driven antitumor immunity. a** Representative flow cytometric analysis images and relative quantification of co-stimulatory molecules CD80+CD86+ gating on CD45+ CD11c+ cells in TDLNs. **b** Representative flow cytometric analysis images and relative quantification of MHC-II expression on CD11c+CD45+ cells in the TDLNs. **c** Representative flow cytometric analysis images and relative quantification of CD3+ T cells gating on CD45+ in tumors. **d** Representative immunofluorescence images of tumors showing CD8+ T cells and relative quantification of CD8+CD3+ T cells gating on CD45+ cells in tumors by flow cytometry. Scale bars = 50 μm. The immunofluorescent images were representative of those generated from three mice each group. **e** Representative flow cytometric analysis images and relative quantification of CD44+CD62L+ Tcm cells gating on CD8+CD3+CD45+ cells in the TDLNs. All data are presented as the mean ± s.e.m. ($n = 5$ per group) and $n$ represents the number of independent animals. Statistical significance was calculated via one-way ANOVA with Tukey's multiple comparisons test. FMO fluorescence-minus-one. TDLN tumor-draining lymph node. Source data are provided as a Source Data file.

(Supplementary Fig. 16a–c). This evidence demonstrated that QD-Cat-RGD-based RT combined with aPD-1 could intensely boost the abscopal effect and inhibit the growth of distant tumors.

Considering that tumor metastasis is responsible for approximately 90% of cancer-related mortality, preventing the spread of metastasis is beneficial for prolonging the overall survival of patients[52]. Encouraged by the increase in the abscopal effect in the subcutaneous tumor model, which implied strong systemic antitumor immunity, we evaluated this therapeutic strategy to inhibit early metastasis and prolong survival using an experimental model of metastasis at an early stage[53–55]. To establish the model of metastasis at an early stage, 4T1 cancer cells stably transfected with the luciferase gene (4T1-Luc) were injected into the right hind flank of mice as primary tumors. When the volume of the primary tumors reached approximately 100 mm³, 4T1-Luc cancer cells were then intravenously injected into the mice to mimic the escape of tumor cells from the primary tumor site into the circulation at an early stage and seeding in distant organs especially the lung. On the following day, the mice were intravenously injected with PBS or QD-Cat-RGD and then underwent surgery or RT to eliminate the primary tumors. Next, the mice were treated with aPD-1 on days 1, 3, and 5 (Fig. 7a). The growth of metastatic tumors was monitored by bioluminescence imaging. The results showed that lung metastases were found in most mice treated with surgery, even those mice treated with QD-Cat-RGD, aPD-1, or the combination treatment of QD-Cat-RGD and aPD-1 (Fig. 7b). In contrast, mice treated with QD-Cat-RGD-based RT exhibited a significantly slowed tumor growth of the lung metastases, while the combination with aPD-1 further reduced lung metastases (Fig. 7b). The survival analysis showed that 80% of mice could survive for 90 days in the RT + QD-Cat-RGD + aPD-1 group (Fig. 7c). Only 20% of mice survived for 90 days in the surgery + QD-Cat-RGD + aPD-1, RT + QD-Cat-RGD, and RT + aPD-1 groups (Fig. 7c). The survival rate of mice in the control, QD-Cat-RGD, aPD-1, QD-Cat-RGD + aPD-1, and RT groups reached 0% before day 50. These results further confirmed that QD-Cat-RGD-based RT combined with immunotherapy could efficiently inhibit metastasis at an early stage and prolong overall survival.

**Combination therapy against B16F10 melanoma.** Based on the above results, QD-Cat-RGD-based RT could induce an effective antitumor immune response in the 4T1 breast cancer model. Therefore, we wondered whether QD-Cat-RGD could potentially be used for other tumor types. Based on the targeting ability of RGD, we chose the B16F10 (integrin $\alpha_v\beta_3$-positive) murine melanoma model as the next tested model[35]. In the B16F10 tumor-bearing mouse model, we found that tumor growth was inhibited by the combination of RT and QD-Cat-RGD (Fig. 7d and Supplementary Fig. 17), consistent with the results obtained from the 4T1 breast cancer model. In addition, the QD-Cat-RGD-based RT resulted in significantly prolonged survival of mice relative to mice treated with PBS (control), QD-Cat-RGD, and RT (Fig. 7e). These results indicated that the combination

strategy of QD-Cat-RGD-based RT can be extended to other types of tumors.

**The cytotoxicity and biosafety of QD-Cat-RGD.** We next evaluated the potential cytotoxicity and biosafety of QD-Cat-RGD nanoprobes. The cytotoxicity of QD-Cat-RGD was measured by Cell Counting Kit-8 (CCK8) assay. The viability of 4T1 cancer cells and normal human immortalized oral epithelial cells (HIOECs) was not significantly affected after incubation with different concentrations of QD-Cat-RGD for 8 h (Supplementary Fig. 18). We also investigated the in vivo biosafety of the injected nanoprobes. H&E staining results for the hearts, livers, spleens, lungs, and kidneys in treated mice showed no abnormality in cellular morphology compared with control mice (Supplementary Fig. 19). We additionally performed routine blood tests and blood biochemical tests 10 days after intravenous injection of the QD-Cat-RGD nanoprobes. Routine blood tests and functional indicators of the liver and kidney, including alanine aminotransferase (ALT), aspartate aminotransferase (AST), carbamide (UREA), and creatinine (CREA), showed no significant difference between the control group and the QD-Cat-RGD injection group (Supplementary Table 1). These results demonstrated that the well-designed QD-Cat-RGD showed no observable cytotoxicity or in vivo toxicity.

## Discussion
RT is closely associated with the role of the immune system in the treatment of tumors[8]. At present, an increasing number of studies on RT and nanotechnology have gradually focused on influencing individual's immune system to treat cancer[56–59]. In addition, several studies claimed that decomposing hydrogen peroxide to relieve hypoxia results in increased the radiosensitivity of cancer cells and weakens the immunosuppressive effect in the TME[24–26]. Recently, the combination of RT and immune checkpoint therapies resulted in almost complete regression of primary tumors and inhibited the development of distant tumors. However, there are still a number of patients who are not responsive or sensitive to this treatment strategy. An important mechanism of radioresistance is due to various inhibitory immune cells in the TME, such as Tregs and M2-like macrophages. Therefore, we urgently need a platform to simultaneously enhance the efficiency of RT and inhibit the immunosuppressive TME to boost antitumor immunity and the systemic abscopal effect caused by the combination of RT and immunotherapy.

In this work, a theranostic nanoprobe was developed by using NIR-IIb emitting QDs for the radioimmunotherapy of cancer. The NIR-IIb emitting and targeting ability of QD-Cat-RGD nanoprobes enabled the in vivo imaging with a high resolution, which could clearly discriminate the margin of the tumor from the surrounding tissues to guide the range of the tumor in RT. Moreover, the QD-Cat-RGD nanoprobe was able to enhance the radiotherapeutic efficacy and immunogenicity of cancer cells via ICD, resulting in a robust immune response after RT. Cat coated on the nanoprobe

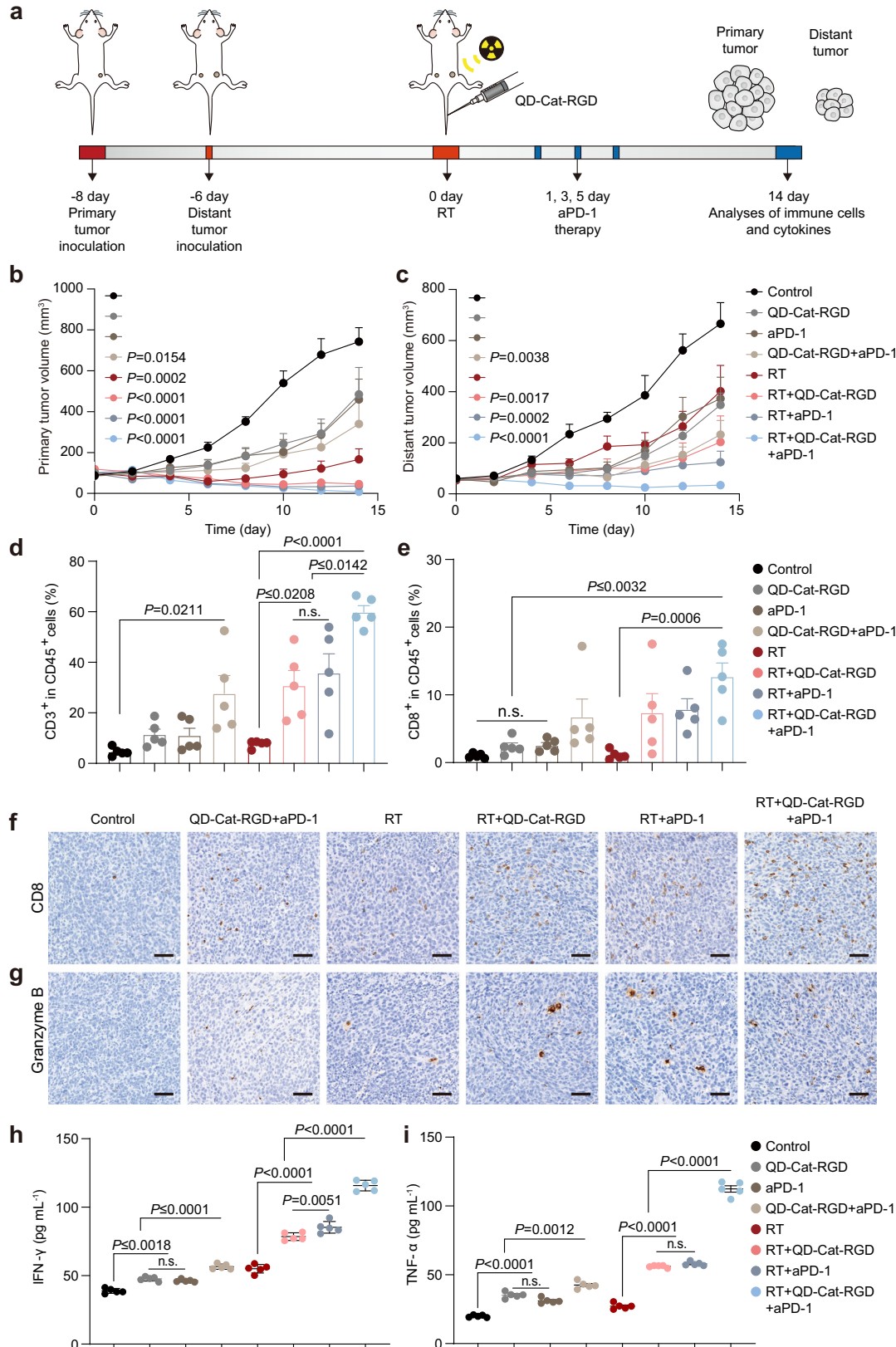

could relieve hypoxic conditions in the tumor site to enhance the radiosensitization of cancer cells and reduce the immunosuppressive TME, which is considered to be a major obstacle for antitumor immunity. In addition, combining nanoprobe-mediated RT with immunotherapy could remarkably boost the abscopal effect leading to a systemic antitumor immune response and elimination of distant metastases. The potential toxic elements in QD-Cat-RGD may limits the clinical transformation of QD-Cat-RGD in RT. This strategy still provides good prospects for the future RT to enhance the therapeutic precision and efficacy of RT and promote antitumor immune response, prolonging survival and improving the quality of life of patients with cancer.

**Fig. 6 QD-Cat-RGD-based RT in combination with aPD-1 for boosting the abscopal effect. a** Schematic showing the experiment using QD-Cat-RGD-based RT combined with aPD-1 to treat mice bearing 4T1 tumors on both sides. The tumor on the right side was designated as the primary tumor for in situ radiation therapy, and the tumor on the left side was designated as a distant tumor. BALB/c mice ($n = 5$ per group) were implanted subcutaneously with $7 \times 10^5$ 4T1 mammary carcinoma cells in the right hind flanks and after 2 days consequently implanted subcutaneously with $7 \times 10^5$ 4T1 cells on left hind flanks. When the primary tumor volumes were 100 mm$^3$, the mice received different treatments. 4T1 tumors were harvested on day 14. **b** Tumor growth curves of primary tumors on the right side in the different groups. The $P$ values are shown to compare with the control group. Photographs of all primary tumors and tumor growth curves of tumors from individual mice are shown in Supplementary Fig. 14. **c** Tumor growth curves of distant tumors on the left side in the different groups. The $P$ values are shown to compare with the control group. Photographs of all distant tumors and tumor growth curves of tumors from individual mice are shown in Supplementary Fig. 15. **d** Quantification of flow cytometric analysis of CD3$^+$ T cells gating on CD45$^+$ in the distant tumors of the different groups. **e** Quantification of flow cytometric analysis of CD8$^+$CD3$^+$ T cells gating on CD45$^+$ in distant tumors of the different groups. **f, g** Representative image of immunostaining of CD8 and granzyme B in distant tumors in the different groups. Scale bars = 50 μm. The images of immunostaining (**f, g**) were representative of those generated from three mice each group. **h, i** Cytokine levels of IFN-γ and TNF-α in distant tumors after different treatments as measured by ELISA. All data except **f** and **g** are presented as the mean ± s.e.m. ($n = 5$ per group) and $n$ represents the number of independent animals. Statistical significance was calculated via one-way ANOVA with Tukey's multiple comparisons test. n.s. not significant. Source data are provided as a Source Data file.

## Methods

**Chemicals**. Lead chloride (PbCl$_2$, 99%, powder) and cadmium oxide (CdO, 99.99%, power) were purchased from Alfa. Sulfur (S, 99.999%, powder) was purchased from Adamas. Oleylamine (OLA, technical grade, 70%), oleic acid (OA, technical grade, 90%), 1-octadecene (ODE), 2-(N-morpholino) ethanesulfonic acid (MES) hydrate, poly(acrylic acid) (Mw ~1,800), N,N′-dicyclohexylcarbodiimide (DCC), N-(3-dimethylaminopropyl)-N′-ethylcarbodiimide hydrochloride (EDC), and catalase (Cat) were purchased from Sigma-Aldrich. Phosphate buffered saline (PBS) was purchased from Hyclone. 8-Arm PEG-amine (Mw ~40 K) was purchased from Advanced BioChemicals. The cyclo(RGDyK) (97%) was purchased from China Peptides. Other reagents were of analytical grade. Ultrapure water was used in all needed experiments.

**Synthesis of PbS/CdS QDs**. PbS/CdS core-shell quantum dots (QDs) were prepared as previously described[30]. S (5 mmol, 0.08 g) and OLA (7.5 mL) were mixed in a two-neck flask at 120 °C for 30 min under a vacuum of 0.1 MPa. PbCl$_2$ (3 mmol, 0.834 g) and OLA (7.5 mL) were mixed in a three-neck flask at 120 °C for 30 min and then heated to 160 °C under a vacuum of 0.1 MPa. The S and PbCl$_2$ were dissolved completely to form S-OLA and Pb-OLA. S-OLA (2.25 mL) was quickly injected into the Pb-OLA under stirring at 160 °C and kept the condition for 1 h. Subsequently, the temperature was quickly cooled down to room temperature, and the growth reaction was terminated by introducing hexane (10 mL) into the reaction solution. The PbS QDs were separated and purified by adding excess ethanol and OA, collected via centrifugation. After centrifugation of the suspension and decantation of the supernatant, the PbS QDs were re-suspended in ODE. Next, CdO (9.2 mmol, 1.2 g), OA (8 mL), and ODE (20 mL) were heated to 200 °C under a vacuum of 0.1 MPa. Subsequently, the temperature cooled down to 100 °C and stabled for 30 min, PbS QDs (5 mL) was bubbled with Ar for 5 min, and then injected into the Cd precursor under stirring at 100 °C and kept the condition for 30 min. The growth reaction was terminated by introducing hexane (5 mL) into the reaction solution. PbS/CdS QDs were precipitated with excess ethanol and then re-dispersed in hexane.

**Preparation of QD-Cat-RGD nanoprobes**. The as-synthesized QDs were modified as previously described[60]. Polyacrylic acid (average Mw ~1,800, 0.9 g) and DCC (1.56 g) were mixed into a round-bottom flask. N,N-dimethylformamide (DMF) (10 mL) was added to dissolve the mixture. Subsequently, OLA (1.2 mL, molar ratio of OLA to PAA is 30%) were added dropwise into the reaction flask. The mixture solution was stirred overnight, and 0.5 M HCl (50 mL) were added to the mixture solution. The precipitate was isolated by centrifugation and re-dissolved in methanol (3 mL). Then, 1 M HCl (20 mL) was added to the solution and the precipitate was isolated by centrifugation. The precipitate was dissolved in chloroform (5 mL) and washed by 1 M HCl (10 mL). Collecting and drying over the organic phase with anhydrous Na$_2$SO$_4$. Under a vacuum, chloroform was removed and the oleyamine-branched poly(acrylic acid) (OPA) was collected (with an average Mw of ~3,000 determined by gel permeation chromatography). Subsequently, PbS/CdS QDs (5.0 mg) were added to mixture solution (15 mg OPA dissolved in 2.0 mL chloroform). The mixture was stirred for 30 min at room temperature and the solvent was removed by a rotary evaporator under a vacuum. Under the sonication, the products were dissolved completely in 2 mL of 50 mM sodium carbonate solution. The QDs were precipitated for 1 h at 416,000g by ultracentrifuge, and dissolved in 2 mL pH 8.5 MES buffer (0.01 M). The 8-Arm PEG-amine molecules, RGD peptides, and Cat were dissolved in 500 μL pH 8.5 MES buffer (0.01 M) and gradually added to the OPA-modified QD solution with stirring. EDC (15 mg) were dissolved in 150 μL pH 8.5 MES buffer (0.01 M), and gradually added to the mixture solution with stirring and reacted overnight. The QD-Cat-RGD nanoprobes were precipitated with ultra-centrifugation at 416,000g for 30 min to remove excess reactants. The precipitate was dissolved in 1× PBS and stored at 4 °C.

**Nanoprobe characterization**. TEM images were collected using a JEM-2100F electron microscope (JEOL) (operating at a 200 kV accelerating voltage). The zeta potentials and DLS data were recorded on a Malvern Nano-ZS ZEN3600 zetasizer. EDS data were obtained by using an EDX spectrometry (EDAX Inc.) equipped on a JEM 2010F microscope. The generation of O$_2$ was measured by an oxygen probe (JPBJ-608 portable Dissolved Oxygen Meters, Shanghai INESA Scientific Instrument Factory). The spectra were obtained from a Lambda 750 S UV-Vis-NIR spectrometer (Malvern). Photoluminescence spectra were measured using a FLS1000 fluorescence spectrometer with a Xenon lamp exciter and a PM1700 near infrared photomultiplier detector (Edinburgh Instruments).

**Fluorescence imaging of cells and animals in the NIR-IIb window**. For NIR-IIb imaging, cells were coincubated with QD-Cat (300 μg mL$^{-1}$) or QD-Cat-RGD (300 μg mL$^{-1}$) in PBS for 30 min followed by washing with PBS three times. The cells were fixed with 4% paraformaldehyde for 30 min. Finally, cells in the NIR-IIb window were imaged by a ×10 Mitutoyo Plan Apo NIR infinity corrected objective (Edmund Optics). Fluorescence images in the NIR-IIb window were imaged by a two-dimensional InGaAs camera (NIR vana, Princeton Instruments) and Light-Field software v.6.4 (Princeton Instruments)[31,61].

For mouse NIR-IIb imaging, 4T1 tumor-bearing mice were intravenously injected with 150 μL QD-Cat-RGD (2 mg mL$^{-1}$). NIR-IIb fluorescence images of the mouse body were recorded by a NIR-II small animal InGaAs camera (NIR vana, Princeton Instruments). The excitation power density was 25 or 50 mW cm$^{-2}$ provided by an 808 nm laser as previously described[31,61]. The exposure time for NIR-IIb fluorescence imaging was 200 ms.

**In vivo pharmacokinetics and biodistribution of QDs**. The 4T1 tumor-bearing mice intravenously injected with QD-Cat-RGD (150 μL, 2 mg mL$^{-1}$) or QD-Cat (without RGD) (150 μL, 2 mg mL$^{-1}$) nanoprobes were used for pharmacokinetics and biodistribution studies. At 0.5, 1, 3, 8, 24, 48, and 72 h p.i., 50 μL blood from treated mouse was collected. By using ICP-OES to calculate the concentration of Pb in blood to estimate the percentage of the QD-Cat-RGD in blood. The tissues including spleen, liver, lung, kidney, and tumor were collected, weighed, and then dissolved completely in 5 mL digest solution (HNO$_3$:H$_2$O$_2$ = 4:1) for 2 h until the solution turned to clear. Subsequently, the solutions were diluted by ultrapure water to 10 mL and analyzed by ICP-OES to measure the concentration of Pb in the resulting solutions. The calculation formula of % ID g$^{-1}$ is (the amount of Pb element in organs) × (the amount of Pb element in ID × the weight of organs)$^{-1}$ × 100%. The pharmacokinetic parameters were calculated by the software PKSolver (v. 2.0)[62].

**Cellular experiments**. The murine breast cancer cell line 4T1 (Cat CRL-2539) and the murine melanoma cell line B16F10 (Cat CRL-6475) were purchased from the American Type Culture Collection (ATCC). HIOEC was a gift from the Shanghai Ninth People's Hospital. 4T1 and B16F10 cells were cultured in RPMI 1640 medium modified (HyClone) containing 10% FBS (Gibco) and 1% penicillin (HyClone). HIOEC cells were cultured in serum-free keratinocyte medium (Gibco). For hypoxia exposure, cells were incubated under hypoxic conditions (37 °C, 1% O$_2$, 5% CO$_2$) using an incubator (Galaxy 170 R, Eppendorf). Clonogenic assays of cells in vitro were performed as previously described[63]. For clonogenic assays, cells were seeded in six-well plates (2,000 cells per well). Then, cells under hypoxic conditions were inoculated with QD-Cat-RGD (50 μg mL$^{-1}$) or QD-RGD (50 μg mL$^{-1}$) for 8 h under hypoxic conditions and subsequently treated with RT (6 Gy). At the end of the experiment, cells were fixed with 4% paraformaldehyde for 20 min and stained with crystal violet for 20 min following by washing with PBS. The CCK8 assay (Dojindo) was performed according to the manufacturer's protocol.

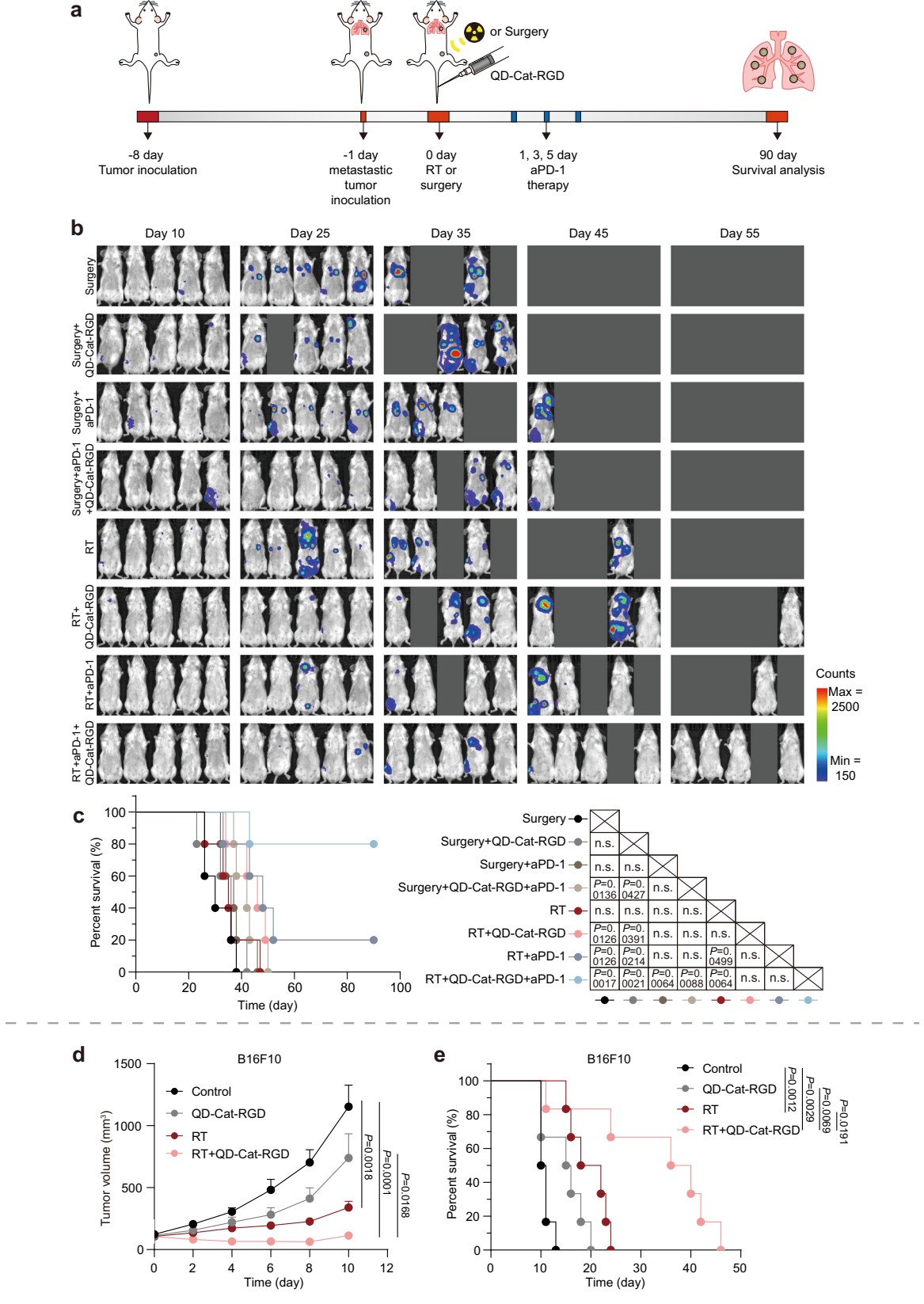

**Cytokine and ATP detection**. For the secretion levels of HMGB1 and ATP from cells in vitro, cells were first inoculated with QD-Cat-RGD (50 µg mL⁻¹) or QD-RGD (50 µg mL⁻¹) for 8 h and then treated with RT (6 Gy). Culture medium samples were isolated from the cells with different treatments under hypoxic conditions (37 °C, 1% $O_2$, 5% $CO_2$) after 24 h. HMGB1 was detected by a mouse HMGB1 ELISA kit (IBL-International, Cat ST51011). ATP was detected by an ATP assay kit (Beyotime, Cat S0026). Tumor tissues were collected from mice and then homogenized in PBS buffer. The protein concentrations were normalized by a BCA protein kit (Beyotime, Cat P0012S). The intratumoral levels of IFN-γ (4A Biotech, Cat CME0003), TNF-α (4A Biotech, Cat CME0004), and IL-10 (4A Biotech, Cat CME0016) were measured with mouse ELISA kits according to the manufacturer's protocol.

**Fig. 7 QD-Cat-RGD-based RT for treating cancer in the 4T1 experimental early metastasis model and B16F10 melanoma model. a** Schematic showing the experiment using QD-Cat-RGD-based RT to treat mice with the 4T1 experimental model of metastasis at an early stage. BALB/c mice ($n = 5$ per group) were implanted subcutaneously with $7 \times 10^5$ 4T1 mammary carcinoma cells on right hind flanks and after 7 days consequently implanted intravenously with $2 \times 10^5$ 4T1 cells to mimic the early stage of metastasis in which cancer cells escaped from the primary tumor site into the circulation and seeded in distant organs especially the lung. When primary tumor volumes were 100 mm³, mice received different treatments. **b** Bioluminescence images tracking the spreading and growth of intravenously injected 4T1-Luc cells in BALB/c mice ($n = 5$ per group) treated with surgery, surgery + QD-Cat-RGD, surgery + aPD-1, surgery + QD-Cat-RGD + aPD-1, RT, RT + QD-Cat-RGD, RT + aPD-1 or RT + QD-Cat-RGD + aPD-1 on days 10, 25, 35, 45, and 55. **c** Survival analysis of 4T1 lung metastatic mice ($n = 5$ per group) after the different treatments. **d** C57BL/6 mice ($n = 6$ per group) were implanted subcutaneously with $3 \times 10^5$ B16F10 murine melanoma cells in the right hind flanks. When tumor volumes were 100 mm³, mice received PBS, QD-Cat-RGD, RT, or RT + QD-Cat-RGD treatment. Tumor growth curves of B16F10 tumors. Photographs of all tumor-bearing mice at day 8 are shown in Supplementary Fig. 17. **e** Survival analysis of mice ($n = 6$ per group) with B16F10 tumors treated with PBS (control), QD-Cat-RGD, RT or RT + QD-Cat-RGD. All data are presented as the mean ± s.e.m. ($n = 5$–6 per group) and $n$ represents the number of independent animals. Statistical significance was calculated via one-way ANOVA with Tukey's multiple comparisons test (**d**) and log-rank (Mantel–Cox) test (**c**, **e**). n.s. not significant. Source data are provided as a Source Data file.

---

**Animal experiments**. To evaluate the therapeutic efficacy, female BALB/c and C57BL/6 mice (6–8 weeks) were purchased from Hubei Provincial Academy of Preventive Medicine. Ethical approval of this study was obtained from the Animal Ethics Committee of the School and Hospital of Stomatology of Wuhan University (approval number: S07920070E). All animal experimental procedures were performed in accordance with the Regulations for the Administration of Affairs Concerning Experimental Animals approved by the State Council of the People's Republic of China. All mice were housed under specific pathogen-free (SPF) conditions (temperature ~22 °C, humidity ~50%) with a 12/12 h dark/light cycle. 4T1 ($7 \times 10^5$) cells or B16F10 cells ($3 \times 10^5$) resuspended in 100 μL of RPMI medium were subcutaneously injected into the right hind flank of each BALB/c mouse or C57BL/6 mouse. After the tumor volume reached approximately 100 mm³, the mice were randomly divided into groups ($n = 5$ or 6 per group). The tumor-bearing mice were intravenously injected with a single dose of QD-Cat-RGD (150 μL, 2 mg mL⁻¹), QD-Cat (150 μL, 2 mg mL⁻¹), QD-RGD (150 μL, 2 mg mL⁻¹), or PBS (150 μL). The location and range of RT with the help of NIR-IIb imaging 2 h after injection (Supplementary Fig. 8a). Then, we used the lead shield to localize the range of RT determined by previous imaging (Supplementary Fig. 8b,c). A small animal irradiator (RS2000pro, Rad Source Technologies) was used in this experiment. Tumor volume was calculated as follows: width² × length × 0.5. All mice in this study were euthanized under CO₂ anesthesia if the volume of primary tumor reached a maximum allowable volume of 1,500 mm³ or if tumor burden compromised the animal welfare. For depletion of CD8⁺ T cells, the aCD8 antibodies (200 μg per mouse, Bioxcell, clone 53-6.7, Cat BE0004-1) were injected intraperitoneally on days 0, 4, 8, and 12.

To establish a distant metastasis model, female BALB/c mice (6–8 weeks) were purchased from Hubei Provincial Academy of Preventive Medicine. For the primary tumor inoculation, 4T1 cells ($7 \times 10^5$) suspended in 100 μL RPMI medium were subcutaneously injected into the right hind flank of each female BALB/c mouse. Two days after the first tumor inoculation, 4T1 cells ($7 \times 10^5$) were subcutaneously injected into the left hind flank of each female BALB/c mouse as a distant tumor mimic. Eight days after the first tumor inoculation, the tumor-bearing mice were treated with PBS (control group), QD-Cat-RGD, aPD-1, QD-Cat-RGD + aPD-1, RT (12 Gy), RT + QD-Cat-RGD, RT + aPD-1, and RT + QD-Cat-RGD + aPD-1. QD-Cat-RGD was injected into mice via the tail vein. The aPD-1 antibodies (200 μg per mouse, Bioxcell, clone RMP1-14, Cat BE0146) were injected intraperitoneally on days 1, 3, and 5.

To establish the experimental model of metastasis at early stage, 7 days after primary tumor injection, luciferase-transgenic 4T1 (4T1-Luc) cells ($2 \times 10^5$) were administered intravenously via tail vein infusion into each BALB/c mouse. Eight days after the first tumor inoculation, 4T1 tumor-bearing mice were treated with surgery, surgery + QD-Cat-RGD, surgery + aPD-1, surgery + QD-Cat-RGD + aPD-1, RT, RT + QD-Cat-RGD, RT + aPD-1 or RT + QD-Cat-RGD + aPD-1. The aPD-1 antibodies were injected intraperitoneally on days 1, 3, and 5. Metastatic cancer was monitored by the IVIS Spectrum Imaging System (PerkinElmer) and analyzed by Living Image software (PerkinElmer). Before the mice were imaged, ᴅ-luciferin (150 mg kg⁻¹) was intraperitoneally injected into the mice.

**Western blot analysis**. Tissue harvested from treated mice was lysed with RIPA buffer (Beyotime). The protein concentrations were normalized by a BCA protein kit (Beyotime, Cat P0012S). Western blot analysis was performed as previously described[64]. Briefly, the proteins were separated by SDS-PAGE using wet electroblotting systems (Bio-Rad) and transferred to PVDF membranes (Roche). Membranes were blocked in nonfat milk (5%) for 1 h and incubated with primary antibodies overnight at 4 °C. Antibodies against the following proteins were used for the western blot assay: GAPDH (1:5,000, Proteintech, Cat HRP-60004) and HIF-1α (1:500, Novus Biologicals, Cat NB-100-105). Signal detection was performed by using an ECL kit (Advansta). The signals were collected and analyzed

using an Odyssey system (Li-Cor Biosciences) and Image Studio software v. 5.2 (Li-Cor Biosciences).

**IHC and histological analysis**. H&E staining and IHC staining were performed as previously described[65]. Briefly, after deparaffinization and rehydration, 4 μm paraffin-embedded tumor tissue sections were counterstained with H&E (ZSGB-BIO) and then dehydrated with graded alcohol and clearing in xylene. For IHC, after deparaffinization and rehydration, 4 μm paraffin-embedded tumor tissue sections were retrieved by citrate acid solution (pH 6.0) for 5 min at high pressure. The slides were subsequently incubated in 3% hydrogen peroxide and in goat serum. Primary antibodies against CD3ε (1:200, Cell Signaling Technology, Cat 99940), CD8α (1:400, Cell Signaling Technology, Cat 98941), Foxp3 (1:400, Cell Signaling Technology, Cat 12653), granzyme B (1:200, Cell Signaling Technology, Cat 44153), and HIF-1α (1:150, Novus, Cat NB100-105) were used. HRP-conjugated secondary antibodies (undiluted, anti-rabbit/anti-mouse, Cell Signaling Technology, Cat 8114/8125) were then used. The signals were detected by DAB substrate kit (Cell Signaling Technology). Slides were scanned using an Aperio ScanScope CS scanner (Leica). CaseViewer v. 2.4 (3DHISTECH) was used for creating slide presentations. Aperio ImageScope software v. 12.3.2 (Leica) was used for histoscore analysis of these detection indicators. The histoscore was normalized to 0–300.

**Immunofluorescence**. Immunofluorescence staining was performed as previously described[64]. Primary antibodies against anti-phospho-histone H2AX (Ser139/Tyr142) (1:100, Cell Signaling Technology, Cat 5438), calreticulin (1:400, Cell Signaling Technology, Cat 12238), and HMGB1 (1:400, Abcam, Cat ab18256) were used. Fluorescence-conjugated secondary antibodies (1:500, anti-rabbit, Dylight 488/594, Abbkine, Cat 23220/23430) and anti-fade fluorescence mounting medium with DAPI (ZSGB-BIO) were then used. The slides were examined with an FV1000 confocal microscope (Olympus) using FV10-ASW software v. 4.0 (Olympus). The fluorescence signals were analyzed using Image-Pro Plus software v. 6.0 (Media Cybernetics).

**Flow cytometric analysis and FACS**. For detection of calreticulin expression on the cell surface, an anti-calreticulin antibody (1:100, Cell Signaling Technology, Alexa Fluor 488, Cat 62304) was used to stain the cells after different treatments. For control, rabbit IgG isotype control (1:100, Cell Signaling Technology, Alexa Fluor 488, Cat 2975) was used. For the annexin V/PI assay, cells were digested by EDTA-free trypsin and washed with PBS buffer followed by detection with Annexin V apoptosis detection kit (eBioscience). The stained cells were run on a flow cytometer (Beckman) using the CytExpert software v. 2.3 (Beckman) and then analyzed by FlowJo software (v. 10, TreeStar).

For lymphocyte analysis in vivo, tumors and TDLNs were collected from mice and made into cell suspensions by gentleMACS™ dissociator and digestive enzyme (Miltenyi Biotec) according to the manufacturer's instructions. Live or dead cells were separated by Fixable Viability Dye (eBioscience, eFluor 780 Cat 65-0865-14). Cells were stained with fluorescence-labeled anti-CD45 (1:500, eBioscience, perCP-Cyanine5.5, clone 30-F11, Cat 45-0451-82), anti-CD3e (1:500, eBioscience, FITC, clone 145-2C11, Cat 11-0031-82), anti-CD4 (1:500, eBioscience, eFluor 450, clone RM4-5, Cat 48-0042-82), anti-CD8a (1:500, Biolegend, APC, clone 53-6.7, Cat 100712), anti-CD8a (1:500, Biolegend, Alexa Fluor 594, clone 53-6.7, Cat 100758), anti-CD11b (1:500, Biolegend, FITC, clone M1/70, Cat 101206), anti-F4/80 (1:300, eBioscience, PE-Cyanine7, clone BM8, Cat 25-4801-82), anti-CD80 (1:300, eBioscience, APC, clone 16-10A1, Cat 17-0801-82), anti-CD206 (1:200, Biolegend, PE, clone C068C2, Cat 141705), anti-CD11c (1:500, eBioscience, FITC, clone N418, Cat 11-0114-82), anti-CD80 (1:300, eBioscience, PE, clone 16-10A1, Cat 12-0801-82), anti-CD86 (1:300, eBioscience, APC, clone GL1, Cat 17-0862-82), anti-MHC-II (1:300, eBioscience, PE-Cyanine7, clone M5/114.15.2, Cat 25-5321-82), anti-CD25 (1:200, eBioscience, APC, clone PC61.5, Cat 17-0251-82), anti-Foxp3 (1:100,

eBioscience, PE, clone FJK-16s, Cat 12-5773-82), anti-CD44 (1:500, eBioscience, PE, clone IM7, Cat 12-0441-82), and anti-CD62L (1:200, eBioscience, APC, clone MEL-14, Cat 17-0621-82) antibodies following the manufacturers' instructions. The stained cells run on a flow cytometer (Beckman) using the CytExpert software v. 2.3 (Beckman). The gating strategy used for the study is shown in Supplementary Fig. 9. The positive gate of CD206$^{high}$ or CD80$^{high}$ TAMs were determined by fluorescence-minus-one (FMO) control[66]. The data acquired flow cytometer was analyzed by the FlowJo software (v. 10, TreeStar).

To isolate the live CD45$^-$ cells from tumor tissue, single cell suspension of tumor tissue was stained with the Fixable Viability Dye (eBioscience, eFluor 780, Cat 65-0865-14) and anti-CD45 (1:500, eBioscience, perCP-Cyanine5.5, clone 30-F11, Cat 45-0451-82) antibody. Live CD45$^-$ cells (the sorting gating strategy is shown in Supplementary Fig. 3a) were sorted on a MoFlo XDP cell sorter (Beckman) in PBS with 2% FBS.

**Statistical analysis**. Data are represented as the mean ± standard error of the mean (s.e.m.) or standard deviation (s.d.) as indicated in the figure legends. One-way ANOVA with Tukey's multiple comparisons was used for multiple comparisons when more than two groups were compared, and two-tailed Student's t-test was used for two-group comparisons. Survival benefit was determined using a log-rank (Mantel–Cox) test. All statistical tests were performed using GraphPad Prism software v. 8.0 (GraphPad Software) and Excel 2016 software (Microsoft). In all types of statistical analysis values of $P < 0.05$ were considered significant.

**Reporting summary**. Further information on research design is available in the Nature Research Reporting Summary linked to this article.

## Data availability
The data generated in this study are available within the Article, Supplementary Information or Source Data file. Source data are provided with this paper.

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

## Acknowledgements

This work is supported by the National Key R&D Program of China (2020YFA0908800, 2017YFSF090107) and the National Natural Science Foundation of China grant to Z.-J.S. (82072996 and 81874131), M.Z. (21974104), and R.C. (22174105). We thank Shuyan Liang and Zhixin Qiu from Wuhan Biobank Co., Ltd (Wuhan, China) for their excellent technical assistance on flow cytometry and FACS.

## Author contributions

Z.-J.S., M.Z., and R.C. conceived and designed the experiments. H.L., M.W., B.H., S.-W.Z., J.-J.Z., and D.-R.C. performed the experiments. H.L., M.W., and B.H. analyzed the data. H.L., M.W., Z.-J.S., M.Z., and R.C. wrote the manuscript. All authors have given approval to the final version of the manuscript.

## Competing interests

The authors declare no competing interests.
