## [Peer Review File · Nature Communications]

Reviewers' Comments:

Reviewer #1:

Remarks to the Author:

Sun and coworkers constructed a stabilized theranostic nanoprobe based on quantum dots emitting in the near-infrared IIb (NIR-IIb, 1,500-1,700 nm) window modified by catalase, arginine-glycine-aspartate (RGD) peptides and poly(ethylene glycol) (PEG).

It is interesting that nanoprobe significantly relieved intratumoral hypoxia and reduced the tumor infiltration of immunosuppressive cells, and combined with immunotherapy, the animal study results are substantial and impressive. However, several important issues should be thoroughly addressed.

1. The authors should check the scale bar again in "Fig S2A" is not "um" but "nm". If the scale bar is correct, the authors should give the discussion.
2. What is the loading capacity of RGD and catalase on the quantum dots.
3. The resolution of picture in Fig. 1c is not high enough. It needs to be replaced.
4. The concentration of QD-Cat-RGD in tumor site needs to be determined again by ICP. And this experiment is lack of the control group that QD without RGD.
5. The flow chart of Fig.3d should be represented by point chart instead of terrain chart.
6. Please change the column chart in Fig 3b, h and i to point chart
7. Please give the statistic chart of flow cytometry of Fig. 3g.
8. Although QD-Cat-RGD could significantly reinforce the ICD of cancer cells induced by RT in vitro, the authors should also analyse the surface calreticulin expression, the ATP and HMGB1 secretion in 4T1 tumor after RT in vivo.
9. The position of the cross quadrant in figure Fig 4f and Fig5e is not correct, the dividing point should be in the middle of the two cell groups, and the existing position should be shifted to the upper right. Please give the correct statistical chart after re-analysis
10. The authors claim of inducing robust immune responses through the treatment would be significantly enhanced by more direct evidence of anti-tumor T cell induction, for example, demonstration that depletion of T-cells hurts the response to the therapy.

Reviewer #2:

Remarks to the Author:

In this manuscript, the authors examine the combination of radiotherapy (RT) and administration of a nanoprobe composed of fluorescent PbS/CdS quantum dots conjugated to catalase and RGD, an avb3 integrin-targeting peptide (QD-Cat-RGD). They show that they are able to visualize the tumor using infrared (1600 nm) fluorescence imaging and examine the effect of separate or combined application of RT, QD-Cat-RGD and anti-PD1. The combination of RT and hypoxia-relieving treatment is not novel but certainly very promising, as is the use of NIR-II emitting nanoprobe to image tumors in preclinical models. The novel ideas here lie (i) in using targeted NIR-II QDs to guide the RT; (ii) in using heavy atoms in PbS/CdS as radiosensitizers to enhance RT effects; (iii) in using the QD nanoprobe as carriers to transport catalase into the tumors. Their results are very promising and interesting and will certainly be of high interest to the community. However, some of these novel ideas are not completely validated by the experiments performed in this study.

For example, the ability of the designed QDs to target tumors is not thoroughly investigated. In particular the effect of RGD is not examined in vivo. 4T1 tumors typically exhibit a certain level of EPR-mediated uptake. Since the rationale behind the design of the probe is image-guided RT, the authors should further investigate the targeting properties of the QD-Cat-RGD. They could eg. compare the tumor uptake with and without RGD, or in avb3-negative tumor.

As for using high-Z elements to enhance the effect of RT, the authors show indeed that RT+QD-Cat-RGD is more efficient in many aspects than RT alone, but there is always a significant effect of QD-Cat-RGD alone, without RT (see Figure 4 and 6 for example), compared to the control. The authors should address this point in the discussion: is the in vivo radio-sensitization effect due to the catalase, to the presence of high-Z elements, or to an indirect radio-sensitization effect? Is the combined effect a cumulative effect or a synergy? For example, what would happen if the RT was applied before the QD-Cat-RGD administration?

The origin of the radio-sensitizing effect of high Z nanoparticles is debated: given the small concentration of these elements in the tumor, the increase in physical dose deposition is not significant (see for example Rosa et al Cancer Nano 8, 2 (2017)). To be able to state that "QD-Cat-RGD nanoprobe with a high-Z element was able to enhance the radiotherapeutic efficacy", the authors should try to separate the different components of their system, for example to test the effect of RT + QD-RGD (no catalase), or at least discuss these points in a more balanced fashion in the discussion.

Finally, a missing important element is the dose of nanoprobe delivered to the tumor: it is indeed quite important to better characterize the targeting ability as well as to understand the mechanism behind the radiosensitizing effect. It would be relatively easy to perform elemental analysis on the tumor tissue to determine the average concentration of QDs, and thus, of catalase present in the tumor micro-environment, the spleen, liver and lung in these experiments. In addition, it would enable a comparison between the in vivo distribution and the range of concentration used in the cytotoxicity study (Figure S9).

Minor points:

The authors should provide absorption and fluorescence spectra of QD used, as well as an estimation of their fluorescence quantum yield.

What is the laser intensity (mW/cm²) used for the photo-stability measurement? Is this the same as the one used for imaging?

The authors state that "With QDs modified by PEG, QD-Cat-RGD nanoprobe could circulate in the blood for a long time, and only approximately 16.7% of the QD-Cat-RGD nanoprobe had been taken

up by the hemocytes 30 min postinjection (p.i.) (Fig. 2c).", but the blood residence/circulation time and uptake by hematocytes are unrelated, as nanoparticles are usually eliminated from the circulation by uptake by macrophages in the liver and spleen or by accumulation in the lung (as seems to be the case here, and which is usually seen as a consequence of the instability and aggregation of NPs in the bloodstream).

From the experimental section it is not clear how catalase is coupled to the QDs. Is it only by adsorption? no reagent seem to be used to covalently couple the catalase on the QDs. What is the approximate number of active catalase per QD?

Reviewer #3:

Remarks to the Author:

Summary

The manuscript describes the combination of a nanoprobe with radiation therapy to improve local and distant tumor control. The nanoprobe is designed to target the tumor via the well-studied RGD motifs, and are intended to relieve hypoxia, permitting increased radiation induced DNA damage. The authors demonstrate that the nanoprobe improves cancer killing in vitro, and provide some evidence of altered hypoxia responses in vivo. The combination of RT and nanoprobe improves tumor control, and improves control of delayed administration distant tumors or metastases. Some additional effects of anti-PD1 are demonstrated.

The main problem with the manuscript is that the core contention is not well demonstrated. It is unclear whether these probes delivered IV and partially enriching at the tumor can improve hypoxia in the tumor environment purely through this level of delivery of catalase. The experiments as presented do not prove this mechanism. There are issues in the experimental designs and results that bring the proposed mechanism into question. This, combined with the relative lack of novelty in the manuscript limit its value to the field.

Major issues

The authors state that they use the signal from the nanoparticle to direct their radiation therapy, but their radiation equipment is a RS2000Pro cabinet irradiator. This has no targeting capacity, instead is a cone-beam x-ray source. These facts don't match. The investigators must alter their language, and explain how tumor-selective treatment was given, with dosimetry validation and what non-tumor structures were included in the field. If imaging was used to target delivery, the

method to integrate imaging with treatment should be described.

The approach used to demonstrate in vitro killing by RT plus nanoprobe is a problem (Figure 3). The cultures were not under hypoxic conditions – in fact one major criticism of standard in vitro culture of cancer cells is that it is fully normoxic. If the nanoprobe can help kill cancer cells in this setting, then it is unlikely that overcoming hypoxia is part of the mechanism.

The imaging location of the nanoprobe appears to show poor localization to the tumor core, where hypoxia is most relevant. The probes appear to accumulate at the tumor margins where the more normal vasculature is most able to deliver these probes. In addition, trans-vascular passage of these probes into the tumor stroma and to the cancer cells has not been demonstrated.

If the nanoprobe function via improved oxygenation of the tumor, then there may need to be an explanation for their ability to control tumor growth even in the absence of radiation therapy. Why do tumors grow slower when oxygenation is improved? Notably, in the dual tumor model, the probes can almost cure distant tumors as a single agent, when administered 1d following that second tumor implantation. This second tumor would both have a small number of cancer cells in the implantation site to bind the probes, and low hypoxia. When hypoxia develops through tumor growth, the nanoprobe would remain the same in number with increased numbers of cancer cells, yet they have an effect. These data suggest hypoxia regulation is not the mechanism.

The immune mechanism of primary tumor treatment is not proven. While there are a range of immune correlates in the tumor following radiation therapy with the nanoprobe present, there is no proof that the mechanism is immune. As the investigators show, the combination is more effective at killing cancer cells in vitro. Therefore, the efficacy to the local tumor could be entirely independent of immune mechanisms. Additional studies are necessary to demonstrate that the local tumor control mechanism is immune.

The flow cytometry of macrophages in Figure 4 is unconvincing. There are not distinct populations, rather a slight alteration in MFI of CD80 and CD206.

In figure 5, cells in the TDLN are analyzed. There is a little confusion since Figure 5d appears to be IHC of the tumor, but based on the text in the manuscript, Figure 5c and 5e are analyses of T cells in the TDLN. These data are unexpected. Firstly, in an untreated TDLN only 5% of the CD45+ cells are shown to be CD3+ cells. This is not plausible. Proportions more like 40-60% of CD45+ cells are more normal. Secondly, all of these cells are CD44+. This means that there are no naïve cells in these lymph nodes. Again, this is not plausible.

The dual flank tumor model is quite artificial. Delivery of a second tumor 7 days after the first results in concomitant immunity effects in the second tumor, and makes it highly responsive to therapy. This second tumor is injected only 1 day prior to therapy.

The response to treatment of the second tumor model does not correlate well with CD8 T cell infiltrate, though there is a better correlation with CD3 T cell infiltrate. These data suggest a non-CD8 T cell mechanism, which is not investigated.

The survival data following spontaneous metastases should offset the above concern, but strangely the authors administer metastatic tumors via IV injection 7 days after injecting the primary tumor. 4T1 is spontaneously metastatic, and should require no IV treatments. However, since the primary tumor treatment occurs at d8 after primary tumor inoculation, this may be before LN and metastatic spread has occurred. To genuinely model treatment of metastatic disease, therapies should be given to mice with established tumors and established metastases.

The discussion states that RT and nanotechnology approaches have mainly ignored the effect on the immune system. This is unfair to the field. There are large numbers of papers with this as a focus, and the topic is widely reviewed.

Minor issues

In figure 3, key facts are missing.

The dose of RT for 1a-j

The timing of gH2ax assessment for 1c

RT is poorly effective at inducing cell death in 4T1 in 1d. The dose used is very relevant.

The immunofluorescence data throughout is of limited value as presented. The figures are far too small to interpret, and are mostly black squares. These are all accompanied by quantitative analysis, so this is a minor issue. However, some other way to present these data is encouraged.

Point-to-point responses to the comments

Reviewer #1

Sun and coworkers constructed a stabilized theranostic nanoprobe based on quantum dots emitting in the near-infrared IIb (NIR-IIb, 1,500-1,700 nm) window modified by catalase, arginine-glycine-aspartate (RGD) peptides and poly(ethylene glycol) (PEG). It is interesting that nanoprobe significantly relieved intratumoral hypoxia and reduced the tumor infiltration of immunosuppressive cells, and combined with immunotherapy, the animal study results are substantial and impressive. However, several important issues should be thoroughly addressed.

Response:

We would like to thank the reviewer for the positive and constructive comments. By responding to the reviewer's comments in detail and revising the manuscript accordingly, we believe our manuscript has been significantly strengthened. We have added sufficient experimental/methodological details in the Methods section. All revisions are highlighted in red color in the revised manuscript and Supplementary Information.

Comment 1:

1. The authors should check the scale bar again in "Fig S2A" is not "um" but "nm". If the scale bar is correct, the authors should give the discussion.

Response 1:

Thank you for pointing out this mistake. We have corrected the scale bar in Fig. S2A to use "nm", which is shown in Fig. R1.

Revision made:

In the revised manuscript, the scale bars of "Supplementary Fig.2a" was corrected to "nm" as shown in Fig. R1.

Fig. R1 TEM image of QD-Cat-RGD nanoprobe extracted from the feces.

Comment 2:

2. What is the loading capacity of RGD and catalase on the quantum dots.

Response 2:

We thank the reviewer for the kind suggestion. According to reviewer's suggestion, the loading capacity of RGD and catalases on the QD-Cat-RGD nanoprobe have been added in the revised manuscript.

To reveal the loading capacity of RGD and catalase (Cat) on the QD-Cat-RGD

nanoprobes, we used a standard BCA protein assay to quantify the unbound Cat after the conjugation and purification of nanoprobes and consequently obtained the amounts of Cat grafted on QDs. The molar ratio of grafted Cat onto QDs was approximately 2.5:1. We further used UV spectrometry to detect the absorbance at 220 nm to quantify the unbound RGD (R. F. Boyer, *Modern Experimental Biochemistry*, 2000, Prentice Hall Press), and further obtained the amounts of RGD grafted on QDs. We calculated that the molar ratio of grafted RGD to QDs was about 30:1.

Revision made:

In the revised manuscript, the following statements in Result section were added:

“To reveal the loading capacity of RGD and Cat on the QD-Cat-RGD nanoprobes, we used a standard BCA protein assay to quantify the unbound Cat after the conjugation and purification of nanoprobes and consequently obtained the amounts of Cat grafted on QDs. The molar ratio of grafted Cat onto QDs was approximately 2.5:1. We further used UV spectrometry to detect the absorbance at 220 nm to quantify the unbound RGD and obtain the amounts of RGD grafted on QDs³³. We calculated that the molar ratio of grafted RGD to QDs was approximately 30:1”.

Comment 3:

3. The resolution of picture in Fig. 1c is not high enough. It needs to be replaced.

Response 3:

Thank you for pointing this out issue. We have replaced the image in “Fig. 1c” with a high-resolution image, as shown in Fig. R2.

Revision made:

In the revised manuscript, the image in “Fig. 1c” has been replaced with a high-resolution image, as shown in Fig. R2.

Fig. R2 Bright-field and corresponding NIR-IIb fluorescent images of 4T1 cells after incubation with PBS (control), QD-Cat (without RGD modification) or QD-Cat-RGD (with

RGD modification). Scale bars = 50 μm .

Comment 4:

4. The concentration of QD-Cat-RGD in tumor site needs to be determined again by ICP. And this experiment is lack of the control group that QD without RGD.

Response 4:

We would like to thank the reviewer for the constructive comments. According to the reviewer's suggestion, we have added the QD-Cat (without RGD modification) group as a control group in NIR-IIb fluorescence imaging and selected Pb element to determine the content of QD-Cat-RGD in tumor and main organs of body by ICP-OES compared with QD-Cat. These new results have added to the revised manuscript.

To compare the image-guided properties of QD-Cat and QD-Cat-RGD, 4T1 tumor-bearing BALB/c mice were intravenously injected with QD-Cat-RGD (150 μL , 2 mg mL^{-1}) or QD-Cat (without RGD) (150 μL , 2 mg mL^{-1}) nanoprobes. The strong signals focused at the tumor site confirmed the gradual infiltration of QD-Cat-RGD nanoprobes with an excellent tumor-targeting ability (Fig. R3a). However, the QD-Cat nanoprobes without RGD modification did not show this ability, with weak signals found in tumor tissue (Fig. R3b). The tumor-to-normal tissue (T/NT) signal ratio of QD-Cat-RGD increased sharply and peaked at approximately 2 h postinjection (p.i.) with a T/NT ratio of approximately 2.742, which was higher than the T/NT ratio of QD-Cat (Fig. R3d).

To explore the precise position of QD-Cat-RGD infiltrated in tumor tissue, serial slices of tumor tissue were imaged in the NIR-IIb window and stained with hematoxylin and eosin (H&E). The merged images showed that the QD-Cat-RGD nanoprobes with RGD modification could deeply infiltrate the tumor tissue core instead of marginal stromal tissue (Fig. R3c). Without RGD modification, QD-Cat nanoprobes were mainly found in the stroma located in the margin of tumor tissue (Fig. R3c). These results suggested that QD-Cat-RGD nanoprobes with the targeting property of RGD could migrate through the vasculature and deeply infiltrate the tumor tissue, reaching the core of tumor which was mainly under hypoxia.

Next, we selected Pb element measured by ICP-OES to investigate the pharmacokinetics of nanoprobes and biodistributions in the main organs of the body, including the liver, spleen, lungs and kidneys, as well as tumors after blood circulation. The blood circulation half-time ($T_{1/2}$) of QD-Cat-RGD was calculated to be approximately 3.37 h (Fig. R3e) according to a two-compartment model. The systemic distribution of Pb revealed a 17.67% ID g^{-1} tumor accumulation efficiency of QD-Cat-RGD (Fig. R3f), which was higher than the 6.84% ID g^{-1} tumor accumulation efficiency of QD-Cat (Fig. R3f).

These above results demonstrated a good targeting property of QD-Cat-RGD by comparing with the QD-Cat (without RGD) group.

Revision made:

We have added the data for Figure R3b, c, d, e and f as "Figure 2b, c, d, e and f" in the revised manuscript.

Fig. R3 *In vivo* fluorescence imaging in the NIR-IIb window and biodistribution of QD-Cat-RGD. **a,b** Whole body and high-magnification fluorescent images of 4T1 tumor-bearing mice intravenously injected with QD-Cat-RGD (**a**) or QD-Cat (**b**) nanoprobes at different time points (5 min, 1 h, 2 h, 4 h, 8 h, 12 h, 24 h) in the NIR-IIb window. Scale bars = 5 mm. The excitation power density was 25 mW cm^{-2} provided by an 808 nm laser. **c** Serial sections of tumor tissue imaged in the NIR-IIb window or stained with H&E. The excitation power density for NIR-IIb fluorescence imaging was 25 mW cm^{-2} provided by an 808 nm laser. Scale bars = 1 mm. **d** The T (tumor)/NT (nontumor) ratio of QD-Cat-RGD or QD-Cat at different time points (5 min, 1 h, 2 h, 4 h, 8 h, 12 h, 24 h). **e** The blood concentration of Pb in BALB/c mice at different time points (0.5 h, 1 h, 3 h, 8 h, 24 h, 48 h,

72 h) after intravenous injection with QD-Cat-RGD nanoprobe. **f** Delivery efficiency for injected dose (ID) of QD-Cat-RGD or QD-Cat in tumors and main organs (livers, spleens, lungs, and kidneys) of mice injected with QD-Cat-RGD or QD-Cat at 24 h p.i. **g** The protein expression level of HIF-1 α in 4T1 tumor tissue in mice at 24 h after intravenous injection with PBS (control), QD-RGD (150 μ L, 2 mg mL⁻¹) and QD-Cat-RGD (150 μ L, 2 mg mL⁻¹) measured by western blot analysis. All data are shown as the mean \pm s.e.m. of three replicates. Statistical significance was calculated via Student's t test (**f**) and one-way ANOVA with Tukey's multiple comparisons test (**g**). * $P < 0.05$; *** $P < 0.001$.

In the revised manuscript, the following statements were added in the Results section:

“To evaluate the image-guided properties of QD-Cat-RGD nanoprobe in vivo, 4T1 tumor-bearing BALB/c mice were intravenously injected with QD-Cat-RGD (150 μ L, 2 mg mL⁻¹) or QD-Cat (150 μ L, 2 mg mL⁻¹) nanoprobe.”

“The strong signals focused at the tumor site confirmed the gradual infiltration of QD-Cat-RGD nanoprobe with an excellent tumor-targeting ability (Fig. 2a). However, the QD-Cat nanoprobe without RGD modification did not show this ability, with weak signals found in tumor tissue (Fig. 2b). The tumor-to-normal tissue (T/NT) signal ratio of QD-Cat-RGD increased sharply and peaked at approximately 2 h postinjection (p.i.) with a T/NT ratio of approximately 2.742, which was higher than the T/NT ratio of QD-Cat (Fig. 2d).”

“To explore the precise position of QD-Cat-RGD infiltrated in tumor tissue, serial slices of tumor tissue were imaged in the NIR-IIb window and stained with hematoxylin and eosin (H&E). The merged images showed that the QD-Cat-RGD nanoprobe with RGD modification could deeply infiltrate in the tumor tissue core instead of marginal stromal tissue (Fig. 2c). Without RGD modification, QD-Cat nanoprobe were mainly found in the stroma located in the margin of tumor tissue (Fig. 2c). These results suggested that QD-Cat-RGD nanoprobe with the targeting property of RGD could migrate through the vasculature and deeply infiltrate the tumor tissue, reaching the core of tumor which was mainly under hypoxia.”

“Here, we selected Pb element measured by ICP-OES to investigate the pharmacokinetics of nanoprobe and biodistributions in the main organs of the body, including the liver, spleen, lungs and kidneys, as well as tumors after blood circulation. The blood circulation half-time ($T_{1/2}$) of QD-Cat-RGD was calculated to be approximately 3.37 h (Fig. 2e) according to a two-compartment model³⁸. The systemic distribution of Pb revealed a 17.67% ID g⁻¹ tumor accumulation efficiency of QD-Cat-RGD (Fig. 2f), which was higher than the 6.84% ID g⁻¹ tumor accumulation efficiency of QD-Cat (Fig. 2f). The above results demonstrated a good property of QD-Cat-RGD by comparing with the QD-Cat (without RGD) group”.

We have also added the detailed methods about pharmacokinetics and biodistribution of QDs in the Methods:

“In vivo pharmacokinetics and biodistribution of QDs. The 4T1 tumor-bearing mice intravenously injected with QD-Cat-RGD (150 μ L, 2 mg mL⁻¹) or QD-Cat (without RGD)

(150 μL , 2 mg mL^{-1}) nanoprobe were used for pharmacokinetics and biodistribution studies. At 0.5, 1, 3, 8, 24, 48 and 72 h p.i., 50 μL blood from treated mouse was collected. By using inductively coupled plasma optical emission spectrometry (ICP-OES) to calculate the concentration of Pb in blood to estimate the percentage of the QD-Cat-RGD in blood. The tissues including spleen, liver, lung, kidney, and tumor were collected, weighed and then dissolved completely in 5 mL digest solution ($\text{HNO}_3 : \text{H}_2\text{O}_2 = 4:1$) for 2 h until the solution turned to clear. Subsequently, the solutions were diluted by ultrapure water to 10 mL and analyzed by ICP-OES to measure the concentration of Pb in the resulting solutions. The calculation formula of % ID g^{-1} is: $(\text{The amount of Pb element in organs}) \times (\text{The amount of Pb element in ID} \times \text{the weight of organs})^{-1} \times 100\%$. The pharmacokinetic parameters were calculated by the software PKSolver (v. 2.0)⁶⁴”.

Comment 5:

5. The flow chart of Fig.3d should be represented by point chart instead of terrain chart.

Response 5:

Thanks for the advice. We have changed the flow chart of Fig. 3d to point chart according to reviewer’s suggestion which was shown in Fig. R4.

Revision made:

In the revised manuscript, the flow chart in “Fig. 3d” were replaced to the chart as shown in Fig. R4.

Fig.R4 Apoptosis analysis measured by flow cytometry of 4T1 cells at 24 h after different treatments.

Comment 6:

6.Please change the column chart in Fig 3b, h and i to point chart.

Response 6:

We appreciate the reviewer’s suggestion. We have replaced the column charts in “Fig. 3b, h and i” with point charts, which are shown in Fig. R5a-c.

Revision made: In the revised manuscript, the column chart in “Fig. 3b, h and i” was replaced with the point chart shown in Fig. R5a, b and c.

Fig. R5 **a** Histogram plot of the survival fraction of 4T1 cells with different treatments. **b** Histogram plot of secreted ATP production by 4T1 cells with different treatments after 24 h. **c** Histogram plot of HMGB1 released by 4T1 cells with different treatments after 24 h.

Comment 7:

7. Please give the statistic chart of flow cytometry of Fig. 3g.

Response 7:

We thank the reviewer for the suggestion. The statistical chart for the flow cytometry shown in “Fig. 3g” has been added to the revised “Supplementary Fig. 3”, as shown in Fig. R6. The results showed that the surface expression of calreticulin in 4T1 cells treated with QD-RGD- or QD-Cat-RGD-based RT was significantly increased compared with that in 4T1 cells treated with RT only (Fig. R6).

Revision made:

In the revised manuscript, we have added the data of Fig. R6 as “Supplementary Fig.3” in the Supplementary Information.

Fig. R6 Flow cytometry analysis of surface calreticulin expression in 4T1 cells with different treatments after 24 h. MFI, median fluorescence intensity.

Comment 8:

8. Although QD-Cat-RGD could significantly reinforce the ICD of cancer cells induced by RT *in vitro*, the authors should also analyze the surface calreticulin expression, the ATP and HMGB1 secretion in 4T1 tumor after RT *in vivo*.

Response 8:

Thanks for raising this concern. According to the reviewer's suggestion, we tested the QD-Cat-RGD-based RT to promote ICD of cancer cells *in vivo*. The results have been added as "Supplementary Fig. 4" in the revised Supplementary Information.

To evaluate the effect of QD-Cat-RGD-based RT on promoting ICD *in vivo*, the 4T1 tumor-bearing BALB/c mice were intravenously injected with a single dose of QD-Cat-RGD (150 μ L, 2 mg mL⁻¹) or QD-RGD (150 μ L, 2 mg mL⁻¹). Then, the mice were treated with localized RT (12 Gy) 2 h p.i. The tumor tissues were harvested at 24 h after the injection of QD-Cat-RGD or QD-RGD. The results showed that tumor tissue significantly increased the surface expression of calreticulin and ATP and HMGB1 concentrations (Fig. R7 a-c) after mice treated with QD-RGD- or QD-Cat-RGD-based RT, which were consistent with the *in vitro* results.

Revision made:

In the revised manuscript, we have added the data of Fig. R7 as "Supplementary Fig.4" in the Supplementary Information.

Fig. R7 a Flow cytometry analysis of surface calreticulin expression in tumor tissue cells with different treatments after 24 h. **b** Histogram plot of ATP concentration in tumor tissue with

different treatments after 24 h. **c** Histogram plot of HMGB1 concentration in tumor tissue with different treatments after 24 h. All data are shown as the mean \pm s.e.m. of three replicates. Statistical significance was calculated *via* one-way ANOVA with Tukey's multiple comparisons test (h, i). * $P < 0.05$; ** $P < 0.01$; *** $P < 0.001$; n.s., not significant.

In the revised manuscript, the following statements were added into the Results section:

“Moreover, we further tested the effect of QD-Cat-RGD-based RT on promoting ICD in vivo. The 4T1 tumor-bearing BALB/c mice were intravenously injected with a single dose of QD-Cat-RGD (150 μ L, 2 mg mL⁻¹) or QD-RGD (150 μ L, 2 mg mL⁻¹). Then, the mice were treated with localized RT (12 Gy) 2 h p.i. The tumor tissues were harvested at 24 h after the injection of QD-Cat-RGD or QD-RGD. The results showed that tumor tissue significantly increased the surface expression of calreticulin and ATP and HMGB1 concentrations (Supplementary Fig. 4a-c) at 24 h after the mice treated with QD-RGD- or QD-Cat-RGD-based RT, which was consistent with the in vitro results”.

Comment 9:

9. The position of the cross quadrant in figure Fig 4f and Fig5e is not correct, the dividing point should be in the middle of the two cell groups, and the existing position should be shifted to the upper right. Please give the correct statistic chart after re-analysis.

Response 9:

Thank you for pointing out this issue. We apologize for the incorrect gating strategies for “Fig. 4f and 5e”. We have changed the gating strategies for “Fig. 4f and 5e” according to several references (Nat Immunol. 2016, 17, 1322-1333; Nat Commun. 2017, 8, 15338; Nat Commun. 2021, 12, 951)¹⁻³ and reanalyzed the results shown in Fig. R8.

The reanalyzed results did not change the conclusion from previous results that “Compared with the control group, Tregs (CD4⁺CD25⁺Foxp3⁺) in the RT+QD-Cat-RGD group were more significantly reduced than those in the RT group (Fig. 4f and Supplementary Fig. 8c)” and “the central memory CD8⁺ T cell populations (CD8⁺ Tcm, CD44⁺CD62L⁺CD8⁺) in the TDLN of the mice in the RT+QD-Cat-RGD group were significantly increased on day 14 after treatment compared with the control group (Fig.5e)”.

Revision made:

In the revised manuscript, the results in “Fig. 4f and 5e” were replaced to the reanalysis results as shown in Fig. R8a and b.

Fig. R8 a Representative flow cytometric analysis images and relative quantification of Tregs ($CD25^+$ and $Foxp3^+$) gating on $CD4^+CD3^+CD45^+$ cells in tumors. **b** Representative flow cytometric analysis images and relative quantification of $CD44^+CD62L^+$ Tem cells gating on $CD8^+CD3^+CD45^+$ cells in TDLNs. All data are presented as the mean \pm s.e.m. ($n = 5$ per group). Statistical significance was calculated via one-way ANOVA with Tukey's multiple comparisons test. * $P < 0.05$; ** $P < 0.01$.

Comment 10:

10. The authors claim of inducing robust immune responses through the treatment would be significantly enhanced by more direct evidence of anti-tumor T cell induction, for example, demonstration that depletion of T-cells hurts the response to the therapy.

Response 10:

We thank this reviewer for the valuable suggestion. As suggested, we further used antibodies against CD8 (aCD8) to deplete cytotoxic T cells in mice to reveal the effect of cytotoxic T cells in QD-Cat-RGD-based RT. The results have added to the revised Supplementary Information as "Supplementary Fig. 9".

To validate the contribution to anti-tumor effect from T cells especially cytotoxic T cells, $CD8^+$ T cells were depleted using antibodies against CD8 (aCD8) on day 0, 4, 8 and 12 in 4T1 tumor-bearing mice treated with QD-Cat-RGD-based RT (Fig. R9a). The results demonstrated that the antitumor effects of the combination treatment were abrogated with the depletion of $CD8^+$ T cells, resulting in increase of the volume and weight of tumors (Fig. R9b-d). Treatment combining QD-Cat-RGD-based RT with aCD8 also caused decreases in tumor-infiltrated T cells ($CD3^+$) and depletion of cytotoxic $CD8^+T$ cells ($CD8^+CD3^+$) (Fig. R9e, f), confirming that $CD8^+$ cytotoxic T cells were involved in the antitumor effect in primary tumor.

Revision made:

We also have added the data of Fig. R9 as "Supplementary Fig. 9" in the Supplementary Information.

Fig. R9 QD-Cat-RGD-based RT reducing the effect inhibiting tumor growth when depleting the CD8⁺ cytotoxic T cells. **a** Schematic showing the experiment using QD-Cat-RGD-based RT combined with aCD8 to treat mice bearing 4T1 tumors. BALB/c mice (n = 5 per group) were implanted subcutaneously with 7×10^5 4T1 mammary carcinoma cells in the right hind flanks. When the tumor volumes were 100 mm³, mice received different treatments. 4T1 tumors were harvested on day 14. **b** Photographs of all tumors from individual mice. **c** Tumor growth curves in different groups. **d** Histogram plot of tumor weight in different groups. **e** Quantification of flow cytometric analysis of CD3⁺ T cells gating on CD45⁺ in the tumors of the different groups. **f** Quantification of flow cytometric analysis of CD8⁺CD3⁺ T cells gating on CD45⁺ in tumors of the different groups. All data are presented as the mean \pm s.e.m. (n = 5 per group). Statistical significance was calculated *via* one-way ANOVA with Tukey's multiple comparisons test. **P* < 0.05; ***P* < 0.01; ****P* < 0.001.

In the revised manuscript, the following statements in Results were added:

“To validate the contribution to antitumor effect of cytotoxic T cells in this combination therapy, CD8⁺ T cells were depleted using antibodies against CD8 (aCD8) in 4T1 tumor-bearing mice treated with QD-Cat-RGD-based RT (Supplementary Fig. 9a). The results demonstrated that the antitumor effects of the combination treatment were abrogated with the depletion of CD8⁺ T cells (Supplementary Fig. 9b-d). Treatment combining QD-Cat-RGD based-RT with aCD8 caused decreases in tumor-infiltrated T cells (CD3⁺) and depletion of cytotoxic CD8⁺ T cells (CD8⁺CD3⁺) (Supplementary Fig. 9e, f), confirming that CD8⁺ cytotoxic T cells were involved in the antitumor effect in primary tumor”.

In the revised manuscript, the following statements in Methods were added:

“For depletion of CD8⁺ T cells, the aCD8 antibodies (200 µg per mouse, Bioxcell, clone 53-6.7) were injected intraperitoneally on day 0, 4, 8 and 12”.

Reviewer #2

In this manuscript, the authors examine the combination of radiotherapy (RT) and administration of a nanoprobe composed of fluorescent PbS/CdS quantum dots conjugated to catalase and RGD, an avb3 integrin-targeting peptide(QD-Cat-RGD). They show that they are able to visualize the tumor using infrared (1600 nm) fluorescence imaging and examine the effect of separate or combined application of RT, QD-Cat-RGD and anti-PD1. The combination of RT and hypoxia-relieving treatment is not novel but certainly very promising, as is the use of NIR-II emitting nanoprobe to image tumors in preclinical models. The novel ideas here lie (i) in using targeted NIR-II QDs to guide the RT; (ii) in using heavy atoms in PbS/CdS as radiosensitizers to enhance RT effects; (iii) in using the QD nanoprobe as carriers to transport catalase into the tumors. Their results are very promising and interesting and will certainly be of high interest to the community.

Response:

We are very grateful to the reviewer for their careful reading and pointing out the novelty of our work. We also highly appreciate the reviewer’s suggestions for strengthening our work. By responding to the reviewer’s comments in detail and revising the manuscript accordingly, we believe that our manuscript has been significantly strengthened. We have added sufficient experimental/methodological details in the Methods section. All revisions are highlighted in red color in the revised manuscript and Supplementary Information.

Comment 1:

However, some of these novel ideas are not completely validated by the experiments performed in this study. For example, the ability of the designed QDs to target tumors is not thoroughly investigated. In particular the effect of RGD is not examined in vivo. 4T1 tumors typically exhibit a certain level of EPR-mediated uptake. Since the rationale behind the design of the probe is image-guided RT, the authors should further investigate the targeting properties of the QD-Cat-RGD. They could eg. compare the tumor uptake with and without RGD, or in avb3-negative tumor.

Response 1:

We would like to thank the reviewer for the constructive comments. According to the

reviewer's suggestion, we have added the QD-Cat (without RGD modification) group as a control group in NIR-IIb fluorescence imaging to test the effect of RGD. We have also investigated the tumor targeting properties of QD-Cat-RGD by ICP-OES compared with QD-Cat. These new results have been added to the revised manuscript.

To compare the image-guided properties of QD-Cat and QD-Cat-RGD, 4T1 tumor-bearing BALB/c mice were intravenously injected with QD-Cat-RGD (150 μL , 2 mg mL^{-1}) or QD-Cat (without RGD modification) (150 μL , 2 mg mL^{-1}) nanoprobes. The strong signals focused at the tumor site confirmed the gradual infiltration of QD-Cat-RGD nanoprobes with an excellent tumor-targeting ability (Fig. R10a). However, the QD-Cat nanoprobes without RGD modification did not show this ability, with weak signals found in tumor tissue (Fig. R10b). The tumor-to-normal tissue (T/NT) signal ratio of QD-Cat-RGD increased sharply and peaked at approximately 2 h postinjection (p.i.) with a T/NT ratio of approximately 2.742, which was higher than the T/NT ratio of QD-Cat (Fig. R10d).

To explore the precise position of QD-Cat-RGD infiltrated in tumor tissue, serial slices of tumor tissue were imaged in the NIR-IIb window and stained with hematoxylin and eosin (H&E). The merged images showed that the QD-Cat-RGD nanoprobes with RGD modification could deeply infiltrate in the tumor tissue core instead of marginal stromal tissue (Fig. R10e). Without RGD modification, QD-Cat nanoprobes were mainly found in the stroma located in the margin of tumor tissue (Fig. R10c). These results suggested that QD-Cat-RGD nanoprobes with the targeting property of RGD could migrate through the vasculature and deeply infiltrate the tumor tissue, reaching the core of tumor which was mainly under hypoxia.

Next, we selected Pb element measured by ICP-OES to investigate the pharmacokinetics of nanoprobes and biodistributions in the main organs of the body, including the liver, spleen, lungs and kidneys, as well as tumors after blood circulation. The blood circulation half-time ($T_{1/2}$) of QD-Cat-RGD was calculated to be approximately 3.37 h (Fig. R10e) according to a two-compartment model. The systemic distribution of Pb revealed a 17.67% ID g^{-1} tumor accumulation efficiency of QD-Cat-RGD (Fig. R10f), which was higher than the 6.84% ID g^{-1} tumor accumulation efficiency of QD-Cat (Fig. R10f).

These above results demonstrated a good targeting property of QD-Cat-RGD by comparing with the QD-Cat (without RGD modification) group.

Revision made:

We have also added the data of Figure R10b, c, d, e and f as “Figure 2b, c, d, e and f” in the revised manuscript.

Fig. R10 *In vivo* fluorescence imaging in the NIR-IIb window and biodistribution of QD-Cat-RGD. **a,b** Whole body and high-magnification fluorescent images of 4T1 tumor-bearing mice intravenously injected with QD-Cat-RGD (**a**) or QD-Cat (**b**) nanoprobes at different time points (5 min, 1 h, 2 h, 4 h, 8 h, 12 h, 24 h) in the NIR-IIb window. Scale bars = 5 mm. The excitation power density for NIR-IIb fluorescence imaging was 25 mW cm^{-2} provided by an 808 nm laser. **c** Serial sections of tumor tissue imaged in the NIR-IIb window or stained with H&E. The excitation power density was 25 mW cm^{-2} provided by an 808 nm laser. Scale bars = 1 mm. **d** The T (tumor)/NT (nontumor) ratio of QD-Cat-RGD or QD-Cat at different time points (5 min, 1 h, 2 h, 4 h, 8 h, 12 h, 24 h). **e** The blood concentration of Pb in BALB/c mice at different time points (0.5 h, 1 h, 3 h, 8 h, 24 h, 48 h,

72 h) after intravenous injection with QD-Cat-RGD nanoprobe. **f** Delivery efficiency for injected dose (ID) of QD-Cat-RGD or QD-Cat in tumors and main organs (livers, spleens, lungs, and kidneys) of mice injected with QD-Cat-RGD or QD-Cat at 24 h p.i. **g** The protein expression level of HIF-1 α in 4T1 tumor tissue in mice at 24 h after intravenous injection with PBS (control), QD-RGD (150 μ L, 2 mg mL⁻¹) and QD-Cat-RGD (150 μ L, 2 mg mL⁻¹) measured by western blot analysis. All data are shown as the mean \pm s.e.m. of three replicates. Statistical significance was calculated via Student's t test (**f**) and one-way ANOVA with Tukey's multiple comparisons test (**g**). * $P < 0.05$; *** $P < 0.001$.

In the revised manuscript, the following statements were added in the Results section:

“To evaluate the image-guided properties of QD-Cat-RGD nanoprobe in vivo, 4T1 tumor-bearing BALB/c mice were intravenously injected with QD-Cat-RGD (150 μ L, 2 mg mL⁻¹) or QD-Cat (150 μ L, 2 mg mL⁻¹) nanoprobe.”

“The strong signals focused at the tumor site confirmed the gradual infiltration of QD-Cat-RGD nanoprobe with an excellent tumor-targeting ability (Fig. 2a). However, the QD-Cat nanoprobe without RGD modification did not show this ability, with weak signals found in tumor tissue (Fig. 2b). The tumor-to-normal tissue (T/NT) signal ratio of QD-Cat-RGD increased sharply and peaked at approximately 2 h postinjection (p.i.) with a T/NT ratio of approximately 2.742, which was higher than the T/NT ratio of QD-Cat (Fig. 2d).”

“To explore the precise position of QD-Cat-RGD infiltrated in tumor tissue, serial slices of tumor tissue were imaged in the NIR-IIb window and stained with hematoxylin and eosin (H&E). The merged images showed that the QD-Cat-RGD nanoprobe with RGD modification could deeply infiltrate in the tumor tissue core instead of marginal stromal tissue (Fig. 2c). Without RGD modification, QD-Cat nanoprobe were mainly found in the stroma located in the margin of tumor tissue (Fig. 2c). These results suggested that QD-Cat-RGD nanoprobe with the targeting property of RGD could migrate through the vasculature and deeply infiltrate the tumor tissue, reaching the core of tumor which was mainly under hypoxia.”

“Here, we selected Pb element measured by ICP-OES to investigate the pharmacokinetics of nanoprobe and biodistributions in the main organs of the body, including the liver, spleen, lungs and kidneys, as well as tumors after blood circulation. The blood circulation half-time ($T_{1/2}$) of QD-Cat-RGD was calculated to be approximately 3.37 h (Fig. 2e) according to a two-compartment model³⁸. The systemic distribution of Pb revealed a 17.67% ID g⁻¹ tumor accumulation efficiency of QD-Cat-RGD (Fig. 2f), which was higher than the 6.84% ID g⁻¹ tumor accumulation efficiency of QD-Cat (Fig. 2f). The above results demonstrated a good property of QD-Cat-RGD by comparing with the QD-Cat (without RGD) group”.

We have also added the detailed methods about pharmacokinetics and biodistribution of QDs in the Methods:

“In vivo pharmacokinetics and biodistribution of QDs. The 4T1 tumor-bearing mice intravenously injected with QD-Cat-RGD (150 μ L, 2 mg mL⁻¹) or QD-Cat (without RGD) (150 μ L, 2 mg mL⁻¹) nanoprobe were used for pharmacokinetics and biodistribution studies.

At 0.5, 1, 3, 8, 24, 48 and 72 h p.i., 50 μ L blood from treated mouse was collected. By using inductively coupled plasma optical emission spectrometry (ICP-OES) to calculate the concentration of Pb in blood to estimate the percentage of the QD-Cat-RGD in blood. The tissues including spleen, liver, lung, kidney, and tumor were collected, weighed and then dissolved completely in 5 mL digest solution ($\text{HNO}_3 : \text{H}_2\text{O}_2 = 4:1$) for 2 h until the solution turned to clear. Subsequently, the solutions were diluted by ultrapure water to 10 mL and analyzed by ICP-OES to measure the concentration of Pb in the resulting solutions. The calculation formula of % ID g^{-1} is: (The amount of Pb element in organs) \times (The amount of Pb element in ID \times the weight of organs) $^{-1} \times 100\%$. The pharmacokinetic parameters were calculated by the software PKSolver (v. 2.0)⁶⁴”.

Comment 2:

As for using high-Z elements to enhance the effect of RT, the authors show indeed that RT+QD-Cat-RGD is more efficient in many aspects than RT alone, but there is always a significant effect of QD-Cat-RGD alone, without RT (see Figure 4 and 6 for example), compared to the control. The authors should address this point in the discussion: is the in vivo radio-sensitization effect due to the catalase, to the presence of high-Z elements, or to an indirect radio-sensitization effect? Is the combined effect a cumulative effect or a synergy? For example, what would happen if the RT was applied before the QD-Cat-RGD administration?

Response 2:

Thanks for reviewer’s insightful comment. According to reviewer’s suggestion, we used a group of mice injected with QD-Cat-RGD after RT to exploring the effects from catalases. Further, we also used QD-RGD nanoprobe (without catalase modification) as groups for comparison to reveal the effect of each component in QD-Cat-RGD in RT treatment. These new results have added to the revised manuscript.

For exploring the effect from QD-Cat-RGD for alleviating hypoxia and relieving immunosuppression, we added QD-RGD as a control group which was not modified with catalase and had no oxygenation ability. The results showed that QD-Cat-RGD could delay tumor growth (Fig. R11a, b) when RT did not apply. However, tumor volumes in the QD-RGD group were not significantly decreased, which implied an antitumor effect from relieving hypoxia by catalase. Further, a significant decrease in the M2-like macrophage infiltration (Fig. R11c) and a slight decrease in the Treg infiltration in the TME (Fig. R11f) were found in QD-Cat-RGD group. The intratumoral level of interleukin-10 (IL-10), an immunosuppressive cytokine predominantly secreted by M2 macrophages was also found to decrease in the QD-Cat-RGD group (Fig. R11g). DC and T cell infiltration were found to increase in TME after treated with QD-Cat-RGD (Fig. R11j, k). These results revealed that catalase modified on QDs could independently play a role in inhibiting tumor growth through reducing immunosuppressive cell infiltration and increasing T cell infiltration in the TME. These findings were consistent with previous reports that alleviating the hypoxia can reduce infiltration of immunosuppressive cell in the TME, which delayed tumor growth⁴⁻⁶.

We next evaluated the synergistic antitumor effect from alleviating immunosuppression by relieving hypoxia and reinforcing RT efficiency by QD-Cat-RGD nanoprobe when applying RT. We added a group that combined RT with QD-Cat (without oxygenation ability) and a group that QD-Cat-RGD injection was after RT treatment as comparable groups to explore the synergistic effect in suppressing tumor growth. The combination of alleviating immunosuppression and reinforcing RT efficiency exhibited a synergistic effect in suppressing tumor growth (Fig. R11d). RT+QD-Cat-RGD (after RT injection) and RT+QD-RGD can slightly suppress tumor growth but no significance was found when compared with RT group (Fig. R11d). M2-like macrophage infiltration and intratumoral level of IL-10 in the TME decreased after treated with QD-Cat-RGD-based RT (Fig. R11e, i). Moreover, DCs, T cells and CD8⁺ cytotoxic T cells in the TME significantly increased in the RT+QD-Cat-RGD group (Fig. R11l, m, o). These results suggested that catalase and high-Z element Pb in QD-Cat-RGD exhibited a synergistic effect in RT for suppressing tumor growth through inhibiting immunosuppression and facilitating cytotoxic T cell-mediated antitumor immunity.

Revision made:

In the revised manuscript, we have added the data of Fig. R11 as “Supplementary Fig. 10” in the Supplementary Information.

Fig. R11 BALB/c mice (n=5 per group) were implanted subcutaneously with 7×10^5 4T1 mammary carcinoma cells in the right hind flanks. When the tumor volumes were 100 mm³, mice were intravenously injected with a single dose of QD-Cat-RGD (150μL, 2 mg mL⁻¹), QD-RGD (150μL, 2 mg mL⁻¹) or phosphate-buffered saline (PBS, 150 μL). Tumors were harvested on day 14. **a** Tumor growth curves in different groups. **b, d** Histogram plot of tumor volume in non-RT groups (**b**) and RT groups (**d**). **c, e** The quantification of M2-like macrophages (CD206^{high}) gating on F4/80⁺CD11b⁺CD45⁺ cells in non-RT groups (**c**) and RT groups (**e**). **f, h** The quantification of Tregs (CD25⁺ and Foxp3⁺) gating on CD4⁺CD3⁺CD45⁺ cells in non-RT groups (**f**) and RT groups (**h**). **g, i** Cytokine level of IL-10 measured by ELISA in non-RT groups (**g**) and RT groups (**i**). **j, l** The quantification of DC (CD11c⁺) gating on CD45⁺ cells in non-RT groups (**j**) and RT groups (**l**). **k, m** The quantification of T cells (CD3⁺) gating on CD45⁺ cells in non-RT groups (**k**) and RT groups (**m**). **n, o** The quantification of cytotoxic T cells (CD8⁺CD3⁺) gating on CD45⁺ cells in non-RT groups (**n**) and RT groups (**o**). Statistical significance was calculated *via* one-way ANOVA with Tukey's multiple comparisons test. **P* < 0.05; ***P* < 0.01; ****P* < 0.001.

In the revised manuscript, the following statements were added:

“For exploring the effect from QD-Cat-RGD for alleviating hypoxia and relieving immunosuppression, we added QD-RGD as a control group which was not modified with Cat and had no oxygenation ability. The results showed that QD-Cat-RGD could delay tumor growth (Supplementary Fig. 10a, b) when RT did not apply. However, tumor volumes in the QD-RGD group were not significantly decreased, which implied an antitumor effect of relieving hypoxia by Cat. Furthermore, a significant decrease in the M2-like macrophage infiltration (Supplementary Fig. 10c) and a slight decrease in the Treg infiltration in the TME (Supplementary Fig. 10f) were found in QD-Cat-RGD group. The intratumoral level of interleukin-10 (IL-10), an immunosuppressive cytokine predominantly secreted by M2 macrophages was also found to decrease in the QD-Cat-RGD group (Supplementary Fig. 10g). DC and T cell infiltrations were found to increase in the TME after treatment with QD-Cat-RGD (Supplementary Fig. 10j, k). These results revealed that Cat modified on QDs could independently play a role in inhibiting tumor growth by reducing immunosuppressive cell infiltration and increasing T cell infiltration in the TME. These findings were consistent with previous reports demonstrating that alleviating hypoxia could reduce the infiltration of immunosuppressive cells in the TME, which delayed tumor growth^{26,48,49}. We next evaluated the synergistic antitumor effect of alleviating immunosuppression by relieving hypoxia and reinforcing RT efficiency by QD-Cat-RGD nanoprobe when combined with RT. We added a group that combined RT with QD-Cat (without Cat modification) and a group that QD-Cat-RGD injection was after RT treatment as comparable groups to explore the synergistic effect in suppressing tumor growth. The combination of alleviating immunosuppression and reinforcing RT efficiency exhibited a synergistic effect in suppressing tumor growth (Supplementary Fig. 10d). RT+QD-Cat-RGD (after RT injection) and RT+QD-RGD slightly suppressed tumor growth but no significance was found when compared with RT group (Supplementary Fig. 10d). M2-like macrophage infiltration and intratumoral level of IL-10 in the TME decreased after treatment with QD-Cat-RGD-based RT (Supplementary Fig. 10e, i). Moreover, DCs, T cells and CD8⁺ cytotoxic T cells in the

TME were significantly increased in the RT+QD-Cat-RGD group (Supplementary Fig. 10l, m, o). These results suggested that Cat and high-Z element Pb in QD-Cat-RGD exhibited a synergistic effect in RT for suppressing tumor growth through inhibiting immunosuppression and facilitating cytotoxic T cell-mediated antitumor immunity”.

Comment 3:

The origin of the radio-sensitizing effect of high Z nanoparticles is debated: given the small concentration of these elements in the tumor, the increase in physical dose deposition is not significant (see for example Rosa et al Cancer Nano 8, 2 (2017)). To be able to state that “QD-Cat-RGD nanoprobe with a high-Z element was able to enhance the radiotherapeutic efficacy”, the authors should try to separate the different components of their system, for example to test the effect of RT + QD-RGD (no catalase), or at least discuss these points in a more balanced fashion in the discussion.

Response 3:

We appreciate the reviewer’s comment. According to the reviewer’s suggestion, QD-RGD (without catalase) nanoprobe were used to detect the improvement in radiotherapeutic efficacy of Pb element in nanoprobe. We tested the surface calreticulin expression and the secretion of ATP and HMGB1 in 4T1 tumor tissue after mice treated with different treatments *in vivo*. Compared with RT group, RT+QD-RGD significantly increased surface calreticulin expression, ATP secretion and HMGB1 release but lower than RT+QD-Cat-RGD group (Fig. R12a-c). Moreover, RT+QD-RGD had a slight ability to inhibit tumor growth compared with RT group as the results shown in Fig. R11d. These findings indicated that QD-Cat-RGD with high-Z element Pb could enhance the efficacy of RT, which were consistent with *in vitro* experiments.

As suggested, we also changed this section in the discussion with a more balanced fashion: “*QD-Cat-RGD nanoprobe was able to enhance the radiotherapeutic efficacy*”.

Revision made:

In the revised manuscript, we have added the data of Fig. R12a-c as “Supplementary Fig.4a-c” in the Supplementary Information.

Fig. R12 a Flow cytometry analysis of surface calreticulin expression in tumor tissue cells with different treatments after 24 h. **b** Statistic chart of ATP concentration in tumor tissue with different treatments after 24 h. **c** Statistic chart of HMGB1 concentration in tumor tissue with different treatments after 24 h. All data are shown as the mean \pm s.e.m. of three replicates. Statistical significance was calculated *via* one-way ANOVA with Tukey's multiple comparisons test (h, i). * $P < 0.05$; ** $P < 0.01$; *** $P < 0.001$; n.s., not significant.

In the revised manuscript, the following statements were added in the Result section:

“Moreover, we further tested the effect of QD-Cat-RGD-based RT on promoting ICD *in vivo*. The 4T1 tumor-bearing BALB/c mice were intravenously injected with a single dose of QD-Cat-RGD (150 μ L, 2 mg mL⁻¹) or QD-RGD (150 μ L, 2 mg mL⁻¹). Then, the mice were treated with localized RT (12 Gy) 2 h *p.i*. The tumor tissues were harvested at 24 h after the injection of QD-Cat-RGD or QD-RGD. The results showed that tumor tissue significantly increased the surface expression of calreticulin and ATP and HMGB1 concentrations (Supplementary Fig. 4a-c) at 24 h after the mice treated with QD-RGD- or QD-Cat-RGD-based RT, which was consistent with the *in vitro* results”.

In the discussion, the previous statement was changed to:

“QD-Cat-RGD nanoprobe was able to enhance the radiotherapeutic efficacy”.

Comment 4:

Finally, a missing important element is the dose of nanoprobes delivered to the tumor: it is indeed quite important to better characterize the targeting ability as well as to understand the

mechanism behind the radiosensitizing effect. It would be relatively easy to perform elemental analysis on the tumor tissue to determine the average concentration of QDs, and thus, of catalase present in the tumor micro-environment, the spleen, liver and lung in these experiments. In addition, it would enable a comparison between the *in vivo* distribution and the range of concentration used in the cytotoxicity study (Figure S9).

Response 4:

We appreciate the reviewer’s comment. As suggested, we further selected Pb element measured by ICP-OES to investigate the biodistributions of nanoprobe in tumors and the main organs of the body, including the liver, spleen, lungs and kidneys after blood circulation. These new results have added to the revised manuscript.

We used the QD-Cat (without RGD modification) as a control. The systemic distribution of Pb revealed a 17.67% ID g⁻¹ tumor accumulation efficiency of QD-Cat-RGD, which was higher than the 6.84% ID g⁻¹ tumor accumulation efficiency of QD-Cat (Fig. R13a).

We calculated the ratio of nanoprobe weight to cell number among different organs and compared the data with maximal concentration in the cytotoxicity study (Table R1). All ratios in organs were far less than the maximum in the cytotoxicity study (Table R1), which implied nontoxic potential of QD-Cat-RGD nanoprobe *in vivo*.

Revision made:

In the revised manuscript, we have added the data of Fig. R13 as “Fig. 2f”.

Fig. R13 Delivery efficiency for injected dose (ID) of QD-Cat-RGD or QD-Cat in tumors and main organs (livers, spleens, lungs, and kidneys) of mice injected with QD-Cat-RGD or QD-Cat at 24 h p.i. Statistical significance was calculated via Student’s t test. **P* < 0.05.

Maximal concentration in cytotoxicity study (µg/cell)	Liver (µg/cell)	Spleen (µg/cell)	Tumor (µg/cell)	Lung (µg/cell)	Spleen (µg/cell)
1*10 ⁻³	1.93*10 ⁻⁷	9.35*10 ⁻⁸	1.43*10 ⁻⁷	9.65*10 ⁻⁸	1.78*10 ⁻⁸

Table R1 The ratio of nanoprobe weight to cell number among different organs and compared the data with maximal concentration in cytotoxicity study.

In the revised manuscript, the following statements were added in the Results section:

“The systemic distribution of Pb revealed a 17.67% ID g⁻¹ tumor accumulation efficiency of QD-Cat-RGD (Fig. 2f), which was higher than the 6.84% ID g⁻¹ tumor accumulation efficiency of QD-Cat (Fig. 2f)”.

We have also added the detailed methods about pharmacokinetics and biodistribution of QDs in the Methods:

“In vivo pharmacokinetics and biodistribution of QDs. The 4T1 tumor-bearing mice intravenously injected with QD-Cat-RGD (150 μL, 2 mg mL⁻¹) or QD-Cat (without RGD) (150 μL, 2 mg mL⁻¹) nanoparticles were used for pharmacokinetics and biodistribution studies. At 0.5, 1, 3, 8, 24, 48 and 72 h p.i., 50 μL blood from treated mouse was collected. By using inductively coupled plasma optical emission spectrometry (ICP-OES) to calculate the concentration of Pb in blood to estimate the percentage of the QD-Cat-RGD in blood. The tissues including spleen, liver, lung, kidney, and tumor were collected, weighed and then dissolved completely in 5 mL digest solution (HNO₃ : H₂O₂ = 4:1) for 2 h until the solution turned to clear. Subsequently, the solutions were diluted by ultrapure water to 10 mL and analyzed by ICP-OES to measure the concentration of Pb in the resulting solutions. The calculation formula of % ID g⁻¹ is: (The amount of Pb element in organs) × (The amount of Pb element in ID × the weight of organs)⁻¹ × 100%. The pharmacokinetic parameters were calculated by the software PKSolver (v. 2.0)⁶⁴”.

Comment 5:

The authors should provide absorption and fluorescence spectra of QD used, as well as an estimation of their fluorescence quantum yield.

Response 5:

Thanks for reminding us of this issue. In the revised manuscript, we added the absorption and fluorescence spectra of QD-Cat-RGD (Fig. R14a, b) in revised Supplementary Information as “Supplementary Fig. 1c, d”.

The fluorescence quantum yield (QY) of the nanoprobe was calculated to be 2.2% (referenced to IR-26 dye with a QY of 0.05%).

Revision made:

In the revised manuscript, the statements were added in the Results:

“The absorption spectrum and fluorescence emission spectrum of QD-Cat-RGD nanoparticles are shown in Supplementary Fig. 1c and d. The fluorescence quantum yield (QY) of the nanoprobe was calculated to be 2.2% (referenced to IR-26 dye with a QY of 0.05%)”.

The data in Fig. R14a and b were added as “Supplementary Fig. 1c, d” in the revised Supplementary Information.

Fig. R14 a The absorption spectrum of QD-Cat-RGD. **b** The fluorescence emission spectrum of QD-Cat-RGD.

The detailed information about absorption and fluorescence spectra in the Methods section in the revised manuscript:

“The spectra were obtained from a Lambda 750S UV–Vis–NIR spectrometer (Malvern). Photoluminescence spectra were measured using a FLS1000 fluorescence spectrometer with a Xenon lamp exciter and a PM1700 near infrared photomultiplier detector (Edinburgh Instruments)”.

Comment 6:

What is the laser intensity (mW/cm^2) used for the photo-stability measurement? Is this the same as the one used for imaging?

Response 6:

Thank you for your valuable comment. The laser intensity used for photo-stability measurement is $25 \text{ mW}/\text{cm}^2$ with an 808 nm laser, which is same to the laser intensity used for imaging. This information was added in the figure legends of “Fig. 2a, b, c” and “Supplementary Fig.1e, 2c”.

Revision made:

The figure legends of Fig.2a, b, c were changed to:

*“**a,b** Whole body and high-magnification images of 4T1 tumor-bearing mice intravenously injected with QD-Cat-RGD (**a**) or QD-Cat (**b**) nanoprobes at different time points (5 min, 1 h, 2 h, 4 h, 8 h, 12 h, 24 h) in the NIR-IIb window. Scale bars = 5 mm. The excitation power density was 25 mW cm^{-2} provided by an 808 nm laser. **c** Serial sections of tumor tissue imaged in the NIR-IIb window or stained with H&E. The excitation power density was 25 mW cm^{-2} provided by an 808 nm laser.”.*

The figure legend of Supplementary Fig. 1e, 2c were changed to:

*“**e** The fluorescence intensity of QD-Cat-RGD in fetal bovine serum (FBS) under continuous 808 nm laser exposure (25 mW cm^{-2}) for 1 h”.*

*“**c** Fluorescence intensity in the NIR-IIb window of different organs. Scale bar = 1 cm. The excitation power density for NIR-IIb fluorescence imaging was 25 mW cm^{-2} provided by an 808 nm laser”.*

Comment 7:

The authors state that “With QDs modified by PEG, QD-Cat-RGD nanoprobe could circulate in the blood for a long time, and only approximately 16.7% of the QD-Cat-RGD nanoprobe had been taken up by the hemocytes 30 min postinjection (p.i.) (Fig. 2c).”, but the blood residence/circulation time and uptake by hematocytes are unrelated, as nanoparticles are usually eliminated from the circulation by uptake by macrophages in the liver and spleen or by accumulation in the lung (as seems to be the case here, and which is usually seen as a consequence of the instability and aggregation of NPs in the bloodstream).

Response 7:

We thank the reviewer for this important comment. We apologize for the inaccurate statement that “*With QDs modified by PEG, QD-Cat-RGD nanoprobe could circulate in the blood for a long time, and only approximately 16.7% of the QD-Cat-RGD nanoprobe had been taken up by the hemocytes 30 min postinjection (p.i.)*”. We highly agree with point mentioned by the reviewer that “nanoparticles are usually eliminated from the circulation by uptake by macrophages in the liver and spleen or by accumulation in the lung”.

According to the reviewer’s suggestion, we tested the blood concentration of Pb by ICP-OES and obtained the pharmacokinetic parameters and blood circulation half-time of QD-Cat-RGD to replace the old statement. The new result showed that the blood circulation half-time ($T_{1/2}$) of QD-Cat-RGD was calculated to be approximately 3.37 h according to a two-compartment model (Fig. R15). These new results have added to the revised manuscript as Fig. 2e.

Revision made: In the revised manuscript, the following statements were added in the Results section:

“Here, we selected Pb element measured by ICP-OES to investigate the pharmacokinetics of nanoprobe and biodistributions in the main organs of the body, including the liver, spleen, lungs and kidneys, as well as tumors after blood circulation. The blood circulation half-time ($T_{1/2}$) of QD-Cat-RGD was calculated to be approximately 3.37 h (Fig. 2e) according to a two-compartment model³⁸.”

The data of Fig. R15 has been added in the manuscript as “Fig. 2e”.

Fig. R15 The blood concentration of Pb in BALB/c mice at different time points (0.5 h, 1 h, 3 h, 8 h, 24 h, 48 h, 72 h) after intravenous injection with QD-Cat-RGD nanoprobe.

Comment 8:

From the experimental section it is not clear how catalase is coupled to the QDs. Is it only by adsorption? no reagent seem to be used to covalently couple the catalase on the QDs. What is the approximate number of active catalase per QD?

Response 8:

We thank the reviewer for the kind suggestion. According to reviewer's suggestion, the loading capacity of RGD and catalases on the QD-Cat-RGD nanoprobe were added in the revised manuscript. The detailed steps of how catalase is coupled to the QDs was added in the Methods section.

RGD and catalase were conjugated on the surface of QDs via N-(3-(dimethylamino)propyl)-N'-ethylcarbodiimide hydrochloride (EDC) chemistry (Hermanson, G. T. *Bioconjugate Techniques*. Academic Press, 2008)⁷. The carboxylic groups on QDs were activated by EDC molecules, and then react with the amine groups on RGD and catalase to form amide bonds.

To reveal the loading capacity of RGD and catalase on the QD-Cat-RGD nanoprobe, we used a standard BCA protein assay to quantify the unbound catalase after the conjugation and purification of nanoprobe and consequently obtained the amounts of catalase grafted on QDs. The molar ratio of grafted catalase to QDs was about 2.5:1. We next used the UV spectrometry to detect the absorbance at 220 nm to quantify the unbound RGD (Boyer, R. F. *Modern Experimental Biochemistry*. Prentice Hall PTR, 2000), and further obtained the amounts of RGD grafted on QDs. We calculated that the molar ratio of grafted RGD to QDs was about 30:1.

We have added the detailed description about preparation of QD-Cat-RGD nanoprobe in the Methods.

Revision made:

In the revised manuscript, the statements about how catalase is coupled to the QDs were added in Results:

“Then, 8-Arm PEG-NH₂ molecules, RGD peptides, and Cat were conjugated on the surface of OPA-modified QDs via N-(3-(dimethylamino)propyl)-N'-ethylcarbodiimide hydrochloride (EDC) chemistry³². The carboxylic groups on QDs were activated by EDC molecules and then reacted with the amine groups on Cat to form amide bonds.”.

In the revised manuscript, the following statements about the loading capacity of RGD and catalase on the QD-Cat-RGD nanoprobe in Result section were added:

“To reveal the loading capacity of RGD and Cat on the QD-Cat-RGD nanoprobe, we used a standard BCA protein assay to quantify the unbound Cat after the conjugation and purification of nanoprobe and consequently obtained the amounts of Cat grafted on QDs. The molar ratio of grafted Cat onto QDs was approximately 2.5:1. We further used UV spectrometry to detect the absorbance at 220 nm to quantify the unbound RGD and obtain the amounts of RGD grafted on QDs³³. We calculated that the molar ratio of grafted RGD to QDs

was approximately 30:1”.

In Methods, we added the detailed description about the preparation of QD-Cat-RGD nanoprobes:

“Preparation of QD-Cat-RGD Nanoprobes. The as-synthesized QDs were modified as previously described⁶². Polyacrylic acid (average Mw ~ 1800, 0.9 g) and DCC (1.56 g) were mixed into a round-bottom flask. DMF (10 mL) was added to dissolve the mixture. Subsequently, OLA (1,2 mL, molar ratio of OLA to PAA is 30%) were added dropwise into the reaction flask. The mixture solution was stirred overnight, and 0.5 M HCl (50 mL) were added to the mixture solution. The precipitate was isolated by centrifugation and re-dissolved in methanol (3 mL). Then, 1 M HCl (20 mL) was added to the solution and the precipitate was isolated by centrifugation. The precipitate was dissolved in chloroform (5 mL) and washed by 1 M HCl (10 mL). Collecting and drying over the organic phase with anhydrous Na₂SO₄. Under a vacuum, chloroform was removed and the oleyamine-branched poly (acrylic acid) (OPA) were collected (with an average Mw of ~ 3000 determined by gel permeation chromatography). Subsequently, PbS/CdS QDs (5.0 mg) were added to mixture solution (15 mg OPA dissolved in 2.0 mL chloroform). The mixture was stirred for 30 min at room temperature and the solvent was removed by a rotary evaporator under a vacuum. Under the sonication, the products were dissolved completely in 2 mL of 50 mM sodium carbonate solution. The QDs were precipitated for 1 h at 75,000 rpm by ultracentrifuge, and dissolved in 2 mL pH 8.5 MES buffer (0.01 M). The 8-Arm PEG-amine molecules, RGD peptides, and Cat were dissolved in 500 μL pH 8.5 MES buffer (0.01 M) and gradually added to the OPA-modified QD solution with stirring. EDC (15 mg) were dissolved in 150 μL pH 8.5 MES buffer (0.01 M), and gradually added to the mixture solution with stirring and reacted overnight. The QD-Cat-RGD nanoprobes were precipitated with ultracentrifugation at 75,000 rpm for 30 min to remove excess reactants. The precipitate was dissolved in 1 × PBS and stored at 4 °C.”

Reviewer #3

Summary

The manuscript describes the combination of a nanoprobe with radiation therapy to improve local and distant tumor control. The nanoprobe is designed to target the tumor via the well-studied RGD motifs, and are intended to relieve hypoxia, permitting increased radiation induced DNA damage. The authors demonstrate that the nanoprobes improve cancer killing in vitro, and provide some evidence of altered hypoxia responses in vivo. The combination of RT and nanoprobes improves tumor control, and improves control of delayed administration distant tumors or metastases. Some additional effects of anti-PD1 are demonstrated.

The main problem with the manuscript is that the core contention is not well demonstrated. It is unclear whether these probes delivered IV and partially enriching at the tumor can improve hypoxia in the tumor environment purely through this level of delivery of catalase. The

experiments as presented do not prove this mechanism. There are issues in the experimental designs and results that bring the proposed mechanism into question. This, combined with the relative lack of novelty in the manuscript limit its value to the field.

Response:

We are very grateful for the reviewer's comments. We also highly appreciate the reviewer's suggestions for strengthening our work. We have performed more experiments to address the reviewer's concerns. By responding to the reviewer's comments in detail and revising the manuscript accordingly, we believe this manuscript has been strengthened. In addition, we have added sufficient experimental/methodological details in the Methods section. All revisions are highlighted in red color in the revised manuscript and Supplementary Information.

Comment 1:

The authors state that they use the signal from the nanoparticle to direct their radiation therapy, but their radiation equipment is a RS2000Pro cabinet irradiator. This has no targeting capacity, instead is a cone-beam x-ray source. These facts don't match. The investigators must alter their language, and explain how tumor-selective treatment was given, with dosimetry validation and what non-tumor structures were included in the field. If imaging was used to target delivery, the method to integrate imaging with treatment should be described.

Response 1:

We appreciate the reviewer's comment. We apologize for this misleading statement in the manuscript. In this study, we combined NIR-IIb fluorescence imaging and a special lead shield to localize the range of RT in the tumor site. In the revised manuscript, we have altered our description to clarify the meaning.

Considering that NIR-IIb fluorescence imaging can show the area of tumor in mouse after injection of QD-Cat-RGD, we first determined the location and range of RT with the help of NIR-IIb imaging (Fig. R16a). Then, we used the lead shield to localize the range of RT determined by previous imaging (Fig. R16b, c). The mice were treated with RT (12 Gy) at 2 h p.i. Dosimetry validation was performed by the Accu-Dose+ stream-line diagnostic solution (Radcal).

Revision made:

In the revised manuscript, the description about RT treatment was changed to:

“Considering that NIR-IIb fluorescence imaging can show the area of tumors in mice after injection of QD-Cat-RGD, we first determined the location and range of RT with the help of NIR-IIb imaging (Supplementary Fig. 5a). Then, we used the lead shield to localize the range of RT determined by previous imaging (Supplementary Fig. 5b, c). The mice were treated with RT (12 Gy) 2 h p.i.”

The data of Fig. R16 has been added in the Supplementary Information as Supplementary Fig. 5.

Fig. R16 **a** The location and range of tumor were clearly demonstrated in the NIR-IIb fluorescence imaging after QD-Cat-RGD injection. **b, c** The special lead shield for tumor-bearing mouse to localize the range of RT determined by the NIR-IIb imaging. Dosimetry validation of RT was made by the Accu-Dose+ stream-line diagnostic solution (Radcal).

Comment 2:

The approach used to demonstrate *in vitro* killing by RT plus nanoprobe is a problem (Figure 3). The cultures were not under hypoxic conditions – in fact one major criticism of standard *in vitro* culture of cancer cells is that it is fully normoxic. If the nanoprobe can help kill cancer cells in this setting, then it is unlikely that overcoming hypoxia is part of the mechanism.

Response 2:

Thanks for raising this important concern. We apologized for the wrong normoxic condition used in the experiments *in vitro*. We have repeated the *in vitro* experiments under hypoxic conditions. These results have added to the revised manuscript.

To study the effect of nanoprobe on cells under hypoxia, cells were incubated under a hypoxic condition (37°C, 1% O₂, 5% CO₂) following by different treatment and we repeated the experiments in “Fig. 3” which was shown in Fig. R17.

To evaluate the QD-Cat-RGD nanoprobe’s potential as a radiosensitizer for RT, a colony formation assay was conducted to assess the sensitivity of 4T1 cancer cells to different treatments in hypoxia *in vitro*. The results showed that the proliferative activation of 4T1 cells treated with QD-RGD- or QD-Cat-RGD-based RT in hypoxia significantly decreased compared with RT in hypoxia (Fig. R17a, b). The radiosensitization of QD-RGD and QD-Cat-RGD was further confirmed by p-H2AX staining, which reflected the DNA damage of cancer cells. Consistent with the results of the colony formation assay, the fluorescence intensity of p-H2AX in the nuclei of 4T1 cells was significantly increased in the hypoxia +

RT + QD-RGD and hypoxia + RT + QD-Cat-RGD groups compared with the RT group under a hypoxic condition (Fig. R17c). In addition, an annexin V/propidium iodide (PI) staining assay suggested that the number of vital 4T1 cells decreased in the hypoxia + RT+QD-RGD and hypoxia + RT + QD-Cat-RGD groups compared with the other groups (Fig. R17d).

Considering that QD-Cat-RGD nanoprobe significantly promoted the RT efficacy of directly killing cancer cells, we then evaluated the effect of QD-Cat-RGD-based RT on promoting the ICD of cancer cells. RT can trigger ICD of cancer cells through emission of tumor-associated antigen and ICD-associated danger-associated molecular patterns (DAMPs), such as exposure of calreticulin on the surface of dying cancer cells, and the release of large amounts of high-mobility group box 1 (HMGB1) and adenosine triphosphate (ATP) into the extracellular milieu, consequently resulting in activation of antitumor immunity (Fig. R17e). As measured by fluorescence imaging and flow cytometry, the surface expression of calreticulin in 4T1 cells in the hypoxia + RT + QD-RGD and hypoxia + RT + QD-Cat-RGD groups was significantly increased compared with that in the 4T1 cells treated with hypoxia + RT only (Fig. R17f, g). We also tested ATP and HMGB1 release in cancer cells, two crucial signals during DC recruitment and maturation. At 24 h after RT, 4T1 cells treated with QD-RGD- or QD-Cat-RGD-based RT under a hypoxic condition significantly increased ATP secretion (Fig. R17h) and HMGB1 release (Fig. R17i), as measured by ATP assay and enzyme-linked immunosorbent assay (ELISA). With HMGB1 release, the expression of HMGB1 in the cell nuclei of 4T1 cells in the hypoxia + RT +QD-RGD and hypoxia + RT + QD-Cat-RGD groups decreased compared with those only treated with hypoxia + RT (Fig. R17j).

Revision made:

In the revised manuscript, the “Fig. 3” was replaced to the figure as shown in Fig. R17:

Fig. R17 QD-Cat-RGD as a radiosensitizer for enhancing ICD of cancer cells under hypoxic conditions. 4T1 cells were incubated under hypoxic conditions (37 °C, 1% O₂, 5% CO₂) and irradiated with X-ray doses of 6 Gy *in vitro*. **a** Representative photographs of stained colonies of 4T1 cells treated with hypoxia, hypoxia + QD-RGD, hypoxia + QD-Cat-RGD, hypoxia + RT (6 Gy), hypoxia + RT + QD-RGD or hypoxia + RT + QD-Cat-RGD after 7 days. **b** Histogram plot of the survival fraction of 4T1 cells with different treatments. **c** Immunofluorescence staining of p-H2AX (Ser139) and quantitative analysis of foci density of foci per cell at 4 h after different treatments. Scale bars = 25 μm. **d** Apoptosis analysis measured by flow cytometry of 4T1 cells at 24 h after different treatments. **e** QD-Cat-RGD reinforces the ICD induced by RT and emits several types of DAMPs from dying cancer

cells. **f** Immunofluorescence staining of calreticulin of 4T1 cells with different treatments after 24 h. Scale bars = 25 μm . **g** Flow cytometry analysis of surface calreticulin expression in 4T1 cells with different treatments after 24 h. **h** Histogram plot of secreted ATP production by 4T1 cells with different treatments after 24 h. **i** Histogram plot of HMGB1 released by 4T1 cells with different treatments after 24 h. **j** Immunofluorescence staining of HMGB1 in 4T1 cells with different treatments after 24 h. Scale bars = 25 μm . All data are shown as the mean \pm s.e.m. of three replicates. Statistical significance was calculated *via* one-way ANOVA with Tukey's multiple comparisons test. * $P < 0.05$; ** $P < 0.01$; *** $P < 0.001$; n.s., not significant.

In the revised manuscript, the description of experiments *in vitro* in “Fig. 3” was changed to:

“Reinforcement of ICD of cancer cells induced by RT. Since QD-Cat-RGD based on PbS/CdS QDs contain high-Z element Pb and Cat which could relieve hypoxia via the catalysis of endogenous H_2O_2 in the TME, they have a strong potential to promote RT efficacy. To study the effect of nanoprobles and RT on cells under hypoxia, cells were incubated under hypoxic condition (37 $^\circ\text{C}$, 1% O_2 , 5% CO_2). To evaluate the QD-Cat-RGD nanoprobe's potential as a radiosensitizer for RT, a colony formation assay was conducted to assess the sensitivity of 4T1 cancer cells to different treatments in hypoxia *in vitro*. The results showed that the proliferative activation of 4T1 cells treated with QD-RGD- or QD-Cat-RGD-based RT in hypoxia significantly decreased compared with RT in hypoxia (Fig. 3a, b). The radiosensitization of QD-RGD and QD-Cat-RGD was further confirmed by p-H2AX staining, which reflected the DNA damage of cancer cells^{39,40}. Consistent with the results of the colony formation assay, the fluorescence intensity of p-H2AX in the nuclei of 4T1 cells was significantly increased in the hypoxia + RT + QD-RGD and hypoxia + RT + QD-Cat-RGD groups compared with the RT group under hypoxic conditions (Fig. 3c). In addition, an annexin V/propidium iodide (PI) staining assay suggested that the number of vital 4T1 cells decreased in the hypoxia + RT + QD-RGD and hypoxia + RT + QD-Cat-RGD groups compared with the other groups (Fig. 3d).

Considering that QD-Cat-RGD nanoprobles significantly promoted the RT efficacy of directly killing cancer cells, we then evaluated the effect of QD-Cat-RGD-based RT on promoting the ICD of cancer cells. RT can trigger ICD of cancer cells through emission of tumor-associated antigen and ICD-associated danger-associated molecular patterns (DAMPs), such as exposure of calreticulin on the surface of dying cancer cells, and the release of large amounts of high-mobility group box 1 (HMGB1) and adenosine triphosphate (ATP) into the extracellular milieu, consequently resulting in activation of antitumor immunity (Fig.3e)⁹. As measured by fluorescence imaging and flow cytometry, the surface expression of calreticulin in 4T1 cells in the hypoxia + RT + QD-RGD and hypoxia + RT + QD-Cat-RGD groups was significantly increased compared with that in the 4T1 cells treated with hypoxia + RT only (Fig. 3f, g and Supplementary Fig. 3). We also tested ATP and HMGB1 release in cancer cells, two crucial signals during DC recruitment and maturation⁹. At 24 h after RT, 4T1 cells treated with QD-RGD- or QD-Cat-RGD-based RT under hypoxic conditions exhibited significantly increased ATP secretion (Fig. 3h) and HMGB1 release (Fig. 3i), as measured by ATP assay and enzyme-linked immunosorbent assay (ELISA). With HMGB1 release, the expression of HMGB1 in the cell nuclei of 4T1 cells in the hypoxia + RT + QD-RGD and hypoxia + RT + QD-Cat-RGD groups decreased compared with those only

treated with hypoxia + RT (Fig. 3j).

Moreover, we further tested the effect of QD-Cat-RGD-based RT on promoting ICD in vivo. The 4T1 tumor-bearing BALB/c mice were intravenously injected with a single dose of QD-Cat-RGD (150 μ L, 2 mg mL⁻¹) or QD-RGD (150 μ L, 2 mg mL⁻¹). Then, the mice were treated with localized RT (12 Gy) 2 h p.i. The tumor tissues were harvested at 24 h after the injection of QD-Cat-RGD or QD-RGD. The results showed that tumor tissue significantly increased the surface expression of calreticulin and ATP and HMGB1 concentrations (Supplementary Fig. 4a-c) at 24 h after the mice treated with QD-RGD- or QD-Cat-RGD-based RT, which was consistent with the in vitro results. This evidence demonstrated that QD-Cat-RGD containing high-Z element Pb and Cat could significantly promote RT efficiency and reinforce the ICD of cancer cells under hypoxic conditions, which could potentially promote more recruitment and maturation of DCs⁴¹.”

Comment 3:

The imaging location of the nanoprobe appears to show poor localization to the tumor core, where hypoxia is most relevant. The probes appear to accumulate at the tumor margins where the more normal vasculature is most able to deliver these probes. In addition, trans-vascular passage of these probes into the tumor stroma and to the cancer cells has not been demonstrated.

Response 3:

We thank the reviewer for this helpful comment. According to the reviewer's suggestion, we further revealed the precise position of QD-Cat-RGD nanoprobe in tumor tissue. In addition, we added QD-Cat (without RGD modification) as a control group which have low tumor targeting ability. These new results have been added to the revised manuscript.

To explore the precise position of QD-Cat-RGD infiltrated in tumor tissue, serial section of tumor tissue were imaged in NIR-IIb window and stained with hematoxylin and eosin (H&E). The merged images showed that the QD-Cat-RGD nanoprobe with RGD modification could deeply infiltrate in tumor tissue core instead of marginal stromal tissue (Fig. R18). Without RGD modification, QD-Cat nanoprobe were mainly found in the stroma located in the margin of tumor tissue (Fig. R18). These results suggested that QD-Cat-RGD nanoprobe with the targeting property of RGD could migrate through the vascular, deeply infiltrate in the tumor tissue reaching the core of tumor which was mainly under hypoxia.

Revision made:

The images of Fig. R18 has been added in the manuscript as “Fig. 2c”.

Fig. R18 Serial sections of tumor tissue imaged in NIR-IIb window or stained with H&E. Scale bars = 1 mm. The excitation power density for NIR-IIb fluorescence imaging was 25 mW cm⁻² provided by an 808 nm laser.

In the revised manuscript, the following sentences in Result section were added:

“To explore the precise position of QD-Cat-RGD infiltrated in tumor tissue, serial slices of tumor tissue were imaged in the NIR-IIb window and stained with hematoxylin and eosin (H&E). The merged images showed that the QD-Cat-RGD nanoprobe with RGD modification could deeply infiltrate in the tumor tissue core instead of marginal stromal tissue (Fig. 2c). Without RGD modification, QD-Cat nanoprobe were mainly found in the stroma located in the margin of tumor tissue (Fig. 2c). These results suggested that QD-Cat-RGD nanoprobe with the targeting property of RGD could migrate through the vasculature and deeply infiltrate the tumor tissue, reaching the core of tumor which was mainly under hypoxia.”

Comment 4:

If the nanoprobe function via improved oxygenation of the tumor, then there may need to be an explanation for their ability to control tumor growth even in the absence of radiation therapy. Why do tumors grow slower when oxygenation is improved? Notably, in the dual tumor model, the probes can almost cure distant tumors as a single agent, when administered

1d following that second tumor implantation. This second tumor would both have a small number of cancer cells in the implantation site to bind the probes, and low hypoxia. When hypoxia develops through tumor growth, the nanoprobe would remain the same in number with increased numbers of cancer cells, yet they have an effect. These data suggest hypoxia regulation is not the mechanism.

Response 4:

Thanks for reviewer's insightful comment. As suggested, we added several control groups to further validate that our nanoprobe could alleviate hypoxia, resulting in relieving immunosuppression in the tumor microenvironment (TME) and finally suppressing tumor growth. Moreover, this immuno-effect combined with the radiosensitivity from QD-Cat-RGD can exhibit a synergistic effect in RT treatment for suppressing tumor growth through facilitating cytotoxic T cell-mediated antitumor immunity. These new results shown in Fig. R19 have added to the revised manuscript as "Supplementary Fig. 10".

For exploring the effect from QD-Cat-RGD for alleviating hypoxia and relieving immunosuppression, we added QD-RGD as a control group which was not modified with catalase and had no oxygenation ability. The results showed that QD-Cat-RGD could delay tumor growth (Fig. R19 a, b) when RT did not apply. However, tumor volumes in the QD-RGD group were not significantly decreased, which implied an antitumor effect from relieving hypoxia by catalase (Fig. R19b). Further, a significant decrease in the M2-like macrophages infiltration (Fig. R19c) and a slight decrease in the Tregs infiltration in the TME (Fig. R19f) were found in QD-Cat-RGD group. The intratumoral level of interleukin-10 (IL-10), an immunosuppressive cytokine predominantly secreted by M2 macrophages was also found to decrease in the QD-Cat-RGD group (Fig. R19g). DC and T cells infiltration were found to increase in TME after treated with QD-Cat-RGD (Fig. R19j, k). These results revealed that catalase modified on QDs could independently play a role in inhibiting tumor growth through reducing immunosuppressive cells infiltration and increasing T cells infiltration in the TME. These findings were consistent with previous reports that alleviating the hypoxia can reduce infiltration of immunosuppressive cells in the TME, which delayed tumor growth⁴⁻⁶.

We next evaluated the synergistic antitumor effect from alleviating immunosuppression by relieving hypoxia and reinforcing RT efficiency by QD-Cat-RGD nanoprobe when applying RT. We added a group that combined RT with QD-Cat (without oxygenation ability) and a group that QD-Cat-RGD injection was after RT treatment as comparable groups to explore the synergistic effect in suppressing tumor growth. The combination of alleviating immunosuppression and reinforcing RT efficiency exhibited a synergistic effect in suppressing tumor growth (Fig. R19d). RT+QD-Cat-RGD (after RT injection) and RT+QD-RGD can slightly suppress tumor growth but no significance was found when compared with RT group (Fig. R19d). M2-like macrophages infiltration and intratumoral level of IL-10 in the TME decreased after treated with QD-Cat-RGD-based RT (Fig. R19e, i). Moreover, DCs, T cells and CD8⁺ cytotoxic T cells in the TME significantly increased in the RT+QD-Cat-RGD group (Fig. R19l, m, o). These results suggested that catalase and high-Z element Pb in QD-Cat-RGD exhibited a synergistic effect in RT for suppressing tumor growth through inhibiting immunosuppression and facilitating cytotoxic T cell-mediated antitumor immunity.

Revision made:

In the revised manuscript, we have added the data of Fig. R19 as “Supplementary Fig. 10” in the Supplementary Information.

Fig. R19 BALB/c mice (n=5 per group) were implanted subcutaneously with 7×10^5 4T1 mammary carcinoma cells in the right hind flanks. When the tumor volumes were 100 mm³, mice were intravenously injected with a single dose of QD-Cat-RGD (150μL, 2 mg mL⁻¹), QD-RGD (150μL, 2 mg mL⁻¹) or phosphate-buffered saline (PBS, 150 μL). Tumors were harvested on day 14. **a** Tumor growth curves in different groups. **b, d** Histogram plot of tumor volume in non-RT groups (**b**) and RT groups (**d**). **c, e** The quantification of M2-like macrophages (CD206^{high}) gating on F4/80⁺CD11b⁺CD45⁺ cells in non-RT groups (**c**) and RT groups (**e**). **f, h** The quantification of Tregs (CD25⁺ and Foxp3⁺) gating on CD4⁺CD3⁺CD45⁺ cells in non-RT groups (**f**) and RT groups (**h**). **g, i** Cytokine level of IL-10 measured by ELISA in non-RT groups (**g**) and RT groups (**i**). **j, l** The quantification of DC (CD11c⁺) gating on CD45⁺ cells in non-RT groups (**j**) and RT groups (**l**). **k, m** The quantification of T cells (CD3⁺) gating on CD45⁺ cells in non-RT groups (**k**) and RT groups (**m**). **n, o** The quantification of cytotoxic T cells (CD8⁺CD3⁺) gating on CD45⁺ cells in non-RT groups (**n**) and RT groups (**o**). Statistical significance was calculated *via* one-way ANOVA with Tukey's multiple comparisons test. * $P < 0.05$; ** $P < 0.01$; *** $P < 0.001$.

In the revised manuscript, the following statements were added:

“For exploring the effect from QD-Cat-RGD for alleviating hypoxia and relieving immunosuppression, we added QD-RGD as a control group which was not modified with Cat and had no oxygenation ability. The results showed that QD-Cat-RGD could delay tumor growth (Supplementary Fig. 10a, b) when RT did not apply. However, tumor volumes in the QD-RGD group were not significantly decreased, which implied an antitumor effect of relieving hypoxia by Cat. Furthermore, a significant decrease in the M2-like macrophage infiltration (Supplementary Fig. 10c) and a slight decrease in the Treg infiltration in the TME (Supplementary Fig. 10f) were found in QD-Cat-RGD group. The intratumoral level of interleukin-10 (IL-10), an immunosuppressive cytokine predominantly secreted by M2 macrophages was also found to decrease in the QD-Cat-RGD group (Supplementary Fig. 10g). DC and T cell infiltrations were found to increase in the TME after treatment with QD-Cat-RGD (Supplementary Fig. 10j, k). These results revealed that Cat modified on QDs could independently play a role in inhibiting tumor growth by reducing immunosuppressive cell infiltration and increasing T cell infiltration in the TME. These findings were consistent with previous reports demonstrating that alleviating hypoxia could reduce the infiltration of immunosuppressive cells in the TME, which delayed tumor growth^{26,48,49}. We next evaluated the synergistic antitumor effect of alleviating immunosuppression by relieving hypoxia and reinforcing RT efficiency by QD-Cat-RGD nanoprobe when combined with RT. We added a group that combined RT with QD-Cat (without Cat modification) and a group that QD-Cat-RGD injection was after RT treatment as comparable groups to explore the synergistic effect in suppressing tumor growth. The combination of alleviating immunosuppression and reinforcing RT efficiency exhibited a synergistic effect in suppressing tumor growth (Supplementary Fig. 10d). RT+QD-Cat-RGD (after RT injection) and RT+QD-RGD slightly suppressed tumor growth but no significance was found when compared with RT group (Supplementary Fig. 10d). M2-like macrophage infiltration and intratumoral level of IL-10 in the TME decreased after treatment with QD-Cat-RGD-based RT (Supplementary Fig. 10e, i). Moreover, DCs, T cells and CD8⁺ cytotoxic T cells in the

TME were significantly increased in the RT+QD-Cat-RGD group (Supplementary Fig. 10l, m, o). These results suggested that Cat and high-Z element Pb in QD-Cat-RGD exhibited a synergistic effect in RT for suppressing tumor growth through inhibiting immunosuppression and facilitating cytotoxic T cell-mediated antitumor immunity”.

Comment 5:

The immune mechanism of primary tumor treatment is not proven. While there are a range of immune correlates in the tumor following radiation therapy with the nanoprobe present, there is no proof that the mechanism is immune. As the investigators show, the combination is more effective at killing cancer cells in vitro. Therefore, the efficacy to the local tumor could be entirely independent of immune mechanisms. Additional studies are necessary to demonstrate that the local tumor control mechanism is immune.

Response 5:

We thank the reviewer for the valuable suggestion. We have performed more experiments to address the reviewer’s concerns. Considering the increase in cytotoxic T cells after QD-Cat-RGD-based RT treatment, we used an antibody against CD8 (aCD8) to deplete cytotoxic T cells in mice to reveal the effect of immunity in primary tumor treatment. The results have been added to the revised Supplementary Information as “Supplementary Fig. 9”.

To validate the contribution to anti-tumor effect from T cells especially cytotoxic T cells, CD8⁺ T cells were depleted using antibodies against CD8 (aCD8) on day 0, 4, 8 and 12 in 4T1 tumor-bearing mice treated with QD-Cat-RGD-based RT (Fig. R20a). The results demonstrated that antitumor effects of the combination treatment were abrogated with the depletion of CD8⁺ T cells, resulting in increase of the volume and weight of tumors (Fig. R20b, c, d). The treatment combining QD-Cat-RGD-based RT with aCD8 also caused decreases in tumor-infiltrated T cells (CD3⁺) and depletion of cytotoxic CD8⁺ T cells (CD8⁺CD3⁺) (Fig. R20e, f). These data suggested that CD8⁺ cytotoxic T cells were involved in the antitumor effect in primary tumor.

Revision made:

We also have added the data of Fig. R20 as “Supplementary Fig. 9” in the Supplementary Information.

Fig. R20 QD-Cat-RGD-based RT reducing the effect inhibiting tumor growth when depleting the CD8⁺ cytotoxic T cells. **a** Schematic showing the experiment using QD-Cat-RGD-based RT combined with aCD8 to treat mice bearing 4T1 tumors. BALB/c mice (n = 5 per group) were implanted subcutaneously with 7×10^5 4T1 mammary carcinoma cells in the right hind flanks. When the tumor volumes were 100 mm³, mice received different treatments. 4T1 tumors were harvested on day 14. **b** Photographs of all tumors from individual mice. **c** Tumor growth curves in different groups. **d** Histogram plot of tumor weight in different groups. **e** Quantification of flow cytometric analysis of CD3⁺ T cells gating on CD45⁺ in the tumors of the different groups. **f** Quantification of flow cytometric analysis of CD8⁺CD3⁺ T cells gating on CD45⁺ in tumors of the different groups. All data are presented as the mean \pm s.e.m. (n = 5 per group). Statistical significance was calculated *via* one-way ANOVA with Tukey's multiple comparisons test. **P* < 0.05; ***P* < 0.01;

*** $P < 0.001$.

In the revised manuscript, the following statements in Results were added:

“To validate the contribution to antitumor effect of cytotoxic T cells in this combination therapy, $CD8^+$ T cells were depleted using antibodies against CD8 (aCD8) in 4T1 tumor-bearing mice treated with QD-Cat-RGD-based RT (Supplementary Fig. 9a). The results demonstrated that the antitumor effects of the combination treatment were abrogated with the depletion of $CD8^+$ T cells (Supplementary Fig. 9b-d). Treatment combining QD-Cat-RGD based-RT with aCD8 caused decreases in tumor-infiltrated T cells ($CD3^+$) and depletion of cytotoxic $CD8^+$ T cells ($CD8^+CD3^+$) (Supplementary Fig. 9e, f), confirming that $CD8^+$ cytotoxic T cells were involved in the antitumor effect in primary tumor”.

In the revised manuscript, the following statements in Methods were added:

“For depletion of $CD8^+$ T cells, the aCD8 antibodies (200 μ g per mouse, Bioxcell, clone 53-6.7) were injected intraperitoneally on day 0, 4, 8 and 12”.

Comment 6:

The flow cytometry of macrophages in Figure 4 is unconvincing. There are not distinct populations, rather a slight alteration in MFI of CD80 and CD206.

Response 6:

We thank the reviewer for this comment. We apologize for the missing description for determining the gate of $CD206^{high}$ or $CD80^{high}$ TAMs in the manuscript. Unlike CD8 or CD4 with good separation between positive and negative cell populations, CD80 and CD206 are makers with unclear discrimination between dimly stained and negative cell populations (Nat Rev Immunol. 2004, 4, 648–655) (Immunology. 1996, 89(4), 592–598)^{8,9}. Therefore, we used a 'fluorescence-minus-one' (FMO) control (Nat Rev Immunol. 2004, 4, 648–655)⁹ to determine the positive gate of CD80 and CD86, which is shown in Fig. R21.

We have added the description for determining $CD206^{high}$ or $CD80^{high}$ TAMs in the Methods section.

Fig. R21 The gating strategy of $CD206^{high}$ or $CD80^{high}$ TAMs cells determined by FMO.

Revision made:

We also have added the description for determining the gate of CD206^{high} or CD80^{high} TAMs in the Methods section:

“The gating strategy used for the study was shown in Supplementary Fig. 6. The positive gate of CD206^{high} or CD80^{high} TAMs were determined by fluorescence-minus-one (FMO) control⁶⁸”.

Comment 7:

In figure 5, cells in the TDLN are analyzed. There is a little confusion since Figure 5d appears to be IHC of the tumor, but based on the text in the manuscript, Figure 5c and 5e are analyses of T cells in the TDLN. These data are unexpected. Firstly, in an untreated TDLN only 5% of the CD45⁺ cells are shown to be CD3⁺ cells. This is not plausible. Proportions more like 40-60% of CD45⁺ cells are more normal. Secondly, all of these cells are CD44⁺. This means that there are no naïve cells in these lymph nodes. Again, this is not plausible.

Response 7:

We thank the reviewer for the comment. We apologize for the misunderstanding caused by insufficient information in “Fig. 5” and in figure legend of “Fig. 5”. The data in the “Fig. 5a, b, e” were from TDLNs, and the data in “Fig. 5c, d” were from tumors. To avoid the misunderstanding, we have added the source of the tissue used for flow cytometry in the annotation of Y-axis in “Fig. 5” and figure legend of “Fig. 5”, which were shown in Fig. R22.

And we also apologize for the incorrect gating strategies for Tcm showing in “Fig. 5e”. We have changed the gating strategy for “Fig. 5e” according to several references (Nat Immunol. 2016, 17, 1322-1333; Nat Commun. 2017, 8, 15338; Nat Commun. 2021, 12, 951)¹⁻³ and reanalyzed the results shown in Fig. R22e. The reanalyzed results did not change the conclusion from previous results that *“the central memory CD8⁺ T cell populations (CD8⁺ Tcm, CD44⁺CD62L⁺CD8⁺) in the TDLN of the mice in the RT+QD-Cat-RGD group were significantly increased on day 14 after treatment compared with the control group (Fig.5e)”*.

Revision made:

In Fig. 5, we have changed the annotation of Y-axis to:

“CD80⁺ and CD86⁺ in CD11c⁺ cells (%) in the TDLNs”;

“MHC-II^{high} in CD11c⁺ cells (%) in the TDLNs”;

“CD3⁺ in CD45⁺ cells (%) in the tumors”;

“CD8⁺ in CD45⁺ cells (%) in the tumors”;

“CD44⁺ and CD62L⁺ in CD8⁺ cells (%) in the TDLNs”.

In the revised manuscript, the analysis results in “Fig. 5e” were replaced to the re-analysis results as shown in Fig. R22e.

Fig. R22 QD-Cat-RGD-based RT for promoting DC maturation and antigen presentation in TDLNs and consequently inducing T cell-driven antitumor immunity. a Representative flow cytometric analysis images and relative quantification of co-stimulatory molecules CD80⁺ CD86⁺ gating on CD45⁺ CD11c⁺ cells in TDLNs. **b** Representative flow cytometric analysis images and relative quantification of MHC-II expression on CD11c⁺CD45⁺ cells in the TDLNs. **c** Representative flow cytometric analysis images and relative quantification of CD3⁺ T cells gating on CD45⁺ in tumors. **d** Representative immunofluorescence images of tumors showing CD8⁺ T cells and relative quantification of CD8⁺CD3⁺ T cells gating on CD45⁺ cells in tumors. Scale bars = 50 μ m. **e** Representative flow cytometric analysis images and relative quantification of CD44⁺CD62L⁺ Tcm cells

gating on CD8⁺CD3⁺CD45⁺ cells in the TDLNs. All data are presented as the mean ± s.e.m. (n = 5 per group). Statistical significance was calculated via one-way ANOVA with Tukey's multiple comparisons test. **P* < 0.05; ***P* < 0.01; ****P* < 0.001; *****P* < 0.0001.

Comment 8:

The dual flank tumor model is quite artificial. Delivery of a second tumor 7 days after the first results in concomitant immunity effects in the second tumor, and makes it highly responsive to therapy. This second tumor is injected only 1 day prior to therapy.

Response 8:

We very much appreciate this important observation and comment. According to the reviewer's suggestion, we prolonged the tumor-forming time of the second tumor to 6 days (Nat Nanotechnol. 2007, 12, 877-882; Nat Immunol. 2020, 21, 1160-1171)^{10,11} and repeated the experiments in the metastatic tumor model. The new results shown in Fig. R23 have replaced the old data in "Fig. 6" in the revised manuscript.

To establish the metastatic tumor model to reveal the abscopal effect, 4T1 cancer cells were first injected into the right hind flank of mice on day -8 as primary tumors and subsequently injected into the left hind flanks on day -6 as a distant tumor mimic (Fig. R23a) (Nat Nanotechnol. 2007, 12, 877-882; Nat Immunol. 2020, 21, 1160-1171)^{10,11}. On day 0, the mice were intravenously injected with QD-Cat-RGD (150 μL, 2 mg mL⁻¹), and the primary tumors were locally treated with RT (12 Gy) 2 h p.i. Then, the mice were treated with aPD-1 on day 1, 3 and 5 (Fig. R23a). Tumor growth curves showed that the primary tumor (right side) was inhibited by QD-Cat-RGD-based RT combined with aPD-1 treatment. Additionally, the growth of distant tumors (left side) was inhibited (Fig. R23b, c). Consistent with these results, the numbers of intratumoral T cells (CD3⁺) and cytotoxic T cells (CD8⁺CD3⁺) were significantly increased in distant tumors of the RT+QD-Cat-RGD+aPD-1 group (Fig. R24d-f). Granzyme B, interferon-γ (IFN-γ) and tumor necrosis factor-α (TNF-α), cytotoxic effectors mainly expressed by cytotoxic T cells, were also increased in distant tumors of the RT+QD-Cat-RGD+aPD-1 group, as measured by immunohistochemistry (IHC) and ELISA (Fig. R24g-i). This evidence demonstrated that QD-Cat-RGD-based RT combined with aPD-1 could intensely boost the abscopal effect and inhibit the growth of distant tumors.

Revision made:

In the revised manuscript, the new results as shown in Fig. R23 have replaced the old data as "Fig. 6" in the revised manuscript.

Fig. R23 QD-Cat-RGD-based RT in combination with aPD-1 for boosting the abscopal effect. **a** Schematic showing the experiment using QD-Cat-RGD-based RT combined with

aPD-1 to treat mice bearing 4T1 tumors on both sides. The tumor on the right side was designated as the primary tumor for in situ radiation therapy, and the tumor on the left side was designated as a distant tumor. BALB/c mice (n=5 per group) were implanted subcutaneously with 7×10^5 4T1 mammary carcinoma cells in the right hind flanks and after 2 days consequently implanted subcutaneously with 7×10^5 4T1 cells on left hind flanks. When the primary tumor volumes were 100 mm^3 , the mice received different treatments. 4T1 tumors were harvested on day 14. **b** Tumor growth curves of primary tumors on the right side in the different groups. Photographs of all primary tumors and tumor growth curves of tumors from individual mice are shown in Supplementary Fig. 11. **c** Tumor growth curves of distant tumors on the left side in the different groups. Photographs of all distant tumors and tumor growth curves of tumors from individual mice are shown in Supplementary Fig. 12. **d** Quantification of flow cytometric analysis of CD3^+ T cells gating on CD45^+ in the distant tumors of the different groups. **e** Quantification of flow cytometric analysis of $\text{CD8}^+\text{CD3}^+$ T cells gating on CD45^+ in distant tumors of the different groups. **f, g** Representative image of immunostaining of CD8 and granzyme B in distant tumors in the different groups. Scale bars = $50 \mu\text{m}$. **h, i** Cytokine levels of $\text{IFN-}\gamma$ and $\text{TNF-}\alpha$ in distant tumors after different treatments as measured by ELISA. All data are presented as the mean \pm s.e.m. (n = 5 per group). Statistical significance was calculated *via* one-way ANOVA with Tukey's multiple comparisons test. * $P < 0.05$; ** $P < 0.01$; *** $P < 0.001$; n.s., not significant.

In the revised manuscript, the statements about dual metastatic tumor model in Results were changed to:

“To investigate whether QD-Cat-RGD survived and reinforced the abscopal effect induced by localized RT combined with immunotherapy, we used an antibody against PD-1 (aPD-1) as an ICB to reveal this process. Then, we built a metastatic tumor model to reveal the abscopal effect. 4T1 cancer cells were first injected into the right hind flank of mice on day -8 as primary tumors and subsequently injected into the left hind flanks on day -6 as a distant tumor mimic (Fig. 6a). On day 0, the mice were intravenously injected with QD-Cat-RGD ($150 \mu\text{L}$, 2 mg mL^{-1}), and the primary tumors were locally treated with RT (12 Gy) 2 h p.i. Then, the mice were treated with aPD-1 on day 1, 3 and 5 (Fig. 6a). Tumor growth curves showed that the primary tumor (right side) was inhibited by QD-Cat-RGD-based RT combined with aPD-1 treatment (Fig. 6b, Supplementary Fig. 11). Additionally, the growth of distant tumors (left side) on the left site was inhibited (Fig. 6c and Supplementary Fig. 12). Consistent with these results, the numbers of intratumoral T cells (CD3^+) and cytotoxic T cells ($\text{CD8}^+\text{CD3}^+$) were significantly increased in distant tumors of the RT+QD-Cat-RGD+aPD-1 group (Fig. 6d-f). Granzyme B, interferon- γ ($\text{IFN-}\gamma$) and tumor necrosis factor- α ($\text{TNF-}\alpha$), cytotoxic effectors mainly expressed by cytotoxic T cells, were also increased in distant tumors of the RT+QD-Cat-RGD+aPD-1 group, as measured by immunohistochemistry (IHC) and ELISA (Fig. 6g-i)”.

Comment 9:

The response to treatment of the second tumor model does not correlate well with CD8 T cell infiltrate, though there is a better correlation with CD3 T cell infiltrate. These data suggest a non-CD8 T cell mechanism, which is not investigated.

Response 9:

We thank this reviewer for the valuable suggestion. As suggested, we attempted to further investigate the mechanisms contributing to the suppression of distant tumors. Multiple immunosuppressive cell types such as M2-like macrophages and Tregs combining with secreted immune inhibitory cytokines can hinder antitumor immunity mediated by T cells especially cytotoxic T cells^{12,13}. Encouraged by the results showing that QD-Cat-RGD could relieve hypoxic condition and reduce the infiltration of Tregs and M2-like macrophages in primary tumor (“Fig. 4e, f” in the manuscript), we tested the change in immunosuppressive effect mediated by Tregs and M2-like macrophages in the TME of distant tumor. The new results have been added to the revised manuscript.

The results showed that QD-Cat-RGD could also work in distant tumors. The intratumoral M2-like macrophages (CD206^{high}CD11b⁺F4/80⁺) (Fig. R24a), Tregs (CD4⁺CD25⁺Foxp3⁺) (Fig. R24b) and the level of IL-10 (Fig. R24c) were decreased in distant tumors of the RT+QD-Cat-RGD+aPD-1 group. Despite a negative correlation between tumor size and immunosuppressive cells and IL-10 (Fig. R24a, b), we could not ignore the antitumor effect from CD3⁺ T cells and CD8⁺ T cells because of the better correlation between tumor size and the cell number of CD3⁺ T cells or CD8⁺ T cells (Fig. R24d, e) as well as the protein levels of IFN- γ or TNF- α (Fig. R24f, g), two important effectors mainly from cytotoxic T cells.

Revision made:

In the revised manuscript, the new results as shown in Fig. R24a, b, c have added in the Supplementary Information as “Supplementary Fig. 13a, b, c” in the revised manuscript.

Fig. R24 **a** The quantification of M2-like macrophages (CD206^{high}) gating on F4/80⁺CD11b⁺CD45⁺ cells in distant tumors after different treatment. **b** The quantification of Tregs (CD25⁺ and Foxp3⁺) gating on CD4⁺CD3⁺CD45⁺ cells in distant tumors after different tumors. **c** Cytokine level of IL-10 in distant tumors after different tumors measured by ELISA. **d** The quantification of CD3⁺ in CD45⁺ cells in distant tumors after different treatment. **e** The quantification of CD8⁺CD3⁺ in CD45⁺ cells in distant tumors after different treatment. **f, g** Cytokine levels of IFN- γ and TNF- α in distant tumors after different treatments as measured by ELISA. Statistical significance was calculated *via* one-way ANOVA with Tukey's multiple

comparisons test. * $P < 0.05$; ** $P < 0.01$; *** $P < 0.001$.

In the revised manuscript, the new results were added in the Results section:

“We further investigated the mechanism contributing to the suppression of distant tumor. Encouraged by the results that QD-Cat-RGD could relieve hypoxic condition and reduce the infiltration of immunosuppressive cells in TME which could hinder antitumor immunity mediated by cytotoxic T cells^{52,53}, we tested the changes of Tregs, M2-like macrophages and IL-10 in the TME of distant tumor. QD-Cat-RGD could also work in distant tumors and the intratumoral M2-like macrophages, Tregs and the level of IL-10 were decreased in distant tumors in the RT + QD-Cat-RGD + aPD-1 group (Supplementary Fig. 13a-c).”

Comment 10:

The survival data following spontaneous metastases should offset the above concern, but strangely the authors administer metastatic tumors via IV injection 7 days after injecting the primary tumor. 4T1 is spontaneously metastatic, and should require no IV treatments. However, since the primary tumor treatment occurs at d8 after primary tumor inoculation, this may be before LN and metastatic spread has occurred. To genuinely model treatment of metastatic disease, therapies should be given to mice with established tumors and established metastases.

Response 10:

We appreciate the reviewer’s comment. We apologize for the incorrect statement about the mouse model. In this section, we actually used an experimental model of metastasis at early stage to evaluate this therapeutic strategy to inhibit the spread of metastasis in other organs and prolong survival.

Considering tumor metastasis is responsible for approximately 90% of cancer-related mortality, preventing the spread of metastasis is beneficial for prolonging overall survival of patients¹⁴. Encouraged by the increase in abscopal effect in the subcutaneous tumor model which implied a strong systemic antitumor immunity, we thereby evaluated this therapeutic strategy to inhibit early metastasis and prolong survival using an experimental model of metastasis at early stage¹⁵⁻¹⁷. To establish the model of metastasis at early stage, 4T1 cancer cells stably transfected with the luciferase gene (4T1-Luc) were injected into the right hind flank of mice as primary tumors. When the volume of the primary tumors reached approximately 100 mm³, 4T1-Luc cancer cells were then intravenously injected into the mice to mimic the escape of tumor cells from primary tumor site into the circulation at early stage and seeding in distant organs especially lung. These results further confirmed that QD-Cat-RGD-based RT combined with immunotherapy could efficiently inhibit metastasis at early stage and prolong overall survival.

We adjusted the description of this experimental model of metastasis at early stage in the revised manuscript to eliminate readers’ misunderstandings.

Revision made:

In the revised manuscript, the statements about the experimental early metastasis model were revised to:

“Considering that tumor metastasis is responsible for approximately 90% of cancer-related mortality, preventing the spread of metastasis is beneficial for prolonging the overall survival of patients⁵⁴. Encouraged by the increase in the abscopal effect in the subcutaneous tumor model, which implied strong systemic antitumor immunity, we evaluated this therapeutic strategy to inhibit early metastasis and prolong survival using an experimental model of metastasis at an early stage⁵⁵⁻⁵⁷. To establish the model of metastasis at an early stage, 4T1 cancer cells stably transfected with the luciferase gene (4T1-Luc) were injected into the right hind flank of mice as primary tumors. When the volume of the primary tumors reached approximately 100 mm³, 4T1-Luc cancer cells were then intravenously injected into the mice to mimic the escape of tumor cells from the primary tumor site into the circulation at an early stage and seeding in distant organs especially the lung”.

In the revised manuscript, the conclusion in this experimental model was revised to:

“These results further confirmed that QD-Cat-RGD-based RT combined with immunotherapy could efficiently inhibit metastasis at early stage and prolong overall survival”.

Comment 11:

The discussion states that RT and nanotechnology approaches have mainly ignored the effect on the immune system. This is unfair to the field. There are large numbers of papers with this as a focus, and the topic is widely reviewed.

Response 11:

We sincerely thank the reviewer for such careful reading and we apologize for the incorrect statement in Discussion.

We completely agreed with the reviewer that the immune system as a vital role through RT with nanotechnology have been widely reviewed. The incorrect statement has been corrected as follows: “At present, an increasing number of studies on RT and nanotechnology have gradually focused on influencing individual’s immune system to treat cancer (Nat Nanotechnol 12, 877-882; Nat Commun 11, 5687; Nat Commun 9, 2351; Nat Rev Immunol 20, 321-334)^{10,18-20”}.

Revision made:

In the Discussion section, we have corrected the wrong statement as follows:

“At present, an increasing number of studies on RT and nanotechnology have gradually focused on influencing individual’s immune system to treat cancer^{58-61”}.

Comment 12:

Minor issues

In figure 3, key facts are missing.

The dose of RT for 1a-j;

The timing of gH2ax assessment for 1c;

RT is poorly effective at inducing cell death in 4T1 in 1d. The dose used is very relevant.;

Response 12:

Thanks for reviewer's comments. The dose of RT used in Fig.3 is 6 Gy. And the timing of gH2ax assessment is at 4h after different treatment. We have added these key factors in the figure legend of Fig.3 according to reviewer's suggestion.

Revision made:

In the figure legend of "Fig. 3", the following statements were added:

"a 4T1 Cells were irradiated with X-ray doses to 6 Gy in vitro. Representative photographs of stained colonies of 4T1 cells treated with PBS, QD-RGD, QD-Cat-RGD, RT (6 Gy), RT (6 Gy) + QD-RGD or RT (6 Gy) + QD-Cat-RGD after 7 days".

"c Immunofluorescence staining of p-H2AX (Ser139) and quantitative analysis of foci density of foci per cell at 4 h after different treatments. Scale bars = 25 μm".

Comment 13:

The immunofluorescence data throughout is of limited value as presented. The figures are far too small to interpret, and are mostly black squares. These are all accompanied by quantitative analysis, so this is a minor issue. However, some other way to present these data is encouraged.

Response 13:

We thank the reviewer for the constructive suggestion. According to the suggestion, we have magnified immunofluorescence images in "Fig.3c, j" to better identify the cells in the images, which was shown in Fig. R25a, b.

Revision made:

The immunofluorescence images of "Fig. 3c, j" were magnified as shown in Fig. R25a, b.

Fig. R25 a Immunofluorescence staining of p-H2AX (Ser139) and quantitative analysis of foci density of foci per cell at 4 h after different treatments. Scale bars = 25 μ m. **b** Immunofluorescence staining of HMGB1 in 4T1 cells with different treatments after 24 h. Scale bars = 25 μ m.

Reference

- 1 Chinen, T. *et al.* An essential role for the IL-2 receptor in Treg cell function. *Nat Immunol* **17**, 1322-1333, doi:10.1038/ni.3540 (2016).
- 2 Kondo, T. *et al.* Notch-mediated conversion of activated T cells into stem cell memory-like T cells for adoptive immunotherapy. *Nat Commun* **8**, 15338, doi:10.1038/ncomms15338 (2017).
- 3 Li, Y. *et al.* Targeting IL-21 to tumor-reactive T cells enhances memory T cell responses and anti-PD-1 antibody therapy. *Nat Commun* **12**, 951, doi:10.1038/s41467-021-21241-0 (2021).
- 4 DePeaux, K. & Delgoffe, G. M. Metabolic barriers to cancer immunotherapy. *Nat Rev Immunol*, doi:10.1038/s41577-021-00541-y (2021).
- 5 Dong, Z., Yang, Z., Hao, Y. & Feng, L. Fabrication of H₂O₂-driven nanoreactors for innovative cancer treatments. *Nanoscale* **11**, 16164-16186, doi:10.1039/c9nr04418c (2019).
- 6 Chen, Q. *et al.* Nanoparticle-Enhanced Radiotherapy to Trigger Robust Cancer

- Immunotherapy. *Adv Mater* **31**, e1802228, doi:10.1002/adma.201802228 (2019).
- 7 Hermanson, G. T. *Bioconjugate Techniques*. (Academic Press, 2008).
- 8 Fleischer, J. *et al.* Differential expression and function of CD80 (B7-1) and CD86 (B7-2) on human peripheral blood monocytes. *Immunology* **89**, 592-598, doi:10.1046/j.1365-2567.1996.d01-785.x (1996).
- 9 Perfetto, S. P., Chattopadhyay, P. K. & Roederer, M. Seventeen-colour flow cytometry: unravelling the immune system. *Nat Rev Immunol* **4**, 648-655, doi:10.1038/nri1416 (2004).
- 10 Min, Y. *et al.* Antigen-capturing nanoparticles improve the abscopal effect and cancer immunotherapy. *Nat Nanotechnol* **12**, 877-882, doi:10.1038/nnano.2017.113 (2017).
- 11 Yamazaki, T. *et al.* Mitochondrial DNA drives abscopal responses to radiation that are inhibited by autophagy. *Nat Immunol* **21**, 1160-1171, doi:10.1038/s41590-020-0751-0 (2020).
- 12 Togashi, Y., Shitara, K. & Nishikawa, H. Regulatory T cells in cancer immunosuppression — implications for anticancer therapy. *Nature Reviews Clinical Oncology* **16**, 356-371, doi:10.1038/s41571-019-0175-7 (2019).
- 13 Mantovani, A., Marchesi, F., Malesci, A., Laghi, L. & Allavena, P. Tumour-associated macrophages as treatment targets in oncology. *Nature Reviews Clinical Oncology* **14**, 399-416, doi:10.1038/nrclinonc.2016.217 (2017).
- 14 Seyfried, T. N. & Huysentruyt, L. C. On the origin of cancer metastasis. *Crit Rev Oncog* **18**, 43-73, doi:10.1615/critrevoncog.v18.i1-2.40 (2013).
- 15 Gengenbacher, N., Singhal, M. & Augustin, H. G. Preclinical mouse solid tumour models: status quo, challenges and perspectives. *Nat Rev Cancer* **17**, 751-765, doi:10.1038/nrc.2017.92 (2017).
- 16 Headley, M. B. *et al.* Visualization of immediate immune responses to pioneer metastatic cells in the lung. *Nature* **531**, 513-517, doi:10.1038/nature16985 (2016).
- 17 Li, P. *et al.* Lung mesenchymal cells elicit lipid storage in neutrophils that fuel breast cancer lung metastasis. *Nat Immunol* **21**, 1444-1455, doi:10.1038/s41590-020-0783-5 (2020).
- 18 Gregory, J. V. *et al.* Systemic brain tumor delivery of synthetic protein nanoparticles for glioblastoma therapy. *Nat Commun* **11**, 5687, doi:10.1038/s41467-020-19225-7 (2020).
- 19 Ni, K. *et al.* Nanoscale metal-organic frameworks enhance radiotherapy to potentiate checkpoint blockade immunotherapy. *Nat Commun* **9**, 2351, doi:10.1038/s41467-018-04703-w (2018).
- 20 Irvine, D. J. & Dane, E. L. Enhancing cancer immunotherapy with nanomedicine. *Nat Rev Immunol* **20**, 321-334, doi:10.1038/s41577-019-0269-6 (2020).

Reviewers' Comments:

Reviewer #1:

Remarks to the Author:

I think this work with revision is now acceptable for publication.

Reviewer #2:

Remarks to the Author:

The authors have performed additional experiments and controls to strengthen their study. They have addressed all the points that I had raised during the first round of evaluation. I recommend publication.

Reviewer #3:

Remarks to the Author:

The authors have been highly responsive to review. They have performed additional experiments that clarify or prove mechanisms, and made corrections to figures where the information was unclear.

This reviewer still has some issues with the overall novelty in the manuscript.

More importantly, some issues with the mechanisms that have not been solved in the revision.

1. The question of whether the nanoprobe reach the cancer cells in vivo remains unproven, as the authors have not fully demonstrated that fact. The RGD motifs are vascular lumen-targeting, and the fact that RGD-tagged nanoprobe are located in the interior of the tumor mass does not mean that they have crossed the vasculature. It should be noted that the tumor stroma isn't the tumor margin as suggested by the authors when discussing Figure 2c, but rather the normal cell infiltrates surrounding and supporting vasculature within the tumor mass. To impact cancer cell hypoxia, the nanoprobe must pass through the vasculature and traverse the stroma to the hypoxic regions. We already know that oxygen struggles to sustain this zone – why are the nanoprobe more able to enter hypoxic regions than oxygen?

2. The fact that the catalase-loaded materials also help kill cancer cells in vitro under hypoxic conditions does not mean that they help kill cancer cells by overcoming hypoxia. The original data shows that they also help kill cancer cells under normoxic conditions. In fact, comparing the original data from the first submission and the hypoxia data from the resubmission suggests that radiation alone is more effective under hypoxia than under normoxia (Original RT alone 9% apoptotic, New hypoxia RT alone 31% apoptotic, though the baselines are different. Figure 3d). The DNA damage foci suggest similar issues on comparison. This is problematic. For this reason, the original normoxic data should not be removed from the manuscript, and the fact that the agent is effective under normoxic conditions should be addressed directly.

Point-to-point responses to the comments

Reviewer #3

The authors have been highly responsive to review. They have performed additional experiments that clarify or prove mechanisms, and made corrections to figures where the information was unclear.

This reviewer still has some issues with the overall novelty in the manuscript. More importantly, some issues with the mechanisms that have not been solved in the revision.

Response:

We are very grateful to the reviewer for the careful reading and constructive comments. We have carefully considered the referee's comments and revised the manuscript accordingly and believe that our manuscript has been significantly strengthened. We have added additional experimental details in the Methods section. All revisions are highlighted in red color in the revised manuscript and Supplementary Information.

Comment 1:

1. The question of whether the nanoprobos reach the cancer cells in vivo remains unproven, as the authors have not fully demonstrated that fact. The RGD motifs are vascular lumen-targeting, and the fact that RGD-tagged nanoprobos are located in the interior of the tumor mass does not mean that they have crossed the vasculature. It should be noted that the tumor stroma isn't the tumor margin as suggested by the authors when discussing Figure 2c, but rather the normal cell infiltrates surrounding and supporting vasculature within the tumor mass. To impact cancer cell hypoxia, the nanoprobos must pass through the vasculature and traverse the stroma to the hypoxic regions. We already know that oxygen struggles to sustain this zone – why are the nanoprobos more able to enter hypoxic regions than oxygen?

Response 1:

Thanks for raising this important concern. We apologized for the wrong statement about tumor stroma in “Fig. 2c” and redefined the region that the normal cell infiltrates surrounding and supporting vasculature within the tumor mass as the tumor stroma in the manuscript according to reviewer's comment.

According to reviewer's suggestion, we further explored whether the nanoprobos can reach the cancer cells *in vivo*. For revealing this point, 4T1 tumor-bearing BALB/c mice were intravenously injected with QD-Cat-RGD (150 μ L, 2 mg mL⁻¹) or QD-Cat (without RGD) (150 μ L, 2 mg mL⁻¹) nanoprobos. Then, we sorted the live CD45⁻ cells which were mainly cancer cells from tumor tissue by fluorescent-activated cell sorting (FACS; the sorting gating strategy is shown in Fig. R1a), and investigated the content of nanoprobe associated element Pb in the cells by inductively coupled plasma optical emission spectrometry (ICP-OES). The results showed that the content of Pb in the QD-Cat-RGD group were higher than that in QD-Cat group (Fig. R1b), which suggested that the RGD modification significantly improved the targeting efficiency of nanoprobos to cancer cells *in vivo*. The main reason may be that Arg-Gly-Asp (RGD) motifs have high affinity to integrin $\alpha_v\beta_3$, which not only overexpresses on the tumor vasculature but also on the cancer cells^{1,2}. Several studies have proven that RGD can specifically bind to cancer cells including 4T1 and B16F10 cells^{3,4,5,6,7}. Our data also indicated that the QD-Cat-RGD nanoprobos can bind to integrin $\alpha_v\beta_3$ -positive 4T1 cancer

cells *in vitro* (Fig. 1c in the manuscript) and achieve higher efficiency to 4T1 tumor in mice (Fig. 2f in the manuscript). It is also proven by previous studies that nanoprobe with RGD modification can effectively reach the tumor site and consequently bind with the cancer cells^{8, 9, 10, 11, 12}.

To further reveal the relationship between location of QD-Cat-RGD nanoprobes and hypoxic region in tumor, serial slices of tumor tissue from QD-Cat-RGD group were imaged in the NIR-IIb window, stained with hematoxylin and eosin (H&E), and immunostained with HIF-1 α by immunohistochemistry (IHC). As shown in Fig. R2, the fluorescent signals in NIR-IIb can be found in the Region 1 in the tumor (Fig. R2d), which was near the stroma infiltrated with several vasculatures (Fig. R2e) and with weak nuclear HIF-1 α immunostaining (Fig. R2g). Moreover, the fluorescent signals in NIR-IIb can also be found in the Region 2 in the tumor (Fig. R2d), which was away from vasculatures (Fig. R2f) and with strong nuclear HIF-1 α immunostaining (Fig. R2h). These data suggested that partial QD-Cat-RGD nanoprobes can enter the hypoxic region of tumor *in vivo*.

Revision made:

In the revised manuscript, the statements about the tumor stroma in “Fig. 2c” were revised to:

“The merged images showed that the QD-Cat-RGD nanoprobes with RGD modification could deeply infiltrate in the tumor tissue (Fig. 2c). Without RGD modification, QD-Cat nanoprobes were mainly found in the stroma which was infiltrated with supporting vasculatures (Fig. 2c)”.

In the revised manuscript, the following statements were added in the Results section:

*“Moreover, we sorted the live CD45⁺ cells which were mainly cancer cells from tumor tissue by fluorescent-activated cell sorting (FACS; the sorting gating strategy is shown in Supplementary Fig. 3a), and investigated the content of nanoprobe associated element Pb in these cells by inductively coupled plasma optical emission spectrometry (ICP-OES). The results showed that the content of Pb in the QD-Cat-RGD group were higher than that in QD-Cat group (Supplementary Fig. 3b), which suggested that the RGD modification significantly improved the targeting efficiency of nanoprobes to cancer cells *in vivo*.”*

In the revised manuscript, we have added the data of Fig. R1 as “Supplementary Fig. 3” in the Supplementary Information.

Fig. R1 (Supplementary Fig. 3) Tumor tissue harvested from 4T1 tumor-bearing mice which intravenously injected with QD-Cat-RGD (150 μL , 2 mg mL^{-1}) or QD-Cat (150 μL , 2 mg mL^{-1}) nanoprobe at 4h. Then the live CD45⁺ cells were sorted by FACS. **a** The sorting gating strategy of cells. **b** Content of Pb in the live CD45⁺ measured by ICP-OES. All data are shown as the mean \pm s.e.m. of three replicates. Statistical significance was calculated *via* Student's *t* test. * $P < 0.05$.

We have also added the method about FACS in the Methods:

“To isolate the live CD45⁺ cells from tumor tissue, single cell suspension of tumor tissue was stained with the Fixable Viability Dye (eBioscience, eFluor 780) and anti-CD45 (eBioscience, perCP-Cyanine5.5, clone 30-F11) antibody. Live CD45⁺ cells (the sorting gating strategy is shown in Supplementary Fig. 3a) were sorted on a MoFlo XDP cell sorter (Beckman) in PBS with 2% FBS.”

In the revised manuscript, the following statements were added in the Results section:

“To further reveal the relationship between location of QD-Cat-RGD nanoprobe and hypoxic region in tumor, serial slices of tumor tissue from QD-Cat-RGD group were imaged in the NIR-IIb window, stained with H&E and immunostained with HIF-1 α by immunohistochemistry (IHC). As shown in Supplementary Fig. 4, the fluorescent signals in NIR-IIb can be found in the Region 1 in the tumor (Supplementary Fig. 4d), which was near the stroma infiltrated with several vasculatures (Supplementary Fig. 4e) and with weak nuclear HIF-1 α immunostaining (Supplementary Fig. 4g). Moreover, the fluorescent signals in NIR-IIb can also be found in the Region 2 in the tumor (Supplementary Fig. 4d), which was away from vasculatures (Supplementary Fig. 4f) and with strong nuclear HIF-1 α immunostaining (Supplementary Fig. 4h). These results suggested that QD-Cat-RGD nanoprobe with the targeting property of RGD could migrate through the vasculature and deeply infiltrate into the hypoxic tumor tissue.”

In the revised manuscript, we have added the data of Fig. R2 as “Supplementary Fig. 4” in the Supplementary Information.

Fig. R2 (Supplementary Fig. 4) Tumor tissue harvested from 4T1 tumor-bearing mice which were intravenously injected with QD-Cat-RGD nanoprobe at 4h. **a-d** Serial sections of tumor tissue imaged in the NIR-IIb window, stained with H&E and immunostained with HIF-1 α . The excitation power density for NIR-IIb fluorescence imaging was 50 mW cm^{-2} provided by an 808 nm laser. Scale bars = 1 mm. **e, g** H&E staining (**e**) and immunohistochemical staining with HIF-1 α (**g**) of the Region 1. Scale bars = 100 μm . **f, h** H&E staining (**f**) and

immunostaining with HIF-1 α (h) of the Region 2. Scale bars = 100 μ m.

Comment 2:

2. The fact that the catalase-loaded materials also help kill cancer cells *in vitro* under hypoxic conditions does not mean that they help kill cancer cells by overcoming hypoxia. The original data shows that they also help kill cancer cells under normoxic conditions. In fact, comparing the original data from the first submission and the hypoxia data from the resubmission suggests that radiation alone is more effective under hypoxia than under normoxia (Original RT alone 9% apoptotic, New hypoxia RT alone 31% apoptotic, though the baselines are different. Figure 3d). The DNA damage foci suggest similar issues on comparison. This is problematic. For this reason, the original normoxic data should not be removed from the manuscript, and the fact that the agent is effective under normoxic conditions should be addressed directly.

Response 2:

We appreciate the reviewer’s comment. According to the reviewer’s suggestion, we have added the original normoxic data and indicated the fact that the nanoprobe is effective under normoxic conditions in the manuscript. The format of the figure has been modified to meet the requirements from the editors and reviewers.

Revision made:

In the revised manuscript, the original normoxic data of Fig. R3 were added as “Supplementary Fig.5” in the Supplementary Information.

Fig. R3 (Supplementary Fig. 5) QD-Cat-RGD as a radiosensitizer under normoxic conditions. 4T1 cells were incubated under normoxic conditions (37 °C, 20% O₂, 5% CO₂) and irradiated with X-ray doses of 6 Gy *in vitro*. **a** Representative photographs of stained colonies of 4T1 cells treated with normoxic, normoxic + QD-RGD, normoxic + QD-Cat-RGD, normoxic + RT (6 Gy), normoxic + RT + QD-RGD (6 Gy) or normoxic + RT + QD-Cat-RGD

(6 Gy) after 7 days. **b** Histogram plot of the survival fraction of 4T1 cells with different treatments. **c** Immunofluorescence staining of p-H2AX (Ser139) and quantitative analysis of foci density of foci per cell at 4 h after different treatments. Scale bars = 25 μ m. **d** Apoptosis analysis measured by flow cytometry of 4T1 cells at 24 h after different treatments. All data are shown as the mean \pm s.e.m. of three replicates. Statistical significance was calculated via Student's *t* test (**b, c**). **P* < 0.05; ***P* < 0.01; ****P* < 0.001; n.s., not significant.

In the revised manuscript, the following statements were added in the Results section:

“We also tested the radiosensitization of QD-Cat-RGD nanoprobe under normoxic conditions and found that the nanoprobe also had radiosensitizing effect under normoxic conditions (Supplementary Fig. 5).”

References

1. Danhier F, Le Breton A, Preat V. RGD-based strategies to target $\alpha(v)\beta(3)$ integrin in cancer therapy and diagnosis. *Mol Pharm* **9**, 2961-2973 (2012).
2. Cheng TM, *et al.* Nano-Strategies Targeting the Integrin $\alpha v\beta 3$ Network for Cancer Therapy. *Cells* **10**, (2021).
3. Huang CH, *et al.* Combined Treatment of Heteronemin and Tetrac Induces Antiproliferation in Oral Cancer Cells. *Mar Drugs* **18**, (2020).
4. Hsieh MT, *et al.* Crosstalk between integrin $\alpha v\beta 3$ and ER α contributes to thyroid hormone-induced proliferation of ovarian cancer cells. *Oncotarget* **8**, 24237-24249 (2017).
5. Chin YT, *et al.* Tetrac and NDAT Induce Anti-proliferation via Integrin $\alpha v\beta 3$ in Colorectal Cancers With Different K-RAS Status. *Front Endocrinol (Lausanne)* **10**, 130 (2019).
6. Guo Z, *et al.* Targeting efficiency of RGD-modified nanocarriers with different ligand intervals in response to integrin $\alpha v\beta 3$ clustering. *Biomaterials* **35**, 6106-6117 (2014).
7. Wang L, Zhang T, Huo M, Guo J, Chen Y, Xu H. Construction of Nucleus-Targeting Iridium Nanocrystals for Photonic Hyperthermia-Synergized Cancer Radiotherapy. *Small* **15**, e1903254 (2019).
8. Pasqualini R, Koivunen E, Ruoslahti E. αv integrins as receptors for tumor targeting by circulating ligands. *Nat Biotechnol* **15**, 542-546 (1997).
9. Liu Z, *et al.* In vivo biodistribution and highly efficient tumour targeting of carbon nanotubes in mice. *Nat Nanotechnol* **2**, 47-52 (2007).
10. Xiong L, Shuhendler AJ, Rao J. Self-luminescing BRET-FRET near-infrared dots for in vivo lymph-node mapping and tumour imaging. *Nat Commun* **3**, 1193 (2012).
11. Choi HS, *et al.* Targeted zwitterionic near-infrared fluorophores for improved optical imaging. *Nat Biotechnol* **31**, 148-153 (2013).
12. Fan Z, *et al.* Near infrared fluorescent peptide nanoparticles for enhancing esophageal cancer therapeutic efficacy. *Nat Commun* **9**, 2605 (2018).

Reviewers' Comments:

Reviewer #3:

Remarks to the Author:

The authors have been highly responsive. At this point it is fair to say that they have proven that this reviewer's concerns are unfounded. As a reviewer I really appreciate the extra effort that the authors have put into this work.

I have no further concerns.

Reviewer #4:

Remarks to the Author:

In the revised manuscript, the authors have performed additional experiments to prove the mechanisms of tumoral penetration and cell uptake of QD-Cat-RGD. I agree that the RGD peptide can help QD-Cat-RGD to be taken up and accumulated by tumor cells. However, the evidence that QD-Cat-RGD deeply infiltrates into the hypoxic tumor tissue is still insufficient. Moreover, I also agree with the other reviewers that the overall novelty of the manuscript is insufficient.

1. In this study, the PbS/CdS QDs with high toxic Pb and Cd metals were used to prepare QD-Cat-RGD. Although the authors have proven that QD-Cat-RGD has good biocompatibility in cells and mouse, there is still a great safety risk because QD may be degraded in the body. Therefore, the prospect of clinical transformation of QD-Cat-RGD is not promising.

2. QD-drug-RGD system has been extensively studied for tumor targeting and therapy in the past ten years. For material, the novelty of this work is insufficient. In addition, the therapeutic effect for tumors is mainly derived from the combination of immunotherapy and radiotherapy, which has also been confirmed by many researches and clinical trials.

3. The evidence that QD-Cat-RGD deeply infiltrates into the hypoxic tumor tissue is still insufficient. It is known that tumor tissue away from blood vessels is more prone to hypoxia. This study only uses the RGD-based tumor blood vessel targeting strategy, it is difficult to understand why QD-Cat-RGD can penetrate deeply into the hypoxic area of the tumor. In addition, in Supplementary Fig. 4, I did not find obvious hypoxic areas in the tumor section. I believe that QD-Cat-RGD mainly increases the concentration of the catalase in tumors, thereby helping to improve the therapeutic effect. Authors should be more careful in describing their conclusions.

Point-to-point responses to the comments

Reviewer #4

In the revised manuscript, the authors have performed additional experiments to prove the mechanisms of tumoral penetration and cell uptake of QD-Cat-RGD. I agree that the RGD peptide can help QD-Cat-RGD to be taken up and accumulated by tumor cells. However, the evidence that QD-Cat-RGD deeply infiltrates into the hypoxic tumor tissue is still insufficient. Moreover, I also agree with the other reviewers that the overall novelty of the manuscript is insufficient.

Response:

We are very grateful to the reviewer for the careful reading and constructive comments. We have carefully considered the referee's comments and discussed the points raised by the reviewer. All revisions are shown with tracked changes in the revised manuscript.

Comment 1:

1. In this study, the PbS/CdS QDs with high toxic Pb and Cd metals were used to prepare QD-Cat-RGD. Although the authors have proven that QD-Cat-RGD has good biocompatibility in cells and mouse, there is still a great safety risk because QD may be degraded in the body. Therefore, the prospect of clinical transformation of QD-Cat-RGD is not promising.

Response 1:

Thanks for raising this important concern. In this study, for the problems in the application of radiotherapy, we propose a strategy that simultaneously combines precision imaging of radiotherapy and enhancement of radiotherapy efficacy and promotion of anti-tumor immune responses. Then, we develop the nanoprobe which based on PbS/CdS QDs to achieve these functions. We agree the reviewer's viewpoint that the toxicity of Pb and Cd metals limits its clinical transformation, but this strategy still provides good prospects for the future RT which can combine precision RT with enhancement of radiotherapy efficacy and promotion of anti-tumor immune responses for patients with cancer. According to the suggestion, we have discussed the points raised by the reviewer and acknowledged the limitations in the Discussion section.

Revision made:

In the Discussion section, we have added the statement as follows:

“The potential toxic elements in QD-Cat-RGD may limit the clinical transformation of QD-Cat-RGD in RT. This strategy still provides good prospects for the future RT to enhance the therapeutic precision and efficacy of RT and promote anti-tumor immune response, prolonging survival and improving the quality of life of patients with cancer.”

Comment 2:

2. QD-drug-RGD system has been extensively studied for tumor targeting and therapy in the past ten years. For material, the novelty of this work is insufficient. In addition, the therapeutic effect for tumors is mainly derived from the combination of immunotherapy and radiotherapy, which has also been confirmed by many researches and clinical trials.

Response 2:

Thanks for reviewer's comment. In this study, based on the nanotechnology and multifunction of nanoparticles, we therefore can ameliorate the problems of radiotherapy in many fields including accurate target volume delineation and relief of hypoxia in the tumor microenvironment. Here, we design a nanoprobe based on NIR-IIb emitting quantum dots as radiosensitizers and nanoagents for image-guided RT, and modified with catalase to relieve hypoxia in the tumor microenvironment, enhancing the precision and efficacy of RT and promoting anti-tumor immune responses. Our strategy that combined radiotherapy and QD-Cat-RGD nanoprobe can boost the antitumor immune response and bring the more efficient abscopal effect against metastasis compared with radiotherapy when combined with immunotherapy.

Comment 3:

3. The evidence that QD-Cat-RGD deeply infiltrates into the hypoxic tumor tissue is still insufficient. It is known that tumor tissue away from blood vessels is more prone to hypoxia. This study only uses the RGD-based tumor blood vessel targeting strategy, it is difficult to understand why QD-Cat-RGD can penetrate deeply into the hypoxic area of the tumor. In addition, in Supplementary Fig. 4, I did not find obvious hypoxic areas in the tumor section. I believe that QD-Cat-RGD mainly increases the concentration of the catalase in tumors, thereby helping to improve the therapeutic effect. Authors should be more careful in describing their conclusions.

Response 3:

Thanks for reviewer's suggestion. We have carefully adjusted our conclusions about Supplementary Fig.4 according to reviewer's suggestion.

Revision made:

In the Result section, we have added the statement as follows:

“These results suggested that QD-Cat-RGD nanoprobe with the targeting property of RGD might diffuse in the tumor tissue.”